# Enhanced ice nucleation activity of coal fly ash aerosol particles initiated by ice-filled pores

Nsikanabasi Silas Umo[1], Robert Wagner[1], Romy Ullrich[1], Alexei Kiselev[1], Harald Saathoff[1], Peter G. Weidler[2], Daniel J. Cziczo[3,4], Thomas Leisner[1], and Ottmar Möhler[1]

[1]Institute of Meteorology and Climate Research - Atmospheric Aerosol Research, Karlsruhe Institute of Technology, Hermann-von-Helmholtz Platz 1, 76344 Eggenstein-Leopoldshafen, Germany
[2]Institute of Functional Interfaces, Karlsruhe Institute of Technology, Hermann-von-Helmholtz Platz 1, 76344 Eggenstein-Leopoldshafen, Germany
[3]Earth, Atmospheric and Planetary Sciences, Civil and Environmental Engineering, Massachusetts Institute of Technology, 77 Massachusetts Avenue 54-1324, Cambridge, MA 02139-4307, USA
[4]Now at: Purdue University, Department of Earth, Atmospheric and Planetary Sciences, 550 Lafayette St., West Lafayette, IN 47907, USA

*Correspondence to:* Nsikanabasi Silas Umo (nsikanabasi.umo@partner.kit.edu)

**Abstract.** Ice-nucleating particles (INPs), which are precursors for ice formation in clouds, can alter the microphysical and optical properties of clouds, hence, impacting the cloud lifetimes and hydrological cycles. However, the mechanisms with which these INPs nucleate ice when exposed to different atmospheric conditions are still unclear for some particles. Recently, some INPs with pores or permanent surface defects of regular or irregular geometries have been reported to initiate ice formation at cirrus temperatures via the liquid phase in a two-step process, involving the condensation and freezing of supercooled water inside these pores. This mechanism has therefore been labelled as pore condensation and freezing (PCF). The PCF mechanism allows formation and stabilization of ice germs in the particle without the formation of macroscopic ice. Coal fly ash (CFA) aerosol particles are known to nucleate ice in the immersion freezing mode and may play a significant role in cloud formation. In our current ice nucleation experiments with a particular CFA sample (CFA_UK), which we conducted in the Aerosol Interaction and Dynamics in the Atmosphere (AIDA) aerosol and cloud simulation chamber at the Karlsruhe Institute of Technology, Germany, we observed a strong increase (at a threshold relative humidity with respect to ice of 101 – 105 %) in the ice-active fraction for experiments performed at temperatures just below the homogeneous freezing of pure water. This observed strong increase in the ice-active fraction could be related to the PCF mechanism. To further investigate the potential of CFA particles undergoing the PCF mechanism, we performed a series of temperature-cycling experiments in AIDA. The temperature-cycling experiments involve exposing CFA particles to lower temperatures (down to ~ 228 K), then warming them up to higher temperatures (238 K – 273 K) before investigating their ice nucleation properties. For the first time, we report the enhancement of the ice nucleation activity of the CFA particles for temperatures up to 263 K, from which we conclude that it is most likely due to the PCF mechanism. This indicates that ice germs formed in the CFA particles' pores during cooling remain in the pores during warming and induce ice crystallization as soon as the pre-activated particles experience ice-supersaturated conditions at higher temperatures; hence, showing an enhancement in their ice-nucleating ability compared to the scenario where the CFA particles are directly probed at higher temperatures without temporary cooling. The enhancement in the ice nucleation ability showed a positive correlation with the specific surface area and porosity of the particles. On the one hand, the PCF mechanism can play a significant role in mixed-phase cloud formation in a case where the CFA particles are injected from higher altitudes and then transported to lower altitudes after being exposed to lower temperatures. On the other hand, the PCF mechanism could be the prevalent nucleation mode for ice formation at cirrus temperatures rather than the previously acclaimed deposition mode.

# 1.    Introduction

Understanding the ice nucleation processes remains highly relevant to our knowledge of cloud formation and other applications in cryopreservation, geoengineering, bioengineering, material modifications, aviation, and in agriculture (Kiani and Sun, 2011; Morris and Acton, 2013; Murray, 2017). Ice nucleation by aerosol particles is known to modify cloud properties, hence, playing an important role in modulating the hydrological cycle and climate (Boucher et al., 2013; Seinfeld and Pandis, 2006). Homogeneous ice nucleation occurs when water droplets freeze without the aid of a particle, whereas, when a particle catalyses this process, it is referred to as heterogeneous ice formation (Vali et al., 2015). There are four mechanisms identified for heterogeneous ice nucleation in the atmosphere, which are the immersion, condensation, deposition, and contact modes (Pruppacher and Klett, 2010; Young, 1993). Immersion freezing occurs when an ice-nucleating particle (INP) initiates ice formation when completely immersed in a cloud droplet. Condensation freezing happens when ice nucleates as water is condensed on the INP, whereas deposition nucleation occurs when water vapour directly forms the ice phase on a particle. Contact freezing is triggered when an INP comes in contact with the surface of a supercooled water droplet (from inside or outside) to initiate nucleation and subsequent freezing (Pruppacher and Klett, 2010; Vali et al., 2015). While immersion freezing is relevant in mixed-phase clouds (Murray et al., 2012), the deposition mode mechanism and homogeneous ice nucleation dominate cirrus cloud formation (Hoose and Möhler, 2012).

There is an ongoing debate on whether the direct deposition of water vapour on the surface of an INP is the real process behind ice formation, or whether it is rather the freezing of supercooled liquid water in the pores of such particles that later grows to form a macroscopic ice crystal (Marcolli, 2014 and references therein). The mechanism is termed as the pore condensation and freezing (PCF) process. PCF involves a two-step process – first, capillary condensation of liquid water in the particle pores, and second, freezing of the condensed water. The first step occurs when particles with pores are exposed to a certain relative humidity ($RH_\mathrm{w}$) below water saturation ($RH_\mathrm{w} < 100\ \%$). The $RH_\mathrm{w}$ for pore filling to occur is well-described by the 'negative' Kelvin effect (Fisher et al., 1981). The negative exponential term of the Kelvin equation accounts for the concave meniscus of the condensed water in a pore (Sjogren et al., 2007). When pores with condensed water (step 1) are exposed to sufficiently low temperatures, ice can form in such pores. In an ice-supersaturated environment, these ice-filled pores can then initiate the growth of macroscopic ice crystals on the particles. Ice-filled particle pores can then act as active sites for ice nucleation and growth in an ice-supersaturated environment. In a situation where ice-filled pores (step 2) are preserved even when the system is warmed, they can trigger ice nucleation at higher temperatures. This process is relevant for understanding ice nucleation by porous particles or particles with surface defects. Surface defects on particles such as pores, cavities, cracks, crevices or specific features such as voids, holes, fissures on particles will hereafter be referred to as "pores".

The PCF mechanism is restricted to a certain pore size range due to limitations related to the negative Kelvin effect for water condensation in the pores and the size of the critical ice embryo for ice nucleation and melting. According to classical nucleation theory, a certain critical ice embryo size is required to overcome the energy barrier defined by the Gibbs free energy (Pruppacher and Klett, 2010). Therefore, the pore size should be large enough to accommodate such a critical ice embryo and small enough to enable the capillary condensation of water in the first place. Calculations and previous reports have shown that pore sizes with $3 – 8$ nm diameter are suitable for the PCF mechanism (Wagner et al, 2016, Marcolli, 2017). Also, pore geometry (e.g., cylindrical or ink-bottle-shaped pores) has been shown to be an important parameter for the initial step of the PCF mechanism (Marcolli, 2014, 2017). Moreover, the contact angle between the pore wall and the water curvature affects the onset of the capillary condensation of water according to the Kelvin equation.

The PCF mechanism had already been proposed in the past (e.g. Fukuta, 1966), but recently, there has been renewed interest in understanding this mechanism with more sophisticated experiments (David et al., 2019; Marcolli, 2017 and references therein). Generally, recent studies have suggested that surface defects and pore properties are crucial factors in determining the ice nucleation mechanism of aerosol particles (Campbell et al., 2017; Campbell and Christenson, 2018; He et al., 2018; Kiselev et al., 2016; Li et al., 2018; Whale et al., 2017). For INPs with pores to pre-activate in the atmosphere, the INPs need to undergo some level of processing at different atmospheric conditions before ice nucleation takes place. Here, we define pre-activation as the process whereby ice germs are formed in the particle pores when such particles are temporarily exposed to a lower temperature (Wagner et al., 2016). In addition, the recycling of aerosol particles through regions of varying relative humidity in the atmosphere could also influence their ice nucleation mechanisms (Heymsfield et al., 2005; Knopf and Koop, 2006). Some laboratory experiments have been carefully performed to investigate the pre-activation processes to gain a better understanding of the possible scenarios when the PCF mechanism can contribute to pre-activation. In such experiments, pre-activated pores in the particles have been observed to enhance the particles' ice-nucleating properties (Marcolli, 2017; Wagner et al., 2016). For example, Wagner et al. (2016) reported pre-activation of various particles such as zeolite, illite, desert dust from Israel and Arizona, soot, and Icelandic volcanic ash by the PCF mechanism. These particles all showed varying degree of improvement in their inherent ice nucleation abilities via the PCF mechanism. The ice formation via this mechanism is restricted to a certain pore size range ($5-8$ nm) (Wagner et al., 2016). Aside from pre-existing porous materials, organic-containing aerosol particles such as ultra-viscous or glassy aerosols have shown a considerable augmentation in their ice nucleation activities when pre-processed in clouds (Wagner et al., 2012). This is attributed to the formation of porous particles during the ice-cloud processing. These studies established that in clouds, ice can easily form on pre-activated particles by depositional growth at $RH_{ice} > 100$ % without any specific activation threshold. In contrast, definite ice active sites are required for a classical deposition nucleation process to occur. However, it is not yet clear how this mechanism takes place.

A better understanding of the PCF mechanism by different INPs can provide better insights into the potential contributions of these INPs to the global cloud ice budget. Coal fly ash (CFA) is one group of aerosol particles that are constantly emitted into the atmosphere from the energy production by coal burning (Manz, 1999). About 500 - 800 million tonnes of CFA aerosol particles are produced annually (Adams, 2017; Heidrich et al., 2013; Joshi and Lohita, 1997) and a significant amount of this proportion is injected into the atmosphere – hence, they could contribute to heterogeneous ice formation in clouds. Previously, CFA particles have been shown to nucleate ice in the immersion mode (Grawe et al., 2016, 2018; Umo et al., 2015). Grawe et al. (2018) partly attributed the ice nucleation behaviour of the CFA particles in the immersion freezing mode to the quartz content of the CFA particles. The influence of this quartz content on the particles' immersion freezing ability can be suppressed in a situation where hydratable components form a layer on the particle surface (Grawe et al., 2018). These hydratable components are chemical compounds (e.g. $CaSO_4$) contained in CFA particles that are capable of taking up water at elevated ambient relative humidity. This can lead to the formation of new compounds such as calcite and gypsum. There are large variabilities in the ice nucleation activities of the different CFA samples reported, which could be due to the difference in the mineralogical or chemical compositions, and the extent to which these particles are processed in the atmosphere (Grawe et al., 2018; Losey et al., 2018). The ice-nucleating behaviour of CFA particles, when exposed to various temperature and relative humidity conditions, is still unclear and requires further investigations.

In this study, we investigated the ice nucleation behaviour of different CFA samples at temperatures higher than 238 K. When we tested the ice nucleation ability of these particles at temperature just below the homogeneous freezing of pure water, one of the CFA samples showed a high fraction of ice-active particles at a low relative humidity with respect to ice ($RH_{ice}$ = 101 – 105 %), in apparent contrast to its ice-nucleating ability just above 238 K. This result was indicative of a PCF mechanism as put forward by Marcolli (2014), noting that a variety of aerosol particle types showed a sudden increase in their IN ability just below the homogeneous freezing temperatures. Following our preliminary observations, we decided to prove whether the CFA particles are also prone to the PCF mechanism by adopting a temperature-cycling protocol which is described in full in Section 2.6. We report the ice nucleation behaviour of different CFA aerosol samples when temporarily exposed to lower temperatures at ice-subsaturated conditions and then probed at higher temperatures. The results were then compared to their inherent ice-nucleating abilities at similar temperatures to understand the potential freezing mechanism by CFA in such conditions. Our article is organized in the following sections: the experimental procedure adopted for this study, the description of the results, and the potential atmospheric implications of the new results to ice formation in mixed-phase clouds as well as possible pathways in cirrus clouds. The article concludes by pointing out some future perspectives for research on this subject.

## 2.    Materials and Experimental Methods

### 2.1    Samples

In this study, we used five coal fly ash (CFA) samples that were collected from electrostatic precipitators (EPs) of five different power plants - four in the United States of America and one in the United Kingdom. The four CFA samples from the USA were supplied by the Fly Ash Direct Ltd©, USA. The CFA samples were sourced from the following power plants: Clifty Creek Power Plant in Madison, Indiana (hereafter labelled as, CFA_Cy), Miami Fort Generating Station in Miami Township, Ohio (hereafter labelled as, CFA_Mi), Joppa Generating Station in Joppa, Illinois (hereafter labelled as, CFA_Ja), and J. Robert Welsh Power Plant in Titus County, Texas (hereafter labelled as, CFA_Wh). This is the same set of samples also studied and reported in Garimella (2016). Garimella (2016) grouped CFA_Ja and CFA_Wh fly ash samples as class C type, while CFA_Cy and CFA_Mi are class F which is broadly based on the calcium oxide (CaO) composition. A typical mass fraction of CaO in Class F CFA particles is ~ 1 – 12 wt.%, whereas Class C has higher CaO contents, sometimes up to 40 wt.% (Ahmaruzzaman, 2010). A new CFA standard classification system suggests that CFA samples can be sialic (S), calsialic (CS), ferrisialic (FS), and ferricalsialic (FCS) (Vassilev and Vassileva, 2007). However, no further information on chemical composition was provided by Garimella (2016) for a more quantitative classification of the USA CFA samples.

The UK coal fly ash sample was obtained from one of the major power plants in the UK and is referred to as CFA_UK throughout this report. The operator of the UK power plant prefers anonymity; hence, no specific name is mentioned here. The CFA particles collected from EPs are the same particles that could have been directly released into the atmosphere in situations where EPs malfunction or are inefficient. Also, the CFA particles which are emitted indirectly into the atmosphere by road transportation, application in agricultural fields, industrial sites, road construction, and other sources are the same CFA particles as collected from the EPs (Buhre et al., 2005). First, all raw CFA samples were sieved with a Fritsch Sieve set-up (Analysette 3, 03.7020/06209, Germany) to obtain 0 – 20 μm diameter size fractions, which were later used for the experiments.

### 2.2    AIDA chamber

All investigations were carried out in the Aerosol Interactions and Dynamics in the Atmosphere (AIDA) aerosol and cloud

simulation chamber. This is an 84 m$^3$ sized aluminium vessel sitting in a temperature-controlled housing, where the pressure, temperature, and the relative humidity are well-controlled depending on the experimental requirements. In addition, a suite of instruments is connected to the chamber for direct in-situ measurements or extractive measurements after sampling air from the chamber. A detailed description of the AIDA chamber and its instrumentation has been previously reported in various works (including but not limited to Möhler et al., 2003; Steinke et al., 2011; Wagner et al., 2009). Here, a brief overview of the devices which were employed in our study is highlighted.

A combination of an aerodynamic particle sizer (APS, TSI GmbH, USA), and a scanning mobility particle sizer (SMPS, TSI GmbH, USA) was used to measure the size distribution of the CFA aerosol particles in the AIDA chamber. The SMPS instrument measures in the size range of (13.3 – 835.4 nm), while the APS has a larger detection size range (0.5 – 20 µm). Both instruments were operated at the same time to obtain the fullsize distribution spectrum of the particles. A condensation particle counter (CPC3010, TSI, USA) was used to measure the number of aerosol particles in the chamber per volume. We also deployed two optical particle counters (OPCs, WELAS 2000, PALAS GmbH, Germany), which were connected to the base of the chamber to sample and count aerosol particles, cloud droplets, and ice crystals and also measure their respective optical sizes. Each of the OPCs had a different detection range (0.7 – 46 µm and 5 – 240 µm). The data obtained from the WELAS systems was later used to calculate the ice particle number concentration in the chamber during expansion cooling experiments with an uncertainty of ± 20 %. The water vapour concentration in AIDA at every stage of the experiment was measured with tunable diode laser (TDL) spectrometers, from which the relative humidities with respect to water (RH$_w$) and ice (RH$_{ice}$) were calculated with  ± 5 % uncertainty (Fahey et al., 2014). The spatial and temporal homogeneity of the temperatures in the AIDA chamber is better than ± 0.3 K. In this report, the mean gas temperatures will be given throughout the manuscript.

### 2.3 Aerosol generation and injection into AIDA

CFA aerosol particles were injected into the AIDA chamber with a rotating brush generator (RBG, RBG1000, PALAS GmbH, Germany) connected to the chamber with cleaned Teflon and stainless-steel tubing. We coupled the RBG to two cyclones placed in series to eliminate particles larger than 3 µm diameter. Cyclone 2 ($D_{50}$ cut-off = 3.7 µm) was placed before cyclone 3 ($D_{50}$ cut-off = 2.3 µm) in the set-up. The overall aim was to obtain smaller sized particles (< ~ 2.5 µm), which are more atmospherically relevant, especially for long-range transportation in the atmosphere (Prospero, 1999).

### 2.4 Morphology of CFA – sampling and imaging

Samples of CFA particles were collected on a nuclepore filter (25 mm diameter, 0.02 µm pore size, Whatman®, USA) from the AIDA chamber. The sampling was carried out with a mass flow controller (MFC, Tylan®, UK) running at 2 L min$^{-1}$ for 30 min. The loaded filters were sputter-coated with 1 nm platinum to improve the conductivity, and the images were taken with an environmental scanning electron microscope (ESEM, FEI Quanta 650 FEG). Coating of the filters did not affect the morphology of our samples because the coating thickness was 1 nm and thus below the SEM resolution. A different model of ESEM (ThermoFisher Scientific Quattro S) was used for the USA CFA samples. With this new ESEM model, we were able to obtain images of the CFA particles under grazing viewing angles similar to 3-D images (see Fig. 1).

### 2.5 Surface area and pore size measurement

We adopted the Brunauer–Emmett–Teller (BET) method (Brunauer et al., 1938) to measure and analyse the specific surface areas (SSAs) of the 5 CFA samples. The CFA samples were degassed at ~ 368 K for 24 hours before measuring the molecular

adsorption on the particles (a 5-point BET model was used). During the degassing process < 8.5 % mass loss was recorded for all the CFA samples. Specifically, we used argon gas (87.3 K) as the adsorbent instead of the standard nitrogen gas, hence, we tagged it $BET_{Ar}$. Argon gas provides better adsorption for the estimation of SSA because of its monatomicity and non-localization of the adsorbent during adsorption (Rouquerol et al., 2014; Thommes et al., 2015). This measurement was performed with an Autosorb 1-MP Instrument (Quantachrome, Germany). The pore size volumes were calculated with models based on DFT/Monte Carlo methods assuming a mixture of spherical and cylindrical pores on an oxygen-based substrate (Landers et al., 2013; Thommes et al., 2006). The SSA ($m^2$/g) from the $BET_{Ar}$ measurements and the calculated pore volumes for all the CFA samples are presented in Table 1. All adsorption and desorption isotherms of the different CFA samples are available in the Supplementary Information (Fig. S1).

## 2.6    *Temperature-cycling and ice nucleation experiments in the AIDA chamber*

CFA aerosol particles were first injected into the chamber filled with synthetic air at a particular temperature - hereafter, referred to as start temperature ($T_{start}$) - and mixed with the aid of a big fan installed at the lower level of the chamber. After the injection into the AIDA chamber, the CFA particles were probed in two different ways. In the first type of experiments, the particles' inherent ice nucleation ability was tested at temperatures between 261 K and 228 K by means of an expansion cooling cycle. For this purpose, the pressure of the chamber was reduced with the aid of a vacuum pump (Möhler et al., 2005). Cooling and the concomitant increase of the relative humidity triggered the droplet activation of the particles, and a subset of the CFA particles nucleated ice via immersion freezing during continued pumping. Generally, pumping was stopped when the maximum $RH_{ice}$ was reached.

In the second type of experiments, a temperature-cycling and freezing (TCF) protocol was adopted. Previously, this method had been used for similar experiments with other aerosol types in the AIDA chamber (Wagner et al., 2012, 2016). In the TCF procedure, the CFA particles were injected into the AIDA chamber (~ 1300 - 1600 particles per $cm^3$) at ~ 253 K and cooled to ~ 228 K. During the cooling process, a rate of 5 K $h^{-1}$ was achieved. The CFA aerosol particles were then warmed to 253 K (or the desired $T_{start}$) at 2.5 K $h^{-1}$, as described by Wagner et al. (2016). During the entire cooling and warming process (Fig. 2), the relative humidity prevalent in the AIDA chamber was slightly below ice saturation, as controlled by an ice layer on the inner chamber walls. The slight sub-saturation of the chamber air with respect to ice may be attributed to some internal heat sources which increased the gas temperature by a few tenths of a Kelvin compared to the wall temperature (Wagner et al., 2016). After warming, the particles' ice nucleation ability was probed in an expansion cooling run as described above. Details of the various experiments that we conducted and outcomes are shown in Table 2.

In this study, we used the ice-active fractions to compare the data from the various experiments performed. The fraction of ice frozen (i.e., the ice-activated fraction, $f_{ice}$) was calculated as the number of ice particles detected divided by the total number of seed aerosol particles present in the chamber (Vali, 1971). The uncertainty associated with our $f_{ice}$ calculations is ~ ± 20 % (Möhler et al., 2006). The $f_{ice}$ data in each experiment are plotted in Figs. 3, 4, 6, and S2 - 5. For each experiment, the maximum ice-activated fraction ($f_{ice,max}$) are presented in Figs. 5, 7 and 8.

## 3.    Results and Discussions

The AIDA measurement data showing the inherent ice-nucleating ability of the CFA particles are shown in Fig. 3 (CFA_UK) and in the first columns (A) of Fig. 6 (CFA_Cy), Fig. S2 (CFA_UK repeat), Fig. S3 (CFA_Mi), Fig. S4 (CFA_Ja) and Fig.

S5 (CFA_Wh). Each column of Figures 3, 6, S2, S3, S4 and S5 has 3 panels. The top panels represent the pressure and the temperature profiles before, during, and shortly after the expansion. For each start temperature ($T_{start}$), the expansion started at ~ 1000 hPa down to where the maximum $RH$ (see middle panels) was obtained. The point where the pressure starts rising indicates when the expansion was stopped. The middle panels show the relative humidity data with respect to both water and ice denoted as $RH_w$ and $RH_{ice}$, respectively. The bottom panels show the optical diameters and counts of the aerosol particles, cloud droplets, and ice crystals inferred from the OPCs. The CFA aerosol particles are shown by the dots at the beginning of the plot (α, see Fig. 3A), just before the pumping starts, with diameters < 10 µm. Note that the size scale of the OPCs was calibrated for spherical particles with a refractive index of 1.33. The slightly aspherical shape and much larger refractive index of the CFA particles (Jewell and Rathbone, 2009) lead to a significant overestimation of their true diameters on this size scale. Therefore, some CFA particles are detected at apparent diameters above the minimum cut-off size of our cyclones ($D_{50}$ = 2.3 µm). The particles activated into droplets are indicated by the denser cloud of data points with much bigger sizes, which shows that the CFA particles took up water, got immersed and increased in size (denoted by β, Fig. 3A). Finally, in the case where CFA particles had been activated into cloud droplets, the nucleated ice particles in the later course of the expansion run are indicated by the data points with sizes above the dense cloud of supercooled water droplets (see an illustration in Fig. 3C, denoted by *γ*). In the cirrus regime or after temperature-cycling, the CFA particles can also directly form ice without going through the droplet activation phase (Fig. 3D and Fig. 4A and B). We used a size threshold, empirically set for each experiment, to separate the ice particles from both the CFA seed aerosol particles and the activated cloud droplets, similar to the approach reported in previous AIDA experiments (Steinke et al., 2016; Suski et al., 2018; Ullrich et al., 2016).

The results from the ice nucleation experiments are presented as follows. We start with the description of the inherent ice nucleation behaviour of the CFA samples (Section 3.1); followed by the enhancement of their ice nucleation activities due to pre-activation by the PCF mechanism (Section 3.2), and finally, we discuss potential implications of this mechanism for cloud formation by CFA INPs, especially those that have undergone similar temperature-cycling in the atmosphere (Section 3.4).

### 3.1        *Ice-nucleating activity of CFA particles*

We start our discussion with the CFA_UK particles. When probed in an expansion cooling run at $T_{start}$ = 261 K, the ice-active fraction was generally below the detection limit of 0.02 % (Fig. 3A). However, at $T_{start}$ = 253 K, about 0.19 % of the particles had nucleated ice via the immersion freezing mode in the course of the expansion cooling run until the minimum temperature of 244 K was reached (Fig. 3B). The ice-active fraction encountered during the expansion cooling run at $T_{start}$ = 245 K was by a factor of 10 higher compared to the run started at 253 K. At $t$ ~ 300 s, the homogeneous freezing mode kicked in (see the illustration in Fig. 3C). In our analyses, ice particles detected just before, during, and after such events were omitted from the ice particle counts. In summary, the CFA_UK particles were thus observed to be active in the immersion freezing mode at temperatures below 253 K, however, the ice-activated fractions were rather low and exceeded 1 % only at temperatures very close to the homogeneous freezing threshold of pure water. The homogeneous freezing threshold temperature observed in our experiments (237.0 K) agreed with previous reports (Benz et al., 2005; Schmitt, 2014). In contrast, for the experiment at $T_{start}$ = 228 K (Fig. 3D), more than 64 % (T = 220 K) of the aerosol particles nucleated ice directly from the CFA_UK particles at very low supersaturations. This means that within a change of only 9 K from the homogeneous freezing temperature of pure water (237 K) to the expansion run started at 228 K, the ice-active fraction of the CFA_UK particles increased by almost 2 orders of magnitude. A similar increase in the heterogeneous ice nucleation ability has been previously

observed for zeolite and illite particles (Wagner et al., 2016), and temperature-cycling experiment with these particles have substantiated that the PCF mechanism is the most likely explanation for the sudden increase of the particles' ice nucleation behaviour below the homogeneous freezing temperature of supercooled water. Following the experiment at $T_{start}$ = 228 K, we hypothesized that PCF may also be the dominant nucleation pathway for the CFA particles. To verify this hypothesis, we adopted the TCF approach as discussed in Section 3.2.

Other CFA samples studied here – CFA_Cy, CFA_Mi, CFA_Ja, and CFA_Wh – were also tested for their inherent ice-nucleating properties in the immersion freezing mode at $T_{start}$ = 251 K, 250 K, 251 K, and 248 K, respectively (Figs. 6, S3, S4, and S5). The onset temperatures ($T_{onset}$) are reported in Table 2. Here, we defined our $T_{onset}$ in each experiment as the temperature where the $f_{ice}$ is > 0.1 %. In order to compare the inherent ice nucleation behaviour of the five CFA samples investigated, we have tabulated the $f_{ice,max}$ (%) for experiments with a similar starting temperature of about 250 K (Table 2, experiment numbers 3 and 5 - 8). The results reveal a significant spread in the ice-activated fractions, with CFA_Wh (~ 26 %) > CFA_Ja (~ 17 %) >> CFA_Cy (~ 1.5 %) = CFA_Mi (~1.5 %) > CFA_UK (~ 0.17 %). This huge variation in the particles' inherent ice nucleation activity is probably related to differences in morphology, elemental composition, and/or surface functionalization. The observed differences in their inherent ice-nucleating abilities may also be due to variabilities in their chemical and mineralogical compositions. Garimella (2016) reported that the four CFA samples from the USA belonged to different classes of fly ash and these groupings are based on the chemical compositions (Garimella, 2016). Further analyses on the distribution of the ice nucleation active sites densities of these CFA particles is outside the scope of the current report and will be presented in a separate communication.

Coal fly ash particles from other sources have been reported to nucleate ice inherently at much higher temperatures. Previously studied CFA particles were suspended in deionized water before ice nucleation properties were investigated on a cold stage set-up. For example, a particular sample from one of the UK power plants was reported to nucleate ice in the immersion freezing mode already starting at 257 K (Umo et al., 2015). This sample also showed a steep curve in the $f_{ice}$, indicating the presence of unique ice active sites which may be similar to what we observed in CFA_Ja and CFA_Wh. Grawe et al. (2018) reported even higher freezing temperatures (at 265 K) for CFA particles obtained from a power plant in Germany. This again was attributed to the unique composition of CFA samples. Both studies, however, were performed with drop freezing assay techniques and with much larger particles than reported here. Moreover, in a drop freezing assay method, a droplet can contain many particles, whereas each cloud droplet activated in the AIDA chamber only contains a single particle. Hence, the probability of observing freezing events in drop freezing assay at much higher temperatures was higher than in the AIDA experiments where smaller particle sizes were explored. A combination of both techniques in future studies could ultimately yield a parameterization of the heterogeneous ice nucleation activity of the CFA particles over the entire range of temperatures in the mixed-phase cloud regime. In another study, particles in a plume from a coal-fired power plant were not considered ice active at temperatures above 253 K (Schnell et al., 1976). However, when similar experiments were conducted at a higher supersaturation, the particles' ice nucleation ability increased, indicating that CFA particles could act as good INPs even at temperatures as high as 263 K (Parungo et al., 1978). However, in these experiments, not many details on the exact experimental conditions are available for a direct comparison with our experiments. Also, the particles in the plume were not well characterized, hence, it may have contained any other ambient aerosol particles.

Generally, for investigations in a measurement set-up that requires a dry generation method, much lower temperatures are reported as inherent ice-nucleating temperatures of CFA as INPs. A study of CFA samples from Germany in a laminar flow

tube in Leipzig called Leipzig Aerosol Cloud Interaction Simulator (LACIS) showed ice nucleation from ~ 247 K - 236 K (Grawe et al., 2016, 2018). Although these freezing temperature range is comparable to what we observed with our samples (Table 2), it should be noted here that the particle size of the CFA samples used in Grawe et al. (2016 and 2018) is different from the size range used in our study. First, the average median particle diameter of our CFA samples is 0.58 µm, whereas Grawe et al. (2016) reported an average diameter of 0.3 µm. This can also have an impact on the behaviour of INPs (Garimella, 2016). Second, we should state here that these particles are from different sources, hence, they might have different mineral (or chemical) compositions as well as surface properties. Aerosol compositions and surface properties have been clearly established to influence the ice nucleation behaviour of INPs (Fitzner et al., 2015; Harrison et al., 2016; Isono and Ikebe, 1960; Lupi et al., 2014; Mason and Maybank, 1958). Third, the different measurement techniques applied in each study can also introduce some differences (Grawe et al., 2018). In comparison with other aerosol types, the ice nucleation activities of CFA particles in the immersion freezing mode are considerably higher than e.g. soot particles (Mahrt et al., 2018), but less active compared to some biological materials (Suski et al., 2018). Generally, the ice-nucleating abilities of CFA samples are similar to the ice-nucleating potential of some mineral components of desert or agricultural soil dusts (Grawe et al., 2018; Umo et al., 2015).

### 3.2    Enhancement of the ice-nucleating properties of CFA particles by temperature-cycling

In the previous section, we reported the inherent ice nucleation activity of CFA particles. Here, we show the results for CFA particles that were temporarily exposed to a lower temperature (228 K) before the expansion cooling experiments were conducted. Freezing data after the temperature-cycling and freezing (TCF) procedure are presented in Figs. 4A – C, 6B and panels B and C of S2 – 5.

After the TCF process, experiments were conducted with the processed CFA_UK particles following the schematic in Fig. 2. Specifically, we conducted two independent series of experiments, each with a fresh load of aerosol particles, following the sequences $T_{start}$ ~ 250 K → 254 K → 264 K (series I, experiment #9, #10 and #11, data shown in Fig. 4) and $T_{start}$ ~ 251 K → 254 K → 263 K (series II, experiment #12, #13, and #14, data shown in Fig. S2). As the results from both series are very similar, we focus our discussion on the experiments conducted during series I. At $T_{start}$ = 250 K, we clearly observed an increase in the $f_{ice,max}$ of the CFA_UK particles (up to 11 % at $T$~246 K) compared to the unprocessed CFA_UK particles that only showed $f_{ice,max}$ of 1.6 % at $T$~238 K, which was even at a lower start temperature ($T_{start}$ = 245 K). The processed CFA_UK particles nucleated ice at water-subsaturated conditions with a nucleation threshold in terms of $RH_{ice}$ of only about 101 %. In contrast, the unprocessed CFA_UK particles nucleated ice in the immersion freezing mode after exceeding water saturation during the expansion run (corresponding to $RH_{ice}$ ~ 130 %). This means that there was a change in the ice nucleation mode in comparison with the unprocessed CFA_UK particles in the same $T_{start}$ range. For the processed CFA_UK particles, there was no droplet activation before the emergence of ice, i.e., ice formation cannot be ascribed to 'classical' immersion freezing (Fig. 3A - C). Rather, the ice particles observed were formed directly on the pre-activated CFA_UK particles. Following the history of these particles, we suggest that the ice particles may have been formed by the depositional growth on the ice germs formed in the pores of the particles during temperature-cycling.

After the first expansion at $T_{start}$ = 250 K, we warmed the chamber to 254 K and performed another expansion cooling run. The ice-activated fraction decreased by a factor of 2 compared to the run at $T_{start}$ = 250 K ($f_{ice,max}$ ~ 3 %), but was still significantly higher than what was observed for the unprocessed CFA_UK particles at a similar temperature ($f_{ice,max}$ = 0.19

% at $T{\sim}245$ K). Ice formation by the processed CFA_UK particles again occurred by the depositional growth mode at low ice supersaturation ($RH_{ice,max}{\sim}$ 109 %), whereas the much smaller ice-activated fraction of the unprocessed particles was due to immersion freezing at water-saturated conditions corresponding to $RH_{ice}$ = 124 % at $T_{start}$ = 253 K (Fig. 3).

Afterwards, the same processed CFA_UK aerosol particles were warmed to $T_{start}$ = 264 K for another expansion cooling run (Fig. 4C). At this start temperature, the ice nucleation ability of the unprocessed CFA_UK particles was below our detection limit of 0.02 % for $f_{ice}$. For the processed CFA_UK particles, however, a maximum ice-activated fraction of 1.3 % was observed at $T{\sim}251$ K. In contrast to the runs conducted at $T_{start}$ = 250 K and 254 K, the ice cloud was not formed at low supersaturation values with respect to ice, but appeared just at the instant of droplet activation ($RH_{ice}{\sim}107$ %, Fig. 4C). Given the absence of any ice formation for the unprocessed particles, it is highly probable that the nucleation mode of the processed CFA_UK particles, although being similar to a classical immersion freezing mode, is in fact related to ice growth from an existing ice germ formed during temperature-cycling. This implies that at least 1.3 % of the processed CFA_UK particles still contained ice-filled pores even after warming to 264 K. Such ice formation modes have already been observed for other particle types in similar scenarios (e.g. Mahrt et al., 2018; Wagner et al., 2016), and have been ascribed to the condensational growth of the ice germs formed in the pores or crevices of these particles. Figure 5 shows the summary of the ice nucleation enhancement of CFA_UK particles described above with $f_{ice,max}$ and their corresponding temperatures as well as the respective start temperatures of each experiment. It is clear that for the processed CFA_UK particles, the $f_{ice,max}$ values are significantly higher than those for the unprocessed particles at a similar $T_{start}$.

In contrast to the CFA_UK particles, the CFA particles from the USA power plants showed less modification of their ice nucleation ability after the temperature-cycling process. For none of these particle types, a distinct depositional ice growth mode as shown in Figs. 4A and B for the CFA_UK particles was observed. However, some particle types revealed an improved ice nucleation ability due to the condensational ice growth mode, as exemplified in Fig. 6 for the CFA_Cy particles. Whereas the ice-activated fraction of the unprocessed CFA_Cy particles remained below 0.5 % for temperatures above 244 K (Fig. 6A), the particles subjected to temperature-cycling showed ice formation with $f_{ice}$ > 0.5 % already at 249 K (Fig. 6B). Similar to the experiment with CFA_UK at $T_{start}$=264 K (Fig. 4C), this ice mode was instantaneously formed upon droplet activation, i.e., is most likely related to a condensational ice growth mode. The CFA_Cy particles also showed a tiny depositional growth mode indicated by a few ice particles detected before the droplet activation (Fig. 6B). To better illustrate the partly small differences in the ice nucleation ability of the CFA particles from the USA with and without temperature-cycling, we summarize in Fig. 7 the ice-activated fractions as a function of temperature for both the expansion cooling runs with processed and unprocessed particles. For the corresponding data of the CFA_Cy particles as discussed above, there is a clear shift of the ice nucleation spectrum towards higher temperatures after temperature-cycling. The difference is much less pronounced for other CFA particles from the USA.

Pre-activated CFA_Ja particles did not show any significant improvement of their ice nucleation ability after the temperature-cycling experiment for expansion cooling experiments started at around 250 K (Fig. 7). Obviously, pre-activation cannot compete with the already very high inherent heterogeneous ice nucleation ability of the CFA_Ja particles at this temperature, meaning that there is no further detectable increase in the ice-activated fraction after the TCF cycle. However, the pre-activation phenomenon becomes visible when further warming the pre-activated CFA_Ja particles to a higher starting temperature (256 K, Fig. S4, panel C). Here, the processed CFA_Ja particles showed a small nucleation mode with $f_{ice}$ ~ 1 % at 252 K just when exceeding water saturation during the expansion run. Given that the threshold temperature for exceeding

an ice-activated fraction of 1 % for the unprocessed CFA_Ja particles was as low as 246 K, the observed ice nucleation mode for the processed CFA_Ja particles at 252 K can most likely be ascribed to the condensational growth of pre-existing ice, generated in the pores of the particles during the TCF cycle.

Similar to the CFA_Ja particles, also the CFA_Wh particles did not significantly change their ice nucleation ability after the TCF cycle when probing them at starting temperatures of 248 K - 249 K (Fig. 7), i.e., in a temperature range where the particles' inherent heterogeneous ice nucleation ability is already very high. The smaller nucleation mode with $f_{ice}$ ~ 2 % that was observed after further warming the processed CFA_Wh particles to 256 K (Fig. S5, panel C), however, is likely again due to the condensational ice growth mode. The CFA_Mi particles showed the smallest variation with respect to their ice nucleation ability after the TCF cycle. In addition to the comparable ice nucleation behaviour before and after temperature cycling at a starting temperature around 250 K (Fig. 7; Fig. S3 panels A and B), the processed CFA_Mi particles also revealed only a tiny condensational ice growth mode at a higher starting temperature of 255 K with $f_{ice,max}$ ~ 0.1 % (Fig. S3, panel C).

The degree of ice nucleation enhancement by CFA particles differs from sample to sample. The enhancement capability of the CFA samples studied here are in this order: CFA_UK>>>> CFA_Cy>CFA_Wh>CFA_Ja>CFA_Mi. The ranking is based on the start temperature, $f_{ice,max}$, and the relative humidity as summarized in Figs. 5 and 7. Morphology, chemical composition, surface area, and pore volume are important parameters influencing the efficiency of the PCF mechanism. In the following, we discuss whether differences in these properties can account for the different behaviour of the CFA particles after temperature-cycling.

The morphology of the five samples is shown in Fig. 1 for selected typical particles. The SEM images showed that the CFA particles have some degree of roughness, coatings, layers, and mesh-like structures on their surface. Although the overall particle habit is spherical, as many electron micrographs of CFA have shown (Blissett and Rowson, 2012; Fisher et al., 1978), they have no smooth surface. Of the 5 CFA samples, CFA_UK had the highest degree of deformity on the surface as indicated in Fig. 1a - f. We attempted to focus into the surface (up to ~ 50 - 100 nm resolution) to identify the potential pores and crevices but it was difficult to have a clear view of the pores (Fig. 1c and f). Classical nucleation theory (CNT) and empirical calculations have shown that pore diameters of about $5 - 8$ nm (mesopores) contribute to a particle's pre-activation ability at ice sub-saturated conditions (Marcolli, 2014; Wagner et al., 2016). A more recent study using CNT and molecular dynamics has shown that it is not enough to have pore diameters of the above size but that a network of closely spaced pores is necessary to overcome the free energy required for a macroscopic ice-crystal growth from narrow cylindrical pores (David et al., 2019).

In previous studies, it has been shown that the specific surface area and pore volume of fly ash particles generated from pulverized coal combustion are very likely dependent on the particle size (Schure et al., 1985; Seames, 2003). To better understand the nature of the CFA surfaces, we measured the specific surface area (SSA) of the sieved bulk samples ($0 - 20$ μm) using the BET method but with argon gas rather than nitrogen (Gregg et al., 1967; Thommes et al., 2015). We obtained 5-point $BET_{Ar}$ surface areas as tabulated in Table 1. The $BET_{Ar}$ of CFA_UK had the highest SSA of 14 $m^2\,g^{-1}$, which was a factor of 3 higher than those of the other CFA particles: CFA_Cy (5 $m^2\,g^{-1}$), CFA_Mi (4 $m^2\,g^{-1}$), CFA_Ja (4 $m^2\,g^{-1}$), and CFA_Wh (3 $m^2\,g^{-1}$). The high SSA of CFA_UK is indicative of the presence of crevices in form of pores or grooves and therefore could account for the ice nucleation enhancement exhibited by the pre-activated CFA_UK particles compared to the other CFA particle types in this study. Note that this does not necessarily mean that all particles with high SSA such as soot particles will show pre-activation and ice nucleation enhancement. For example, pre-activation was not observed for

water-processed soot particles (Wagner et al., 2016), however, other soot types have been suspected to show considerable ice activity via the PCF mechanism (Mahrt et al., 2018; Wagner et al., 2016).

We also report the pore volume (PV) of the investigated particles (Table 1). The PV was calculated with a DFT/Monte Carlo model assuming that the pore diameters are not greater than 100 nm. In our results, CFA_UK had the highest PV (0.05 cm$^3$ g$^{-1}$), about 4 to 5 times higher compared to the other CFA samples. Amongst these other CFA samples, there was no clear correlation between PV and corresponding ice nucleation enhancement. For example, CFA_Ja and CFA_Wh had very similar PV (0.009 and 0.010 cm$^3$ g$^{-1}$, respectively) but CFA_Wh showed a higher ice susceptibility to pre-activation than the former. Another example is CFA_Cy (0.012 cm$^3$ g$^{-1}$), which has a PV similar to the CFA_Mi sample (0.013 cm$^3$ g$^{-1}$), but only the processed CFA_Cy particles showed a clear pre-activation ability due to the PCF mechanism. Specific surface areas correlate with the PV (Sigmund et al., 2017), however, it is difficult to ascertain the geometries of the pores or crevices contributing to the surface area. CFA particles are very unique particles in that some of them can be cenospheres (hollow particles with a tiny opening). They can also be plerospheres, i.e. a case whereby smaller particles fill the larger cenospheres (Alegbe et al., 2018; Fisher et al., 1978; Goodarzi, 2006; Goodarzi and Sanei, 2009). The cenospheres and plerospheres present in the CFA samples could increase the pore volume of these particles, hence, leading to a higher uncertainty in estimating the pore size. Currently, it is highly difficult to estimate the pore sizes of the CFA particles based on the PV only except in the case of a well-defined pore model and morphology. We suggest that knowing the possible geometries of defects on the surface of INPs may help to predict their pre-activation behaviour.

### *3.3    Ice nucleation enhancement by CFA particles versus other particle types*

In a previous study, Wagner et al. (2016) investigated the pre-activation behaviour of INPs by the PCF mechanism in the AIDA cloud chamber with a similar measurement routine as described in Section 2.6. In this study, a wide range of INPs was tested including illite NX, diatomaceous earth, zeolites, dust samples from Canary Island, Sahara and Israel, Graphite Spark Generator soot (GSG soot), and volcanic ash (Wagner et al., 2016). It was reported that illite NX, diatomaceous earth, and mesoporous zeolite CBV 400 showed a significant ice nucleation enhancement in the depositional ice growth mode, with ice-active fractions of 5.9 %, 3.8 %, and 3.7 % at a starting temperature of ~ 250 K (Fig. 8). At higher starting temperatures, the ice-activated fractions in the condensational ice growth mode were typically around 1 %. Another group of INPs such as CBV 100 (untreated microporous zeolites), Canary Island dust, and GSG soot showed much smaller depositional ice growth modes with ice-activated fractions below 1 %. Finally, volcanic ash, water-processed GSG soot, as well as Saharan and Israeli dust particles did not show any enhancement after the pre-activation process, neither in the depositional nor the condensational ice growth mode.

In this context, the ice nucleation enhancement observed for the CFA_UK particles at a starting temperature of 250 K in the depositional growth mode with $f_{ice,max}$ ~ 11 % (Fig. 4A) is by far the highest value for any particle type investigated so far (Fig 8). In contrast, the pre-activation efficiency of the CFA particles from the US power plants is comparable in magnitude to the above-mentioned group of CBV100, Canary Island dust, and GSG soot particles with much lower ice-activated fractions. The mean diameters of the particles investigated by Wagner et al. (2016) ranged from 0.21 µm to 0.43 µm, and were thus smaller than the mean diameters of our CFA particles except for CFA_Mi (0.42 µm). Different pore sizes, morphology, and chemical composition of these INPs may control their susceptibility to the PCF pre-activation mechanism. More studies are required to investigate the role that each of these parameters play.

### 3.4    *Potential implication of the pre-activation of CFA particles in clouds*

Ice nucleation by CFA particles pre-activated via the PCF mechanism could be important for different cloud types. When CFA particles are lofted into the atmosphere, these particles can act as INPs or CCN as well as sinks for other atmospheric species (Dlugi and Güsten, 1983; Havlíček et al., 1993; Herndon, 2016; Korfmacher et al., 1980; Muduli et al., 2014). During their residence time in the atmosphere, the CFA particles can be transported through different relative humidity and temperature regimes. If the particles were temporarily exposed to temperatures below 237 K at high ambient relative humidity, their ice nucleation ability might improve by the formation of ice-filled pores. There is a high potential that the pre-activated CFA particles can be re-circulated as INPs via a sedimentation process into the lower atmosphere to contribute to ice formation in mixed-phase clouds as illustrated in Fig. 9. Some of the atmospheric processes that could aid the re-circulation of the pre-activated INPs are radiative cooling, deep convective flows, sedimentation and feeder-seeder mechanisms (Carruthers and Choularton, 1983; Highwood and Hoskins, 1998; Hong et al., 2004; Salathé and Hartmann, 1997). By convective atmospheric dynamics, these pre-activated particles could then be released to lower altitudes and trigger ice formation at higher temperatures than expected from their inherent ice nucleation ability. In addition, some CFA particles that initiated cloud glaciation can also be released via cloud evaporation or the sublimation of the ice particles releasing the CFA ice residues back into the atmosphere. These pre-activated CFA INPs can then re-initiate cloud formation at higher temperatures than inherently expected for the same CFA INPs. This process is not peculiar to CFA particles, but also relevant for other natural and anthropogenic INPs with unique properties such as illite NX, zeolite, and GSG soot that exhibit PCF mechanism and can have a wider atmospheric implication in cloud formation. Despite the dearth of information on the number concentration of CFA particles in the atmosphere at higher altitudes, there are pieces of evidence that CFA particles are found in ice residues of cirrus and mixed-phase clouds (DeMott et al., 2003; Liu et al., 2018).

We suggest that future modelling work should focus on the impact that pre-activated INPs or INPs with ice-filled pores can have on cloud formation processes. Some observations show that more ice particles are observed at higher temperatures than the amount expected by the available INPs (Hobbs and Rangno, 1985). Aside from secondary ice multiplication processes (Hallett and Mossop, 1974; Phillips et al., 2018), it could be possible that pre-activated INPs also contribute to the higher concentration of ice crystals than are observed in some cases. There are other open questions in these areas such as understanding the timescale and frequency (often or episodic) with which this phenomenon occurs in clouds, the impact of this process in mixed-phase and cirrus cloud formation, and the occurrence at regional and global levels. The PCF mechanism could be potentially important for cirrus cloud systems because CFA particles entrained into the upper troposphere at lower temperatures could already have their pores filled with ice. For instance, our experiment with CFA_UK particles at T~220 K showed over 60 % ice activation (Fig. 3D).

### 4.    Conclusions

Coal fly ash (CFA) aerosol particles inherently nucleate ice in the immersion freezing mode as shown from this investigation and in previous studies. Also, an exposure of these particles to favourable atmospheric conditions such as cold temperatures (~ 228 K) at ice sub-saturated conditions can induce the formation of ice germs in the pores of the CFA particles by the pore condensation and freezing (PCF) mechanism. The ice-filled pores in the CFA aerosol particles can then account for their improved ice nucleation efficiencies at higher temperatures, where inherently, CFA will show very poor or no ice nucleation potential at all. This behaviour could be attributed to the degree of surface defects, and porosity of such CFA particles, which differ from sample to sample. In this study, we have clearly shown that CFA_UK particles are capable of enhancing their ice

formation potential up to about 264 K by a factor of 2 for the condensational growth and even higher when they form ice by the depositional growth mode of the pre-existing ice germs.

A more in-depth study in understanding the temperatures and relative humidity ranges in which the ice in the pores can be preserved is important in quantifying the particles' overall ice-nucleating efficiencies. Preservation of ice in the particles'

pores will depend on their temperature and relative humidity histories during atmospheric transport. This will clearly define the viability of INPs to form ice via the PCF mechanism. We suggest that further studies should be focused on investigating the effect of different pore geometries on the ice-nucleating abilities via the PCF mechanism. This can have a wider application in the modelling of cloud formation processes, and would help in constraining the uncertainties associated in the Earth system interactions, e.g. aerosol-cloud interactions. We also suggest that in order to overcome the bias associated with

pore models in estimating pore sizes and diameters for natural aerosol particles, a parameter based on the pore volume, pore size/diameter, and specific surface area should be adopted.

In summary, we identify the following open questions: (1) How do the pore geometries influence the PCF mechanism? This could be useful in predicting the behaviour of INPs in different tropospheric conditions. (2) At what temperature and relative humidity conditions will the pre-activated ice sublime/melt or become ineffective in triggering ice formation? (3) On which

timescale does a potential INP need to be exposed to lower temperatures for pre-activation to occur? (4) What are the typical temperature and relative humidity histories that aerosol particles experience during atmospheric transport? and (5) Aside from the atmospheric implications, how well do we understand this process for other applications, especially in cryopreservation, bioengineering, and agriculture.

*Data availability:* All data shown in this report will be available via KITopen or contact <nsikanabasi.umo@partner.kit.edu>.

*Author contributions:* NSU and RW designed and conducted the experiments with contributions from OM, RU, HS. NSU, RW, RU, TL, AK, DC, and OM analysed the data and discussed the ice nucleation results. PGW characterised the BET and pore volume of the samples and led the discussions of the results. AK and NSU took the SEM images and discussed the morphology of the particles. NSU prepared the manuscripts with contributions from all the co-authors (RW, RU, AK, HS,

PGW, DC, TL, and OM). OM hosted and provided a complementary funding for the project.

*Competing interests:* The authors declare that they have no conflict of interest.

*Acknowledgements:* N. S. Umo acknowledges Alexander von Humboldt Foundation, Germany (1188375) for funding his research fellowship and generously thanks IMK-AAF, KIT for access to the AIDA Cloud/Aerosol Simulation Chamber and other instrumentation. The authors are thankful to the AIDA technical team at IMK-AAF, KIT for their assistance in operating

the AIDA chamber, specifically, George Scheurig, Steffen Vogt, Tomasz Chudy, Rainer Buschbacher, and Olga Dombrowski. The authors acknowledge Professor (Emeritus) Alan Williams of the University of Leeds for providing one of the CFA samples. The two anonymous reviewers of this work are duly commended for their comments and suggestions which were very useful to this work. Part of this work was funded by the Helmholtz Association of German Research Centres through its Atmosphere and Climate Programme.

*Any opinions, findings, and conclusions or recommendations expressed in this material are those of the author(s) and do not necessarily reflect the views of the Alexander von Humboldt Foundation.*

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

**Tables**

**Table 1:** Sources, specific surface areas, pore volume, and the median diameter of coal fly ash aerosol particles used in this study. Argon gas was used for the BET measurements; hence, it is labelled as $BET_{Ar}$. The median diameter was determined from the combined data of the APS and the SMPS instruments. The details of the samples and labels are given in Section 2.1. The countries that the samples originated from are the United Kingdom (UK) and the United States of America (USA).

| Sample labels | Country of origin | $BET_{Ar}$ Specific surface area [m²/g] | Specific Pore volume (~up to 100 nm pore size) [cm³/g] | Median diameter of the CFA particles [μm] |
|---|---|---|---|---|
| CFA_UK | UK | 14 | 0.053 | 0.47 |
| CFA_Cy | USA | 5 | 0.012 | 0.66 |
| CFA_Mi | USA | 4 | 0.013 | 0.42 |
| CFA_Ja | USA | 4 | 0.010 | 0.68 |
| CFA_Wh | USA | 3 | 0.009 | 0.66 |

**Table 2:** Information on the various ice nucleation experiments conducted during two distinct AIDA measurement campaigns named CAINIC01 and EXTRA18. Experiments (1 - 8) were conducted with unprocessed CFA particles while experiments (9 – 24) were performed with processed CFA particles (i.e., after the temperature-cycling process involving intermediate cooling to ~ 228 K, see Section 2.6). The freezing modes mentioned here are based on the classification presented in Vali et al. (2015).

| S/No. | Campaign/ Experiment name | CFA Samples | Start temperature ($T_{start}$) before the expansion (K) | Concentration of CFA particles in the AIDA chamber before the expansion (cm⁻³) | $T_{onset}$* (K) | $f_{ice,max}$ (%) | $RH_{ice}$ (%) at $f_{ice,max}$ | T (K) at $f_{ice,max}$ | Dominant ice nucleation mode observed (based on classical definitions in Vali et al. 2015) |
|---|---|---|---|---|---|---|---|---|---|
| *Experiments before the temperature-cycling process (Unprocessed particles)* | | | | | | | | | |
| 1 | CAINIC01_10 | CFA_UK | 228 | 225 | 227.0 | 64.11 | 104.9 | 219.6 | Deposition |
| 2 | CAINIC01_13 | CFA_UK | 261 | 189 | ND | ND | - | - | Immersion |
| 3 | CAINIC01_14 | CFA_UK | 253 | 218 | 245.5 | 0.19 | 123.8 | 244.7 | Immersion |
| 4 | EXTRA18_03 | CFA_UK | 245 | 218 | 241.4 | 1.61 | 136.8 | 237.7 | Immersion |
| 5 | EXTRA18_05 | CFA_Cy | 251 | 175 | 246.2 | 1.67 | 130.1 | 242.6 | Immersion |
| 6 | EXTRA18_06 | CFA_Ja | 251 | 219 | 246.2 | 16.51 | 117.6 | 243.7 | Immersion |
| 7 | EXTRA18_14 | CFA_Wh | 248 | 228 | 244.8 | 26.01 | 110.8 | 242.8 | Immersion |
| 8 | EXTRA18_15 | CFA_Mi | 250 | 195 | 245.1 | 1.52 | 130.3 | 242.6 | Immersion |
| *Experiments after the temperature-cycling process (Processed particles)* | | | | | | | | | |
| 9 | CAINIC01_18 | CFA_UK | 250 | 523 | 247.9 | 10.50 | 101.1 | 245.7 | Deposition |
| 10 | CAINIC01_19 | CFA_UK | 254 | 453 | 250.5 | 2.87 | 109.0 | 249.2 | Deposition |
| 11 | CAINIC01_22 | CFA_UK | 264 | 195 | 255.7 | 1.27 | 114.2 | 250.7 | Immersion |
| 12 | EXTRA18_23 | CFA_UK | 251 | 641 | 249.0 | 10.38 | 98.77 | 248.2 | Deposition |
| 13 | EXTRA18_24 | CFA_UK | 254 | 589 | 253.8 | 5.41 | 106.3 | 250.3 | Deposition |
| 14 | EXTRA18_26 | CFA_UK | 263 | 442 | 257.4 | 0.36 | 110.3 | 256.5 | Immersion |
| 15 | EXTRA18_28 | CFA_Cy | 253 | 625 | 249.5 | 0.87 | 120.1 | 248.5 | Immersion/ Deposition |
| 16 | EXTRA18_29 | CFA_Cy | 257 | 532 | ND | ND | - | - | Immersion |
| 17 | EXTRA18_30 | CFA_Ja | 249 | 650 | 245.3 | 5.59 | 113.6 | 244.5 | Immersion |
| 18 | EXTRA18_31 | CFA_Ja | 256 | 543 | 251.6 | 0.94 | 118.3 | 250.2 | Immersion |
| 19 | EXTRA18_32 | CFA_Ja | 259 | 448 | 254.5 | 0.28 | 116.0 | 253.2 | Immersion |
| 20 | EXTRA18_33 | CFA_Wh | 249 | 578 | 245.0 | 4.53 | 118.8 | 244.2 | Deposition/ Immersion |
| 21 | EXTRA18_34 | CFA_Wh | 256 | 486 | 252.1 | 3.03 | 117.3 | 251.1 | Immersion |
| 22 | EXTRA18_35 | CFA_Wh | 259 | 405 | 255.6 | 0.39 | - | 253.8 | Immersion |
| 23 | EXTRA18_36 | CFA_Mi | 249 | 612 | 245.6 | 0.30 | 125.2 | 244.8 | Immersion |
| 24 | EXTRA18_37 | CFA_Mi | 255 | 536 | 250.5 | 0.11 | 120.8 | 250.5 | Immersion |
| *Here, the onset freezing temperature is defined as the temperature where > 0.1 % of the particles were ice-active. ND = no data | | | | | | | | | |

**Figures**

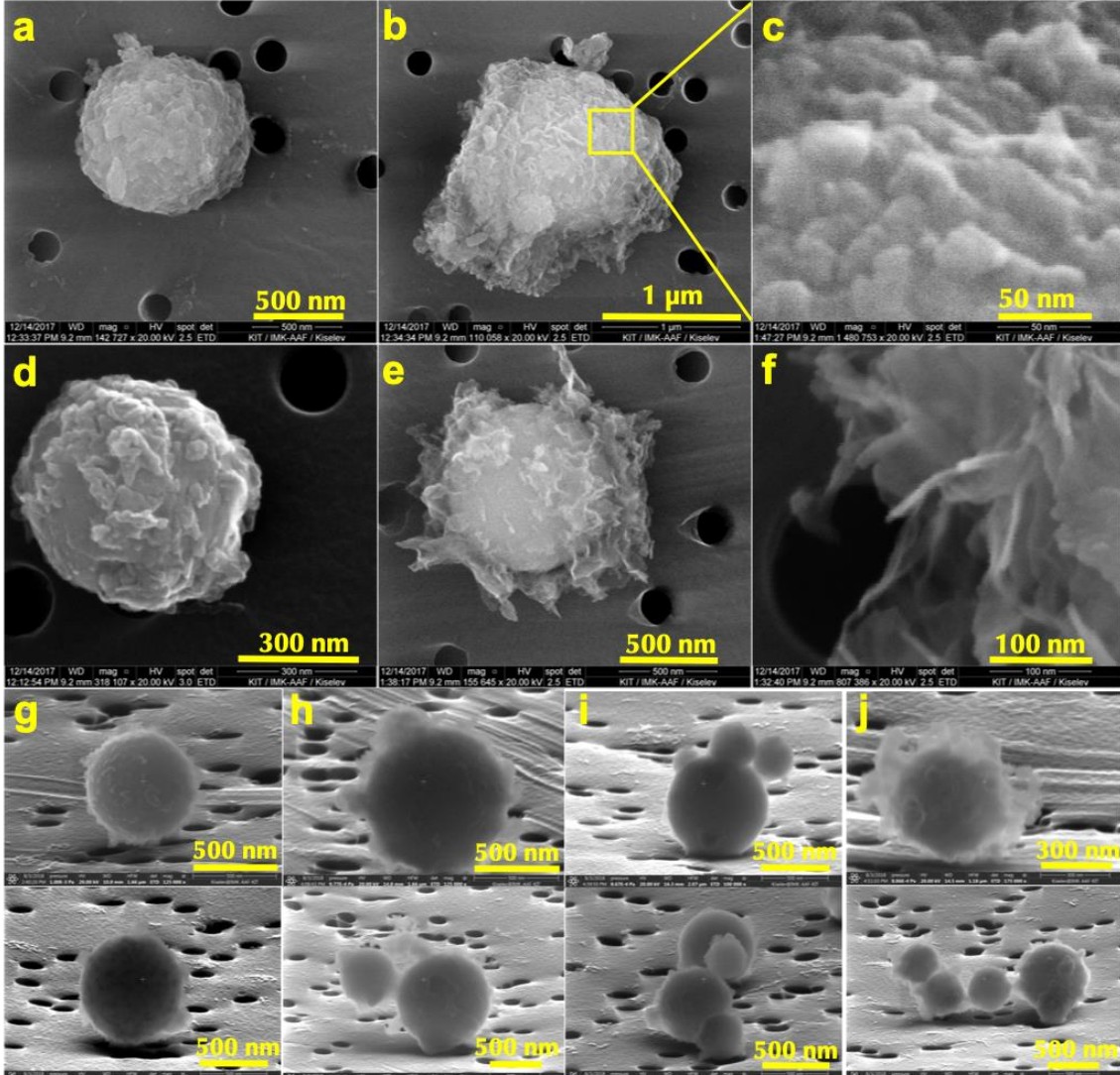

**Figure 1:** Scanning electron microscopy (SEM) images of CFA_UK particles (a - f), CFA_Cy (g), CFA_Mi (h), CFA_Ja (i), and CFA_Wh (j). All particles have a basic spherical shape, which is common to coal fly ash particles: (a) Spherical shape of CFA_UK with surface defects, (b) Meshy or spongy material on the particle surface which looks highly porous, (c) a high magnification image (~ 50 nm) of the pores or surface defects on the CFA_UK aerosol particles, (d) the core of CFA_UK shows a spherical shape like image (a) with scaly materials on the surface, (e) despite the flake-like network materials on the surface of the CFA particles - the basic spherical core is still intact. (f) high magnification of the flaky, meshy material on the particle surface, (g) CFA_Cy particles also show some degree of deposits on the surface, (h) CFA_Mi with light meshy material compared to CFA_UK, (i) CFA_Ja particles with non-smooth surface, and (j) CFA_Wh particles with a denser flaky network material on the surface than the other USA CFA samples. Images of the USA CFA particles taken by Garimella (2016) also showed scaly materials on the surface of the particles. However, CFA_UK particles had more defects and materials on the surface which were very irregularly-shaped.

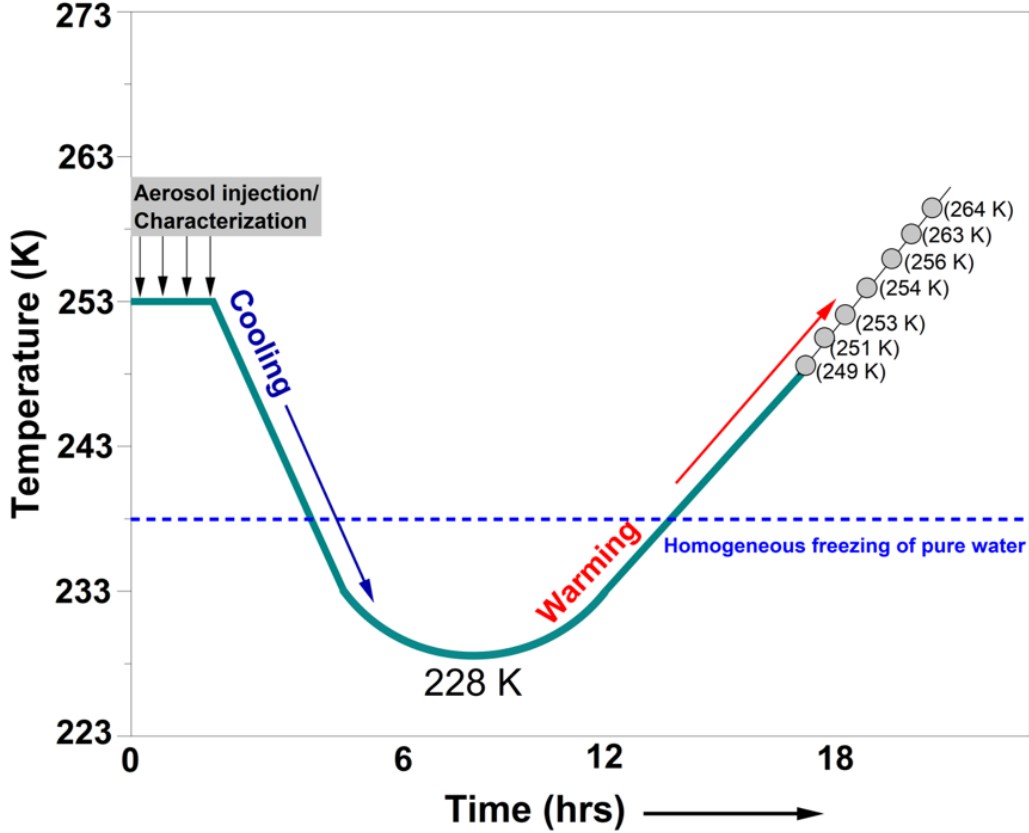

**Figure 2:** A schematic showing the temperature-cycling and freezing (TCF) process adopted in our experiments. The temperatures indicated by the grey circles represent the start temperatures ($T_{start}$) for the ice nucleation experiments conducted after the warming of the AIDA chamber. For each CFA sample, only a subset of the indicated starting temperatures was chosen to conduct the expansion cooling runs (see Table 2). The start temperature of the successive experiment was individually selected based on the degree of activity observed in the previous freezing experiment. The x-axis denotes the overall timescale of the procedure. The homogeneous freezing line of pure water is an indication of the temperature where supercooled water droplets were observed to freeze in previous AIDA experiments (Benz et al., 2005; Schmitt, 2014).

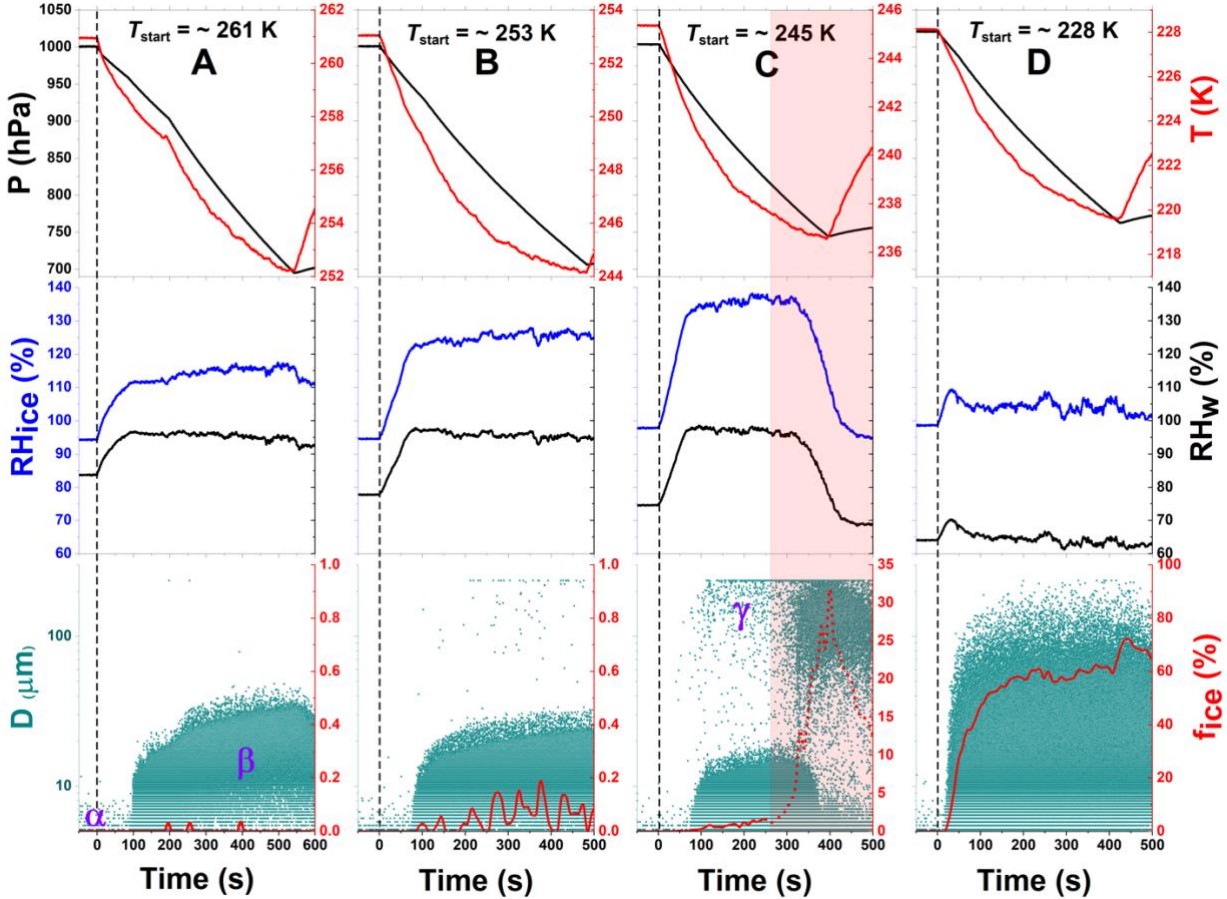

**Figure 3:** Ice nucleation experiment data for unprocessed CFA_UK particles at 261 K, 253 K, 245 K, and 228 K start temperatures ($T_{start}$). These data correspond to experiments #2, #3, #4 and #1 in Table 2, respectively. Each column (A, B, C, and D) has 3 plot panels – top, middle, and bottom. The top panels show the pressure (hPa, black) and the mean gas temperature (K, red) profiles of the AIDA aerosol and cloud simulation chamber throughout the duration of the experiment. The middle panels indicate the changes in the relative humidity with respect to ice ($RH_{ice}$, blue) and water ($RH_w$, black), both in %. The bottom panels illustrate the data for the optical size measurements from the OPCs (green dots). Greek letters point to the various types of particles detected, $\alpha$, CFA seed aerosol particles, $\beta$, cloud droplets, and $\gamma$, ice crystals (see text for details). The bottom panels also include the ice-activated fraction (%) of the aerosol particle population ($f_{ice}$, red line). In column C (bottom panel, shaded region), there is a sudden increase in the number concentration of ice particles due to the onset of homogeneous freezing. The ice-activated fraction due to the homogeneous freezing of water droplets is denoted by the dashed red line to separate it from the heterogeneous immersion freezing mode.

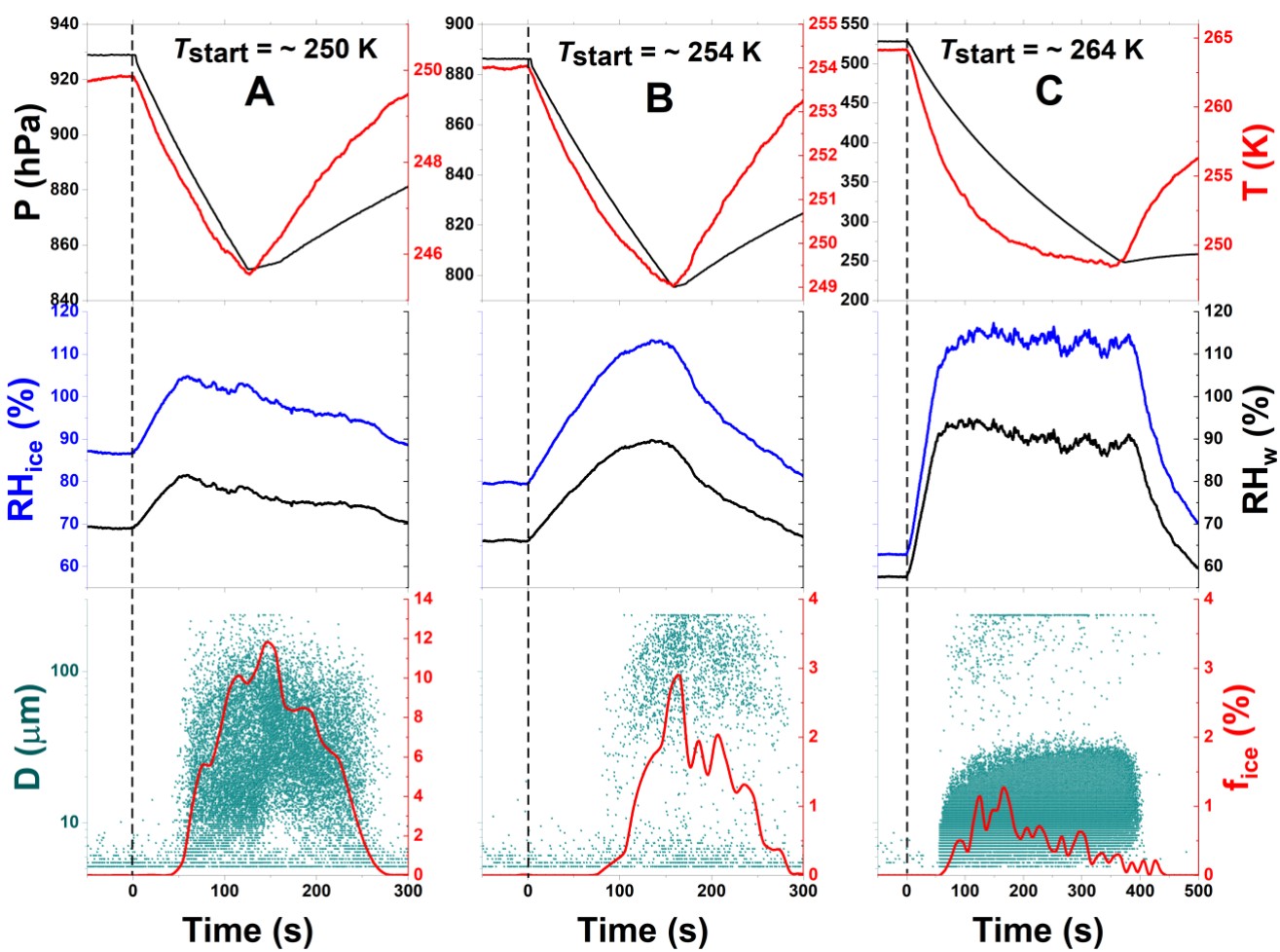

**Figure 4:** Freezing experiment data for processed CFA_UK particles at 250 K, 254 K, and 264 K start temperatures ($T_{start}$). These data correspond to experiments #9, #10, and #11 in Table 2, respectively. Processing involved the intermediate cooling of the particles to 228 K (see Figure 2). The individual panels contain the same data types as in Figure 3.

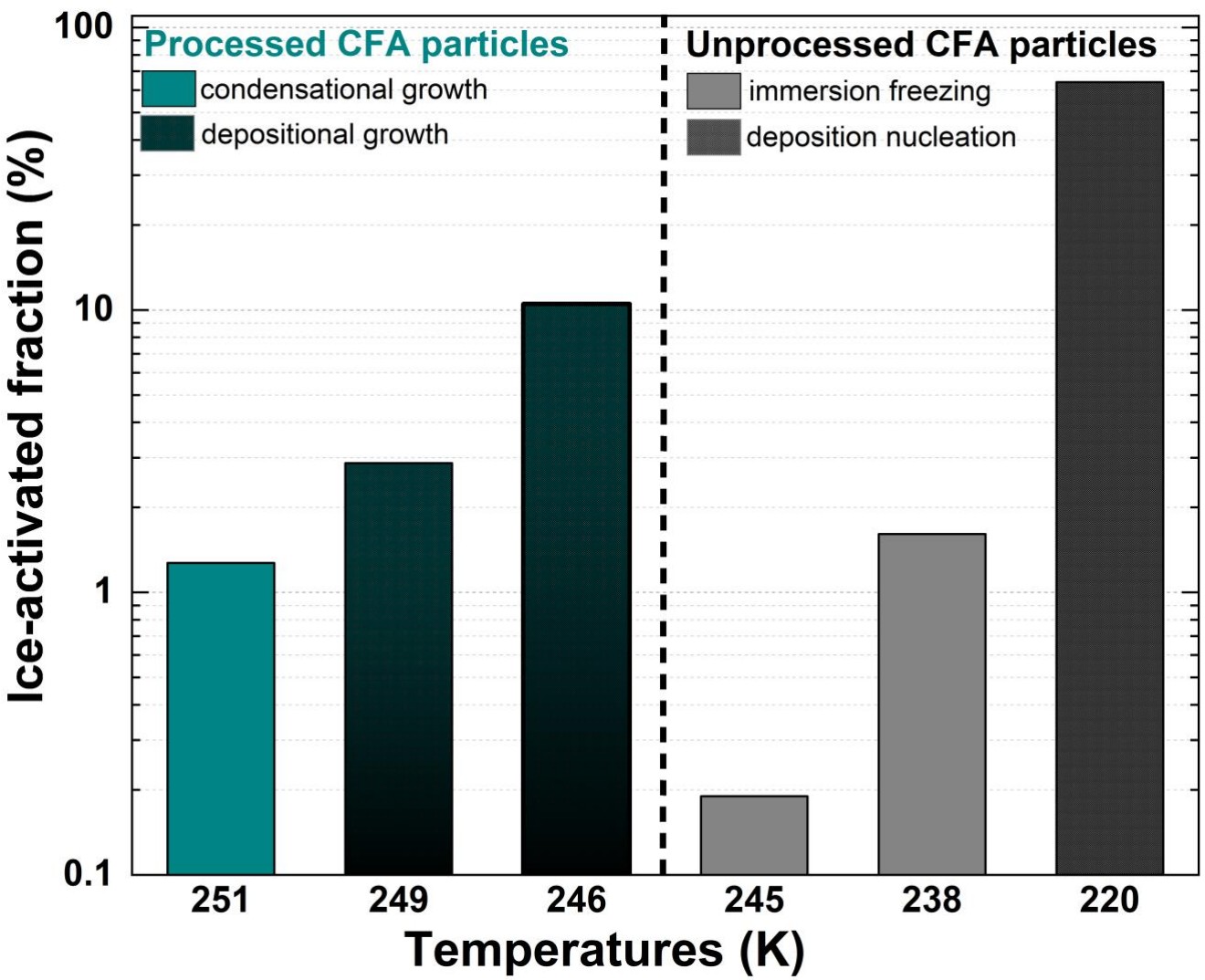

**Figure 5:** Summary of the maximum ice-activated fraction (%) of unprocessed and processed CFA_UK particles as a function of temperature. The temperatures referenced on the x-axis are the temperature at which the maximum ice-activated fraction was reached during each experiment. The grey/black columns on the right-hand side of the plot indicate experiments before the TCF procedure and the cyan/dark cyan columns on the left-hand side show experiments after the TCF process.

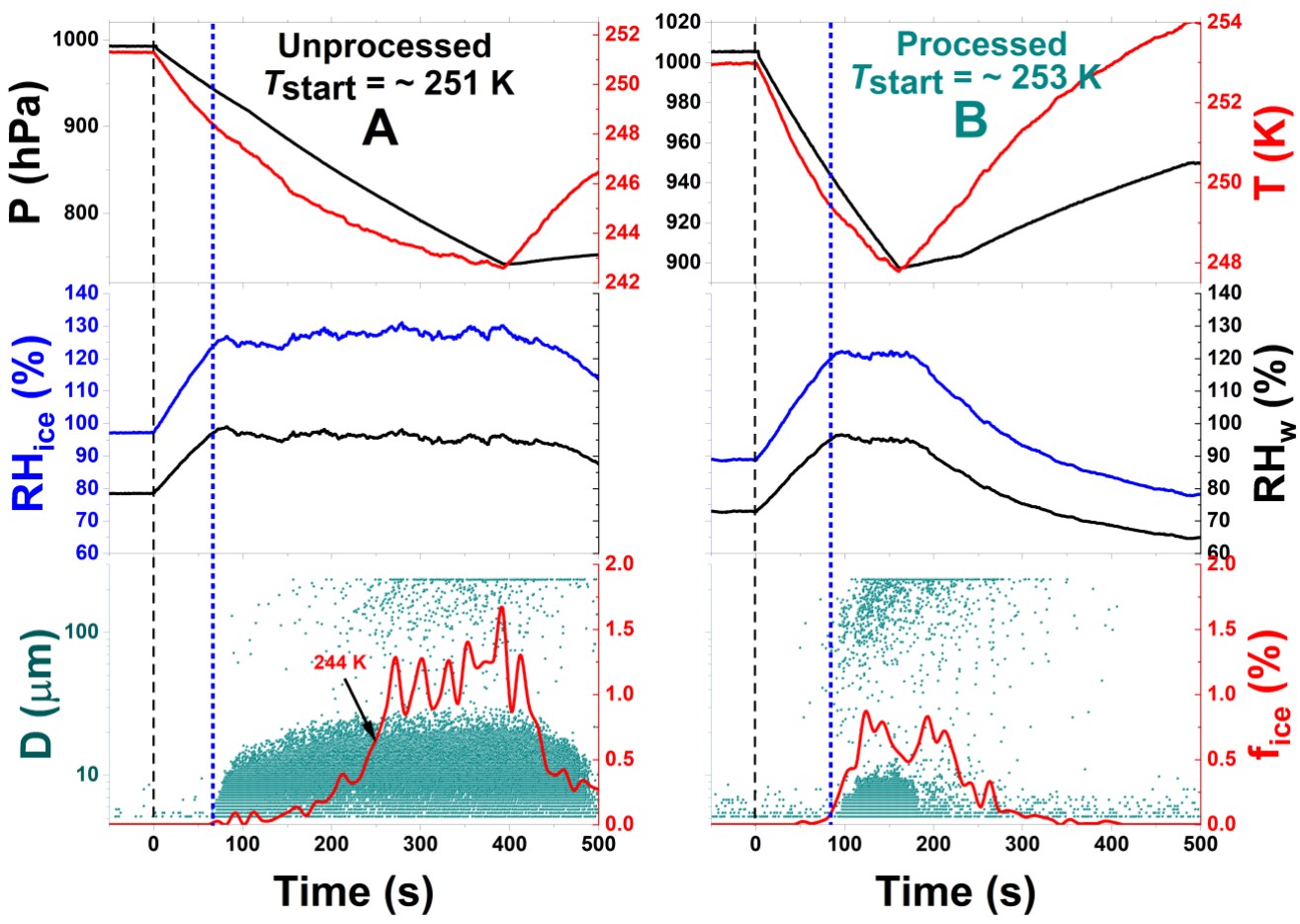

**Figure 6:** Freezing experiment data for unprocessed and processed CFA_Cy particles at 251 K and 253 K start temperatures ($T_{start}$). These data correspond to experiments #5 and #15 in Table 2, respectively. The individual panels contain the same data types as in Figure 3. The short-dashed blue lines indicate the beginning of the cloud droplet formation.

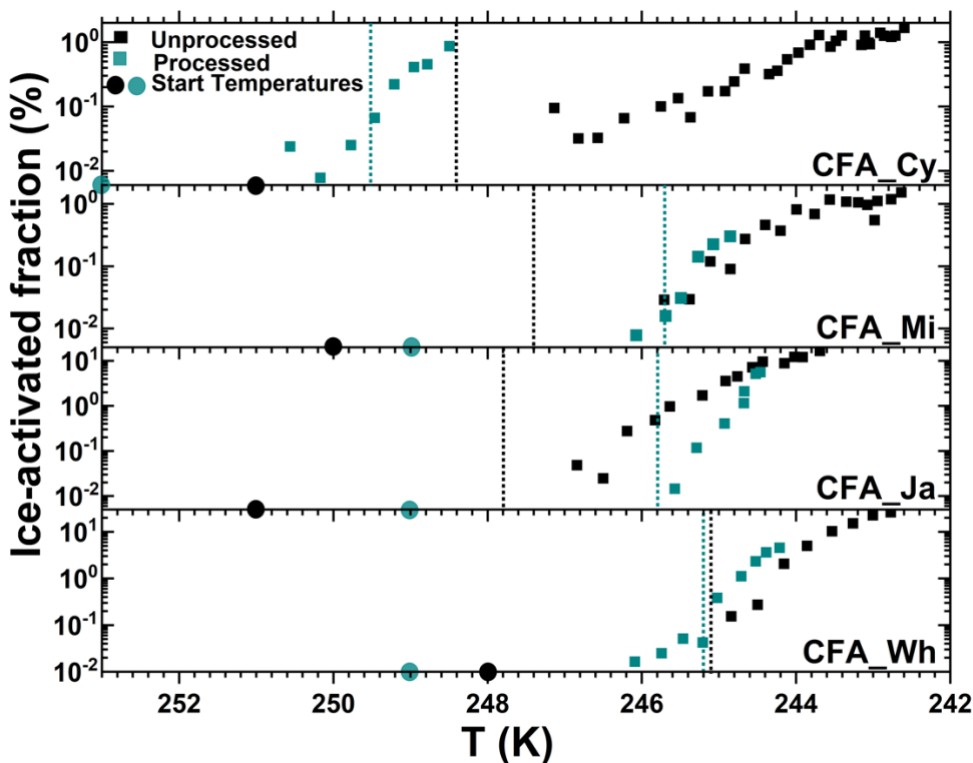

**Figure 7:** Summary of the ice-activated fraction (%) of unprocessed and processed CFA_Cy (5 and 15), CFA_Mi (8 and 23), CFA_Ja (6 and 17), and CFA_Wh (7 and 20) particles as a function of temperature. The number in the brackets are the corresponding experiments numbers in Table 2. The black data points show experiments before the temperature-cycling and freezing (TCF) procedure and the cyan data points represent experiments after the TCF process. The dotted lines correspond to the temperature where water saturation was reached for each experiment.

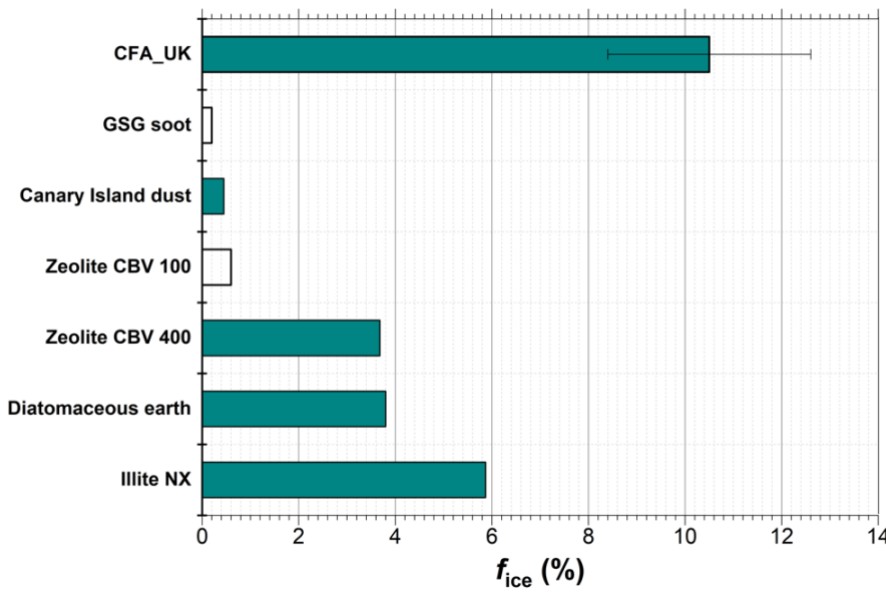

**Figure 8:** Comparison of the ice nucleation enhancement of CFA particles and other particles studied by Wagner et al. (2016). Ice-activated fraction (%) obtained at ~ 250 K start temperature is compared. The filled bars represent ice nucleation via depositional growth while the unfilled bars represent ice nucleation via condensational growth.

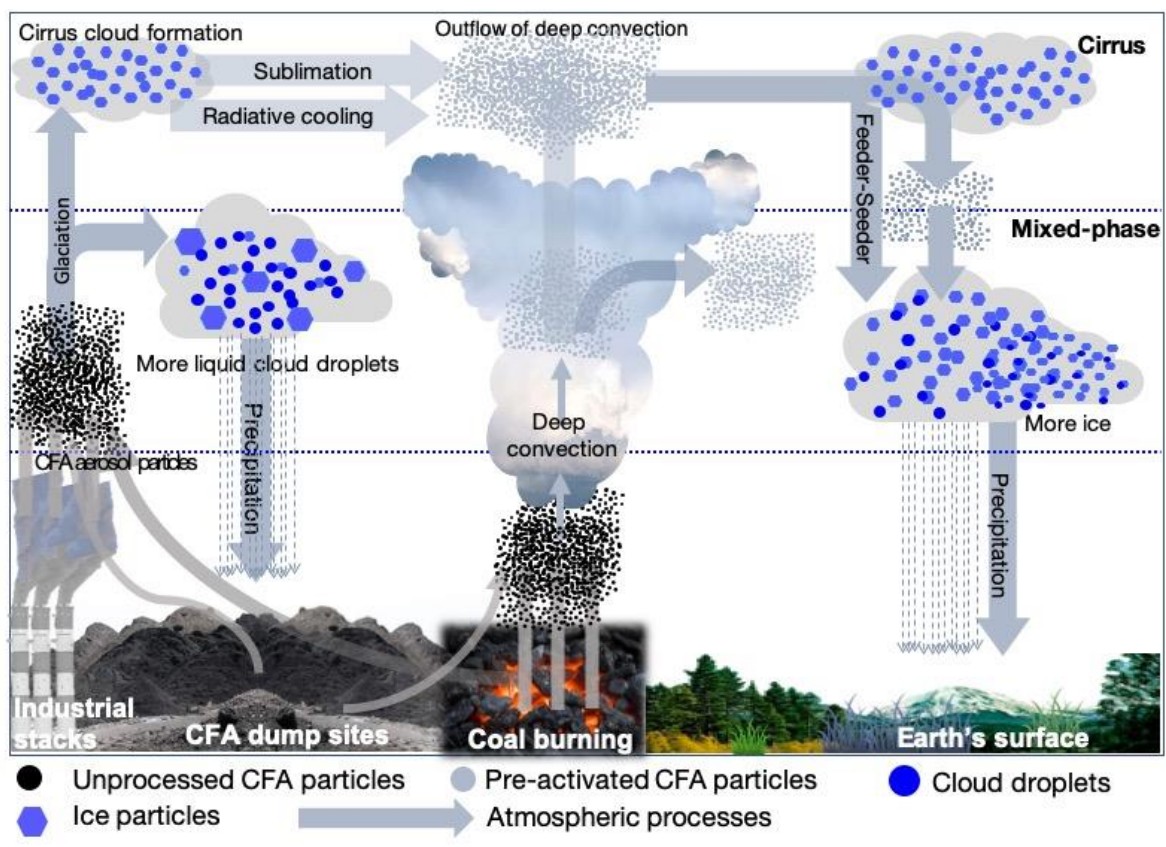

**Figure 9:** A schematic showing possible pathways and interactions of CFA particles in the atmosphere. The arrows represent possible pathways and atmospheric processes that may be relevant for the PCF mechanism in mixed-phase and cirrus cloud regimes. After the emission of the particles to the atmosphere, they can directly trigger heterogeneous ice formation in both cirrus and mixed-phase clouds (left-hand-side). The processing of these particles through lower temperatures can promote ice formation by the pore condensation and freezing mechanism (middle) and generally influence the hydrological cycle.