# Peer review of "Enhanced ice nucleation activity of coal fly ash aerosol particles initiated by ice-filled pores"

_Atmospheric Chemistry and Physics, 2019_

## Referee Comment (RC1) · Anonymous Referee #1 · 25 Feb 2019

**Review Umo et al., 2019:**

**General comment:**

Umo et al. present ice nucleation experiments performed within the large cloud simulation chamber AIDA with different coal fly ash (CFA) samples collected from five different power plants, one situated in the UK and four in the USA. Samples were sieved to isolate the size fraction up to 20 µm diameter and characterized by environmental scanning electron microscopy. In addition, their specific surface areas and pore volume were determined by argon adsorption measurements. There were quite significant differences between the ice nucleation activities of the different CFA samples. The UK sample, which is the best investigated one, showed a strong increase in the ice-active fraction for experiments performed just below the homogeneous freezing temperature of pure water. The authors concluded that this could be related to a pore condensation and freezing process (PCF). To further substantiate the role of pores for the ice nucleation ability of CFA, temperature cycling experiments were performed within the AIDA chamber by precooling the injected particles to 228 K at RH slightly below ice saturation before performing an expansion at warmer temperature. A strong pre-activation was found for the particles with the highest specific surface area and porosity. The authors conclude that the PCF mechanism could be prevalent for the ice nucleation at cirrus temperature and also significant for mixed-phase clouds when CFA particles are injected from higher altitudes.

This study presents innovative experiments aiming to elucidate the relevant ice nucleation mechanisms under cirrus and mixed-phase cloud conditions. Experiments were performed with CFA particles, which are a relevant class of ice nucleating particles from anthropogenic sources. The manuscript is well suited for Atmos. Chem. Phys. and can be recommended for publication after the following points have been addressed satisfactorily:

For all five CFA samples, pre-cooling experiments were described and discussed in the manuscript but only for the CFA UK sample, expansions that reached homogeneous freezing temperatures were mentioned. Have such measurements been carried out also for the CFA samples from the US? If yes, they should be described and discussed in the manuscript.

Experiments of processed and unprocessed samples with different starting temperatures are often compared without discussing the effect of the starting temperature. It would have been more meaningful if processed and unprocessed samples were compared in experiments with the same starting temperature. When such data is not available, the discussion needs to be improved to take the influence of the starting temperature better into account.

The individual experiments need to be characterized better. Table 2 gives an overview over all experiments relevant for this study; however, since the text and figures do not refer to this table, it does not help to obtain an overview over the experiments. Moreover, the table just gives the starting condition and lists no results except the observed freezing mode. The experiments are characterized by their starting temperature throughout the manuscript. Unfortunately, this information is not very useful because the RH and temperature at the freezing onset can only be guessed based on the starting temperature of the expansion. It would be helpful if the RH and the temperature of freezing onset together with $f_{ice}$ were added to Table 2. Moreover, the experiment name should be mentioned in the figure captions, so that the exact conditions can be looked up in Table 2.

Throughout the manuscript, the consistency of the use of present and past tense needs to be checked. Some sentences are hard to understand and should be clarified. Some examples are given in the special comments but the whole manuscript should be worked over.

**Specific comments:**

Page 1, line 25: what is meant here by "partly"? Homogeneous freezing temperatures were only reached with CFA UK. Is this statement based on experiments that are not shown?

Page 4, Sect. 2.5: the adsorption and desorption isotherms should be given as supplementary information.

Page 5, line 26: is the end of the freezing experiment the end of the expansion?

Page 5, line 29: according to Sect. 2.3 the particles should be < 2.5 μm and not just < ~10 μm? Can you comment on this?

Page 7, line 2: do you mean "at" instead of "from"?

Page 7, lines 2 – 5: It is insinuated here that the generation method (dry vs. wet) might influence the ice nucleation activity of CFA particles. However, for a consistent comparison, the available INP area also needs to be taken into account. In Umo et al. (2015), the ice nucleation active site density does not rise above $10^5$ $cm^{-2}$. The surface area of a spherical 1 μm radius particle is only $10^{-7}$ $cm^2$. Therefore, if the CFA particles had the same ice nucleation ability as reported in Umo et al. (2015), only a minor fraction of the particles should be active, which is indeed in accordance with the AIDA experiments.

Page 7, line 5: do you mean "at" instead of "from"?

Page 7, lines 12 – 13: "In this work, the average median particle diameter was 0.58 μm for our CFA samples theirs was size-selected to 0.3 μm." Improve formulation.

Page 7, line 20: While the ice nucleation activity of the CFA particles investigated here is compared to the ones of other studies and other aerosol types, a comparison of the CFA particles investigated in this work among each other is lacking. Figure 7 shows that the ice nucleation activity of CFA_JA and CFA_Wh is one order of magnitude larger than the one of CFA_Cy and CFA_Mi. Are there differences in morphology, elemental composition, or surface functionalization that might explain the differences?

Page 7, lines 31 – 33: "This occurred at a lower $RH_{ice}$ = ~105 % than the experiment with unprocessed CFA_UK particles which $RH_{ice}$ = ~130 % (corresponding to water saturation)." Improve formulation.

Page 8, lines 7 – 8: Was this the third expansion of the same sample or an expansion with a new sample? Please clarify.

Page 8: lines 33 – 35: "At $T_{start}$ = 253 K, the $f_{ice}$ for CFA_Cy particles after the pre-activation process was ~0.86 % slightly lower than what was observed for the unprocessed CFA_Cy particles." Improve formulation.

Page 9, lines 2 and 3: "$T_{start}$ = 255 K (Fig. 7)": Where is this starting temperature shown in Fig. 7?

Page 9, line 10: "256 K start temperature (Fig. 7)": Where is this starting temperature shown in Fig. 7?

Page 9, lines 10 – 12: "Again, for the processed CFA_Ja particles, no appreciable enhancement of its ice formation abilities was observed as the $f_{ice}$ at $T_{start}$ = 249 K was 2 % at $RH_{ice}$ = ~125 %." Do you mean "enhancement compared to the unprocessed CFA_Ja?" Yet, Fig. 7 shows a decrease in $f_{ice}$ rather than "no appreciable enhancement". Please clarify.

Page 9, lines 15 – 16: "We cannot completely rule out that the actual formation mechanism in both scenarios after the temperature cycling is not via a condensational freezing pathway." Formulate clearer, avoid double negative.

Page 9, lines 16 – 17: "This was not seen for the unprocessed CFA_Ja particles; after reaching water saturation, there was a time lag before ice particles were detected." Is this a valid comparison? According to Fig. 7, the unprocessed CFA_Ja sample had a higher starting temperature. The onset of $f_{ice}$ was therefore still at a warmer temperature for processed compared with unprocessed particles. This difference should be taken into account when discussing the effect of processing.

Page 9, line 26: $T_{start}$ = 256 K is not shown in Fig. 7.

Page 9, line 28 – 29: "occurred in a shorter temperature step": please formulate better.

Page 9, line 30: "pre-activation by PCF may not be very important compared to other particles that are less ice-active". Improve formulation.

Page 9, line 33: "and the relative humidity which summary is given in Figs 6 & 7." Improve formulation.

Page 10, lines 29 – 30: An additional factor that should be discussed here is the competition between pre-activation and immersion freezing. The ice nucleation mode is given in Table 2 but this table is neither connected with the text nor with the figures. Figure 7 needs to be improved to clearly state when water saturation is reached.

Page 10, lines 35 – 36: "Depending on the transport of CFA particles in the atmosphere, they can pass through different altitudes and temperature regimes which can naturally provide a temperature-cycling and freezing process for these particles to be pre-activated." Improve formulation.

Page 11, line 8: "There is a need by the modelling community to study the impact that…". Improve formulation.

Page 11, line 26: This is the first and only time that the chemical compositions of CFA particles is mentioned in this manuscript. Indeed, the chemical composition might be relevant to explain the differences in immersion freezing of the different CFA samples. If chemical composition is mentioned in the conclusions, it should also be discussed in the section "Results and Discussion".

Page 18, caption of Table 1: How was the median diameter determined?

Page 18, Table 2: Consider to add $f_{ice}$ and the freezing onset temperature to Table 2.

Figure 3: the tags on the y-axes should be increased for better visibility. The measurements shown in this figure should be related to the experiments listed in Table 2.

Figure 6, figure caption: It should be made clear whether the start temperatures of the experiments are shown in this figure.

Figure 7: It would be helpful to indicate the temperature where water saturation is reached for all experiments. Consider to add experiments with CFA_UK for better comparison.

**Technical comment:**

Page 4, line 10: "range" might be more adequate than "limit" since a range is given in brackets.

Page 4, line 28: "microscope" instead of "microscopy".

Page 5, line 31: "β" does not appear correctly in the pdf.

Page 5, line 33: "γ" does not appear correctly in the pdf.

Page 8, line38: "their" instead of "its".

Page 9, line 29: remove "an".

Page 10, line 14: "than those of other CFA particles" instead of "than the other CFA particles".

Page 12, line 3: "need" instead of "needs".

Page 14, line 26: John is the first name of J. G. Morris. Please revise reference.

---

## Referee Comment (RC2) · Anonymous Referee #2 · 25 Feb 2019

**Review of "Enhanced ice nucleation activity of coal fly ash aerosol particles initiated by ice-filled pores" by N. S. Umo et al. for Atmospheric Chemistry and Physics**

General comments:

In the present study, N. S. Umo and co-workers show and discuss the results of temperature cycling experiments with coal fly ash (CFA) particles at the AIDA cloud chamber. The aim of these experiments was to clarify whether the ice nucleation activity of CFA is increased by the pore condensation and freezing (PCF) mechanism under certain conditions. The authors achieve to convincingly demonstrate that this is the case for some of the used samples. The question of why some samples are more prone to PCF than others is not convincingly answered, but this cannot be expected in the case of CFA, which is a very complex and heterogeneous substance. From my point of view, the study is an interesting addition to recent findings concerning the ice nucleation behavior of CFA particles (Umo et al., 2015; Grawe et al., 2016; Garimella, 2016; Grawe et al., 2018; Losey et al., 2018). However, there are some content-related issues which need more discussion or clarification from my point of view. These are listed below (specific comments).
Parts of the manuscript could benefit from editing with respect to wording and presentation of the results but this can be easily resolved (see technical corrections). The figures are mostly clear, but I was wondering why a significant part of the results are not shown anywhere. The authors should consider including an Appendix for presenting the missing figures.
Generally, I feel that the topic fits the scope of ACP and that the study is worth publishing. To improve the overall significance and readability of the manuscript, I suggest the following minor points.

Specific comments:

1) Effect of size-dependent specific surface area and pore properties:

The investigation of PCF on CFA particles is a daunting task, since CFA is such a heterogeneous substance. I appreciate the authors' attempt at finding possible reasons for differences in the behavior of the different samples, but I also think that their approach lacks the discussion of an important point, i.e., the size-dependence of the particles' specific surface area, pore volume, and pore size. This type of information is probably very hard to come by and I do not expect the authors to perform further analyses. But it should at least be mentioned that the specific surface area and pore volume are very likely dependent of the particle size and that some particles might not even feature pores (Seames, 2003). Hence, the properties of the sieved bulk sample (0-20 μm) might not be representative for the properties of the particles that actually enter the AIDA chamber (< 2.5 μm). This should be made clear in the discussion (P9L37-P10L30).

2) Comparison to Wagner et al. (2016)

When viewing the results, I was wondering how CFA particles compare to other the substances which have already been investigated with the same instrumental setup and measurement routine by Wagner et al. (2016), i.e., zeolite, diatomaceous earth, mineral dust, volcanic ash, and soot. Concerning CFA_UK, it is mentioned very briefly (P6L19-20) that "a similar increase in the heterogeneous ice nucleation ability has been previously observed for zeolite and illite" but nothing is said about the other CFA samples and other substances tested by Wagner et al. (2016). This comparison could be expanded to put the CFA results into perspective.

3) Atmospheric implications

Although the authors discuss the atmospheric implications of their findings, they are missing one major point. How large is the probability that CFA particles reach such high altitudes where they experience 228 K? I understand that CFA particles can influence atmospheric ice nucleation close to the point of emission, but the number concentration of these particles at cirrus level is probably close to zero. Indeed, it is difficult to identify CFA particles in the atmosphere due to their similarities to mineral dust which is why there is a lack of information concerning atmospheric number concentrations of CFA particles. But despite this lack of information, the authors should not leave this issue completely unattended. A remark concerning this should be included in Sec. 3.3.

4) Explanation of the PCF mechanism

- Even though PCF has become an accepted concept in recent years, the process itself should be explained in more detail. The negative Kelvin effect, which is the reason why there is capillary condensation of water vapor at a relative humidity (RH) below water saturation, is not even mentioned. I suggest to include an explanation of the mechanism (capillary condensation of water vapor at RH below water saturation → formation of ice in pores at very low temperatures → pore ice persists as site for ice nucleation at warmer temperatures and ice-supersaturated conditions) where it is first mentioned (P2L14).
- At this point, it should also be mentioned, why PCF is restricted to certain pore sizes (P2L29). Furthermore, I expected a remark concerning the effect of the pore geometry (cylindrical pore, ink-bottle-shaped pore) on PCF. Which types of pores might be present in CFA? All of this needs to be explained in the introduction.
- Instead of Fig. 2, the authors could describe the temperature cycling process in a similar manner as Marcolli (2017; see Fig. 1-4), i.e., showing RH with respect to temperature. The time scales of the different steps could be mentioned in the caption.

5) Methodology

- The AIDA chamber is a well-established instrument but the authors should consider describing the measurement and data evaluation techniques in more detail. For example, there is no mention of the uncertainty of the $f_{ice}$ determination. How does the large error of ±20 % of the ice particle number concentration affect the $f_{ice}$ error?
- The explanation of how $f_{ice}$ is calculated should be included in Sec. 2.6, not in Sec. 3.
- I do not understand how droplets can form in an environment which is slightly subsaturated with respect to liquid water (see Fig. 3, panels A and B). It looks like the black line (RH with respect to water) is below 100 % throughout the duration of the experiments. Is this due to the measurement uncertainty? Please clarify.
- P5L17-18: How can the air in the AIDA chamber be subsaturated with respect to ice when there is an ice layer on the inner chamber walls? Shouldn't it be saturated? Please explain.

6) Presentation of the results

- I was wondering why ice nucleation surface site densities are not included in the discussion of the results. The authors state that "Further analyses on the distribution of the ice nucleation active sites densities of these CFA particles is outside the scope of the current report and will be presented in a separate communication." (P6L32-34). An inclusion of these data in the current paper would make more sense to me, especially for a comparison of the intrinsic ice nucleation behavior of the CFA samples to the results other studies (P6L35-P7L21). Besides, which new information would this other report contain?
- The mentioning of $T_{start}$ instead of the actual temperature at which $f_{ice,max}$ values were derived is not intuitive (see P6L27-28, Sec. 3.2, …). Please include both $T_{start}$ and the actual $T$ in your discussion.
- As stated above, parts of Sec. 3.2 are hard to follow. In the cases of CFA_Mi, CFA_Ja, and CFA_Wh, the authors describe specific features observed in the experiments, but they do not show the corresponding 3-panel plots. Some results are not even included in Fig. 7. I suggest to include the described examples in an Appendix so that the reader can follow the discussion.
- I find the amount of arrows, overlapping shapes, and different colors in Fig. 8 very confusing. What is the difference between the blue, red, light gray and dark gray arrows? What is the difference between the black, light gray, and dark gray particles? A legend, or at least an explanation in the caption, would be helpful. Also, it looks like the sublimation of ice particles directly leads to cloud formation. The authors should revise and thin out this figure to make it more intuitively understandable.
- Why is the CFA_UK sample discussed separately from the U.S. American samples? Fig. 6 and 7 should be combined. I also suggest to change from a bar graph to a scatter plot for more clarity. A change to a logarithmic y-axis should be considered for both Fig. 6 and 7.

Technical corrections:

I generally feel that the authors need to be more precise with their formulations. Sentences and paragraphs are sometimes lengthy due to unnecessary fillers and repetitions. Transition words are partly misleading. There are grammatical errors. Below, I list the issues that caught my eye but I advise the authors to recheck their manuscript carefully to improve readability and understandability.

| | |
|---|---|
| P1L26 | At which RH was this strong increase observed? |
| P1L27 | Change to either "undergoing PCF" or "undergoing **the** PCF mechanism". |
| P1L31-35 | This sentence would benefit from being split into two. |
| P1L36-39 | It is a bit unfortunate that you refer to PCF in general in the first sentence and specifically to PCF on CFA particles in the second. At least this is how I understood it. Please reword. |
| P1L36 | I suggest to introduce "intrinsic" as relating to unprocessed or not pre-activated particles. |
| P1L37 | Change to "on the other **hand**". |
| P1L41 | Change to "highly relevant to **our understanding/knowledge/ comprehension… of** cloud formation". |
| P2L3 | Omit "primary". Heterogeneous ice nucleation is always a primary process. Also, define the terms "homogeneous" and "heterogeneous" before using them. |
| P2L8 | Change to "with **the surface of** a …". |
| P2L11 | Omit "however" at the start of the new paragraph. |
| P2L12-13 | Change to either "such **a** particle" or "such particles". |
| P2L19 | Change to "before ice nucleation **takes place**". |
| P2L20 | Change to "Here, we define…". |

| | |
|---|---|
| P2L21 | There are references to Wagner et al. (2016a), Wagner et al. (2016b), and Wagner et al. (2016). Yet, there's only one Wagner et al. paper from 2016 in the reference list. Please correct. |
| P2L27 | "zeolite" and "illite" should not be capitalized. |
| P2L34-36 | This sentence does not say anything else than the one on P2L16-18. I suggest to remove it (also the following sentence). |
| P2L39 | Change to "global **cloud** ice budget". |
| P2L41 | A "significant amount" is not very precise. Are there really no estimates of the emitted CFA mass? |
| P3L4 | There is no question that immersion freezing was investigated by Grawe et al. (2016, 2018) and Umo et al. (2015). Hence, differences in the freezing behavior of the investigated samples are not due to differences in the freezing mechanism. Please remove this part of the sentence. Furthermore, Grawe et al. (2018) showed that the immersion freezing behavior of CFA can be strongly dependent on the amount of time that the particles spend immersed in the droplet prior to the initiation of freezing. This issue is worth mentioning here because it can also affect the immersion mode AIDA measurements. |
| P3L6 | "various atmospheric conditions". Be more precise. |
| P3L7 | Change to "different CFA **samples**". Also on P3L14. |
| P3L15-17 | "The results from these new laboratory measurements are presented in this report." This sentence is unnecessary and should be deleted. |
| P3L23-25 | How representative is material from the EPs in comparison to material that is emitted into the atmosphere. Please include a statement concerning this matter. |
| P3L25 | Omit "However". |
| P3L25-29 | I am aware that the authors do not focus on the effect of chemical composition of the samples on their ice nucleation behavior. However, it would be interesting to include some more information, e.g., are the samples of class C (high Ca) or class F (low Ca), since it has been shown that the composition affects the ice nucleation measurements. This information is easily obtainable from Garimella (2016) and should be mentioned here. |
| P3L29 | Please note that Losey et al. (2018) investigated the same sample set. This publication should be referenced here as well. |
| P3L30 | Change "name" to "operators/owners". The name itself cannot prefer anonymity. |
| P3L31-33 | Were the other samples not sieved? Why not? |
| P4L3 | Please combine the instrument abbreviation and the manufacturer in one set of parentheses to avoid "(…) (…)". Check throughout the manuscript. |
| P4L20 | Which types of cyclones were used? What is their cut-off diameter? |
| P4L26 | Change "Min" to min". |
| P4L27 | Change "mins" to "min". |
| P4L27 | How does this coating affect the morphology of the particles? Could pores potentially be covered? Please include a short statement. |
| P4L36 | "argon" and "nitrogen" should not be capitalized in running text. |
| P5L21 | "inherent" and "intrinsic" seem to be used synonymously throughout the manuscript. I suggest to avoid the use of both terms and stick to one. |
| P5L31,33 | Make sure that the empty squares are replaced by the Greek letters in the new version of the manuscript. |
| P5L35 | Since you list all experiments in Table 2, you could also include the used size thresholds there. |
| P6L11 | Change "$t = \sim 300$ s" to "$t \sim 300$ s". Also in all other occurring instances. |
| P6L25 | Insert "–" behind "CFA_Wh". |
| P6L26 | Insert "mode" behind "immersion freezing". |
| P6L35- | |
| P7L21 | I agree that a comparison to previous ice nucleation studies with CFA is interesting. However, by only reporting onset ice nucleation temperatures, the reader does not get an idea how the here investigated samples compare to |

those from previous studies. This could be resolved by including a figure showing $n_s(T)$.

P7L3-4 "Both studies can access warmer freezing temperatures for INPs than the dry generation method that our system is designed for.". Please explain shortly why this is the case.

P7L5 The Schnell et al. (1976) reference is not a good choice here. Actually, in this study "no detectable effects from a coal-fired powerplant plume" (see title) on atmospheric ice nucleation were found. Better cite Parungo et al. (1978), who conducted a similar experiment and found an enhanced ice nucleation efficiency of the plume aerosol in comparison to the background aerosol.

P7L10 Which of the two studies are you referring to?

P7L10 Actually, Grawe et al. (2018) state that the amount of hydratable components is important and that quartz only contributes in those samples which contain a small concentration of hydratable components. But this could also be included in the introduction (P3L4).

P7L13 Please cite Garimella (2016) instead of Welti et al. (2009). Garimella (2016) investigated 300 and 700 nm CFA particles, not mineral dust, and found that the immersion freezing efficiency does not scale with the surface area as the smaller particles were relatively more efficient than the larger ones.

P7L17 Please cite Grawe et al. (2018) instead of Hiranuma et al. (2018). Firstly, the manuscript by Hiranuma et al. (2018) is still under review. Secondly, the study by Grawe et al. (2018) is more relevant for the here presented work as they discuss the methodology-dependent freezing behavior of CFA particles, not cellulose.

P8L11-13 Does this mean that 1.2 % of the particles contained pores suitable for PCF? If yes, then please say so.

P8L14 Please check the Mahrt et al. (2018) reference. Mahrt et al. (2018) indeed describe PCF, but they did not conduct temperature cycling experiments. They saw a stepwise increase in the activated fraction of one type of soot particles due to condensation in pores and subsequent homogeneous freezing.

P8L23 Figure 6 is discussed before Fig. 5. Please arrange the order of the attached figures accordingly.

P8L38-39 Please check this sentence for correctness. It does not relate to the previous statement and it seems that something is missing.

P9L5-7 Change beginning of the sentence to "**This** suggests..".

P9L7-8 "Confirm" is a very strong word here, given that the data of this example is not even shown and given that no error estimation for the $f_{ice}$ error is provided.

P9L9 Change to "droplet activation". Also on P9L17.

P9L16 Please omit "One thing is extremely clear that …".

P9L19-30 This paragraph would profit tremendously from the inclusion of a 3-panel-plot of the measurement that you are describing here.

P9L25 I do not understand the use of the word "although" here.

P9L28-29 "ice formation occurred in a shorter temperature step". I am not sure what you mean here. The temperature at which $f_{ice,max}$ was registered is of interest, not the temperature range. Please reword.

P9L32 Change "comparison" to "ranking".

P9L33 Change "$f_{ice}$" to "$f_{ice,max}$".

P9L33 "which summary". Change to form a grammatically correct sentence.

P10L21-23 Please reword this sentence.

P10L25 Is it realistic that cenospheres or plerospheres would be filled by capillary condensation under the conditions in your experiments? How large are the spheres and how large are the openings? According to Marcolli (2017), large pores need very high RH or very low temperatures to be filled.

P10L26 Fischer et al. (1976), who first discovered cenospheres and plerospheres, is the more appropriate reference here.

| | |
|---|---|
| P9L27-30 | I suggest to change the formulation in such a way that it becomes clear that an estimation of the pore size and geometry is **not possible** for CFA. This is due to the heterogeneity of the particles. Other INP types might be better suited for such an estimation. Avoid using the word "pointless". |
| P10L37 | "Rainout" is probably not the right term for cases where only the ice phase is involved. |
| P10L37-38 | I cannot find this statement in the given reference. Please check and remove if necessary. |
| P11L3-4 | This is shown in a very confusing way in Fig. 8. Please adjust. |
| P11L12-13 | "Currently, this is not well understood and requires further research." This sentence can be omitted. The following sentence is completely sufficient. |
| P11L14 | Change "clouds formation" to "cloud formation". |
| P11L15 | Change "studying the dominance of this occurrence" to simply "the occurrence". |
| P11L16 | Change "cloud system" to "cloud system**s**". |
| P11L21 | "However" does not seem like an appropriate transition word. The statement is not in contrast to the previous one. Please reword. |
| P11L22-23 | I am aware that there are lots of different pore types ("pores", "crevices", "cavities"), but it might be best to define one term in the beginning and stick to it throughout the manuscript. |
| P11L25 | Change "ice formation" to "ice nucleation potential". |
| P11L33 | What is meant by "their"? The particles? The pores? Please clarify. |
| P11L35-36 | It should be mentioned here that CFA is not a suitable substance for the investigation of the effect of different pore geometries on PCF. |
| P11L40 | Omit "in this theme". |
| P12L3 | Change to "On which time scale does a potential INP need to…" |
| Fig. 1 | Please change "stuff" in the figure caption to a more scientific word. |
| Fig. 3 | Please include the actual Greek letters in the caption, not the written-out names. |
| Fig. 5 | Please explain the meaning of the blue dashed line in the caption. |
| Fig. 6 | Change to "dark cyan bar" in the caption. There is only one. |

References (which are not already included in the manuscript):

Fisher, G. L., D. P. Y. Chang, and M. Brummer (1976). "Fly ash collected from electrostatic precipitators: microcrystalline structures and the mystery of the spheres". *Science* 192.4239, pp. 553–555.

Losey, D., S. K. Sihvonen, D. Veghte, E. Chong, and M. A. Freedman (2018). "Acidic Processing of Fly Ash: Chemical Characterization, Morphology, and Immersion Freezing". *Environmental Science: Processes & Impacts* 20, pp. 1581–1592.

Parungo, F. P., E. Ackerman, H. Proulx, and R. F. Pueschel (1978). "Nucleation properties of fly ash in a coal-fired power-plant plume". *Atmospheric Environment* 12, pp. 929–935.

Seames, W. S. (2003). "An initial study of the fine fragmentation fly ash particle mode generated during pulverized coal combustion". *Fuel Processing Technology* 81.2, pp. 109–125.

---

## Author Comment (AC1) · 17 May 2019

We are sincerely grateful to the two Reviewers for their interest in our work. The comments, suggestions, and corrections they have provided are very useful feedback to us, and they have gone a long way to improve the revised version of our work. Our responses to the Reviewers' comments are written in italicized **blue** text beneath each comment. The page and line numbers referred to in this response are those of the ACPD version. All changes - corrections, omissions, and additions - are updated in the revised manuscript which is submitted alongside this response. A Supplementary Information document, added to the revised manuscript version, is also submitted.

**Reviewer 1**

**Review Umo et al., 2019:**

**General comment:**

Umo et al. present ice nucleation experiments performed within the large cloud simulation chamber AIDA with different coal fly ash (CFA) samples collected from five different power plants, one situated in the UK and four in the USA. Samples were sieved to isolate the size fraction up to 20 μm diameter and characterized by environmental scanning electron microscopy. In addition, their specific surface areas and pore volume were determined by argon adsorption measurements. There were quite significant differences between the ice nucleation activities of the different CFA samples.
The UK sample, which is the best investigated one, showed a strong increase in the ice-active fraction for experiments performed just below the homogeneous freezing temperature of pure water. The authors concluded that this could be related to a pore condensation and freezing process (PCF). To further substantiate the role of pores for the ice nucleation ability of CFA, temperature cycling experiments were performed within the AIDA chamber by precooling the injected particles to 228 K at RH slightly below ice saturation before performing an expansion at warmer temperature. A strong pre-activation was found for the particles with the highest specific surface area and porosity. The authors conclude that the PCF mechanism could be prevalent for the ice nucleation at cirrus temperature and also significant for mixed-phase clouds when CFA particles are injected from higher altitudes.

This study presents innovative experiments aiming to elucidate the relevant ice nucleation mechanisms under cirrus and mixed-phase cloud conditions. Experiments were performed with CFA particles, which are a relevant class of ice nucleating particles from anthropogenic sources. The manuscript is well suited for Atmos. Chem. Phys. and can be recommended for publication after the following points have been addressed satisfactorily:

For all five CFA samples, pre-cooling experiments were described and discussed in the manuscript but only for the CFA UK sample, expansions that reached homogeneous freezing temperatures were mentioned. Have such measurements been carried out also for the CFA samples from the US? If yes, they should be described and discussed in the manuscript.

*No, expansions just below the homogeneous freezing temperatures of pure water were not carried out for the CFA samples from the USA.*

Experiments of processed and unprocessed samples with different starting temperatures are often compared without discussing the effect of the starting temperature. It would have been more meaningful if processed and unprocessed samples were compared in experiments with the same starting temperature. When such data is not available, the discussion needs to be improved to take the influence of the starting temperature better into account.

*In the manuscript, we compared results of experiments from processed and unprocessed CFA particles at similar start temperatures. In situations where this was not the case, we selected the start temperature of the processed samples which was closest to the start temperature of the unprocessed samples. Moreover, we have improved our discussions taking into account the influence of the starting temperature.*

The individual experiments need to be characterized better. Table 2 gives an overview over all experiments relevant for this study; however, since the text and figures do not refer to this table, it does not help to obtain an overview over the experiments. Moreover, the table just gives the starting condition and lists no results except the observed freezing mode. The experiments are characterized by their starting temperature throughout the manuscript. Unfortunately, this information is not very useful because the RH and temperature at the freezing onset can only be guessed based on the starting temperature of the expansion. It would be helpful if the RH and the temperature of freezing onset together with $f_{ice}$ were added to Table 2. Moreover, the experiment name should be mentioned in the figure captions, so that the exact conditions can be looked up in Table 2.

*We have now referred to Table 2 in the text and figure captions of the revised manuscript where appropriate. Additionally, we have included more information in Table 2, namely (1) the maximum ice-active fraction reached during the experiment ($f_{ice,max}$) and (2) the corresponding temperatures ($T@ f_{ice,max}$) and relative humidities with respect to ice ($RH_{ice}@ f_{ice,max}$).*

Throughout the manuscript, the consistency of the use of present and past tense needs to be checked. Some sentences are hard to understand and should be clarified. Some examples are given in the special comments but the whole manuscript should be worked over.

*Thank you for your comments. We have worked through the entire manuscript and made necessary corrections, clarified, and simplified some sentences to improve their readability and understanding.*

**Specific comments:**

Page 1, line 25: what is meant here by "partly"? Homogeneous freezing temperatures were only reached with CFA UK. Is this statement based on experiments that are not shown?

*First, we have removed "partly" from this statement. You are correct, we performed experiments below homogeneous freezing temperatures only with CFA_UK. We have adjusted the statement to be more specific and it now reads: "In our current ice nucleation experiments with a particular CFA sample (CFA_UK), which we conducted in the Aerosol Interaction and Dynamics in the Atmosphere (AIDA) aerosol and cloud simulation chamber at the Karlsruhe Institute of Technology, Germany, we observed a strong increase in the ice-active fraction for experiments performed at temperatures just below the homogeneous freezing of pure water."*

Page 4, Sect. 2.5: the adsorption and desorption isotherms should be given as supplementary information.

*We have now provided the adsorption and desorption isotherms for the five CFA samples in the Supplementary Information document.*

Page 5, line 26: is the end of the freezing experiment the end of the expansion?

*For our experiments described in this manuscript - yes.*

Page 5, line 29: according to Sect. 2.3 the particles should be < 2.5 μm and not just < ~10 μm? Can you comment on this?

*The size classification of the scattering signals from the OPCs relies on certain assumptions for particle shape/orientation and refractive index. The size scale shown in this article relies on Mie calculations for a particle refractive index of 1.33. It is therefore directly applicable only to e.g. spherical water droplets that form during the expansion cooling runs. For spherical particles with a different refractive index and/or aspherical particles, the derived "optical" diameters do not agree with the "true" particle sizes measured with the SMPS and APS instruments. Concerning the CFA particles, their slightly aspherical particle habits and much larger refractive index of about 1.6* (Jewell and Rathbone, 2009) *lead to a significant overestimation of their "true" diameters on a size scale calibrated for spherical particles with a refractive index of 1.33. Therefore, a small fraction of the CFA particles is classified at apparent diameters above the minimum cut-off size of our cyclones ($D_{50} = 2.3$ μm).*

*We have added the following clarification to P5L29: "Note that the size scale of the OPCs was calibrated for spherical particles with a refractive index of 1.33. The slightly aspherical shape and much larger refractive index of the CFA particles* (Jewell and Rathbone, 2009) *lead to a significant overestimation of their true diameters on this size scale. Therefore, some CFA particles are detected at apparent diameters above the minimum cut-off size of our cyclones ($D_{50} = 2.3$ μm)."*

Page 7, line 2: do you mean "at" instead of "from"?

*We have corrected this to "at".*

Page 7, lines 2 – 5: It is insinuated here that the generation method (dry vs. wet) might influence the ice nucleation activity of CFA particles. However, for a consistent comparison, the available INP area also needs to be taken into account. In Umo et al. (2015), the ice nucleation active site density does not rise above $10^5$ cm$^2$. The surface area of a spherical 1 μm radius particle is only $10^{-7}$ cm$^2$. Therefore, if the CFA particles had the same ice nucleation ability as reported in Umo et al. (2015), only a minor fraction of the particles should be active, which is indeed in accordance with the AIDA experiments.

*Yes, we agree. Also, the droplet freezing assay technique employed in Umo et al. (2015) is sensitive to INP at warmer temperatures compared to AIDA experiments, given that there is a higher chance to observe rare freezing events at warm temperatures (with small values for the ice nucleation active site density) due to the larger size of the droplets. We have extended our discussion on this issue in the revised text on P7L3-4:*

*"Both studies, however, were performed with drop freezing assay techniques and with much larger particles than reported here. Hence, the probability of observing freezing events at much warmer temperatures was higher than in the AIDA experiments where smaller particle sizes were explored. A combination of both techniques in future studies could ultimately yield a parameterization of the heterogeneous ice nucleation activity of the CFA particles over the entire range of temperatures in the mixed-phase cloud regime."*

Page 7, line 5: do you mean "at" instead of "from"?

*Yes, changed to "at".*

Page 7, lines 12 – 13: "In this work, the average median particle diameter was 0.58 μm for our CFA samples theirs was size-selected to 0.3 μm." Improve formulation.

*This sentence now reads: "The average median particle diameter of our CFA samples is 0.58 μm, whereas Grawe et al. (2016) reported an average diameter of 0.3 μm."*

Page 7, line 20: While the ice nucleation activity of the CFA particles investigated here is compared to the ones of other studies and other aerosol types, a comparison of the CFA particles investigated in this work among each other is lacking. Figure 7 shows that the ice nucleation activity of CFA_JA and CFA_Wh is one order of magnitude larger than the one of CFA_Cy and CFA_Mi. Are there differences in morphology, elemental composition, or surface functionalization that might explain the differences?

*Yes, it is true that there is a huge variation in the inherent heterogeneous ice nucleation activity amongst the various CFA particle types. To the best of our knowledge, it is not yet clear and has to be clarified in future investigations which one of the governing factors you mentioned (morphology, elemental composition, surface functionalization) is the key parameter in influencing the particles' ice nucleation abilities.*

*We have included the following short paragraph to summarize and facilitate the comparison of the ice nucleation activity of the CFA particles investigated in our study: "In order to compare the inherent ice nucleation behaviour of the five CFA samples investigated, we have tabulated the maximum ice-activated fraction (%) for experiments with a similar starting temperature of about 250 K (Table 2, experiment numbers 3 & 5 - 8). The results reveal a significant spread in the ice-activated fractions, with CFA_Wh (~ 26 %) > CFA_Ja (~ 17 %) >> CFA_Cy (~ 1.5 %) = CFA_Mi (~1.5 %) > CFA_UK (~ 0.17 %). This huge variation in the particles' inherent ice nucleation activity is probably related to differences in morphology, elemental composition, and/or surface functionalization."*

Page 7, lines 31 – 33: "This occurred at a lower $RH_{ice}$ = ~105 % than the experiment with unprocessed CFA_UK particles which $RH_{ice}$ = ~130 % (corresponding to water saturation)." Improve formulation.

*This sentence now reads: "The processed CFA_UK particles nucleated ice at water-subsaturated conditions with a nucleation threshold in terms of $RH_{ice}$ of only about 101 %. In contrast, the unprocessed CFA_UK particles nucleated ice in the immersion freezing mode after exceeding water saturation during the expansion run (corresponding to $RH_{ice}$ ~ 130 %)."*

Page 8, lines 7 – 8: Was this the third expansion of the same sample or an expansion with a new sample? Please clarify.

*The expansion was performed with the same sample. The sentence now reads: "Afterwards, the same processed CFA_UK aerosol particles were warmed to $T_{start}$ = 264 K for another expansion cooling run (Fig. 4C)."*

Page 8: lines 33 – 35: "At $T_{start}$ = 253 K, the $f_{ice}$ for CFA_Cy particles after the pre-activation process was ~0.86 % slightly lower than what was observed for the unprocessed CFA_Cy particles." Improve formulation.

*The statement has been removed in the revised version.*

Page 9, lines 2 and 3: "$T_{start}$ = 255 K (Fig. 7)": Where is this starting temperature shown in Fig. 7?

*This start temperature is not indicated in Fig. 7. It is now shown in Fig. S2 in the Supplementary Information document. The statement has been removed in the revised version.*

Page 9, line 10: "256 K start temperature (Fig. 7)": Where is this starting temperature shown in Fig. 7?

*This start temperature is not indicated in Fig. 7. It is now shown in Fig. S3 in the Supplementary Information document. We have completely revised the discussions in this paragraph (please see the response to Page 9, lines 16 – 17 comments.*

Page 9, lines 10 – 12: "Again, for the processed CFA_Ja particles, no appreciable enhancement of its ice formation abilities was observed as the $f_{ice}$ at $T_{start}$ = 249 K was 2 % at $RH_{ice}$ = ~125 %." Do you mean "enhancement compared to the unprocessed CFA_Ja?" Yet, Fig. 7 shows a decrease in $f_{ice}$ rather than "no appreciable enhancement". Please clarify.

*Yes, we were referring to the enhancement compared to the unprocessed CFA_Ja particles. Now, we have completely revised the discussions in this paragraph (please see the response to Page 9, lines 16 – 17 comments).*

Page 9, lines 15 – 16: "We cannot completely rule out that the actual formation mechanism in both scenarios after the temperature cycling is not via a condensational freezing pathway." Formulate clearer, avoid double negative.

*Yes, we have completely re-phrased the paragraph describing the experiments with the CFA_Ja particles. This is described in our answer to the following comment).*

Page 9, lines 16 – 17: "This was not seen for the unprocessed CFA_Ja particles; after reaching water saturation, there was a time lag before ice particles were detected." Is this a valid comparison? According to Fig. 7, the unprocessed CFA_Ja sample had a higher starting temperature. The onset of $f_{ice}$ was therefore still at a warmer temperature for processed compared with unprocessed particles. This difference should be taken into account when discussing the effect of processing.

*This is correct – we will clarify our discussion in the revised manuscript text. At starting temperatures around 250 K, there was indeed not much change in the ice nucleation ability of the CFA_Ja particles before and after the TCF cycle (when taking into account the slightly different starting temperatures for the processed and unprocessed particle ensembles). Obviously, pre-activation cannot compete with the already very high inherent heterogeneous ice nucleation ability of the CFA_Ja particles at this temperature, meaning that there is no further detectable increase in the ice-activated fraction. The pre-activation phenomenon is then only visible when further warming the pre-activated CFA_Ja particles to a higher starting temperature (256 K, now included in the Supplementary Information document, Fig. S3, panel C). Here, the processed CFA_Ja particles showed a small nucleation mode with $f_{ice}$ ~ 1 % at 252 K just when exceeding water saturation during the expansion run. Given that the threshold temperature for exceeding an ice-activated fraction of 1 % for the unprocessed CFA_Ja particles is as low as 246 K, the observed ice nucleation mode for the processed CFA_Ja particles at 252 K can most likely be ascribed to the condensational growth of pre-existing ice, generated in the pores of the particles during the TCF cycle.*

*The revised paragraph will read as follows:*

*"Pre-activated CFA_Ja particles did not show any significant improvement of their ice nucleation ability after the temperature-cycling experiment for expansion cooling experiments started at around 250 K (Fig. 7). Obviously, pre-activation cannot compete with the already very high inherent heterogeneous ice nucleation ability of the CFA_Ja particles at this temperature, meaning that there is no further detectable increase in the ice-activated fraction after the TCF cycle. However, the pre-activation phenomenon becomes visible when further warming the pre-activated CFA_Ja particles to a higher starting temperature (256 K, Fig. S3, panel C). Here, the processed CFA_Ja particles showed a small nucleation mode with $f_{ice}$ ~ 1 % at 252 K just when exceeding water saturation during the expansion run. Given that the threshold temperature for exceeding an ice-activated fraction of 1 % for the unprocessed CFA_Ja particles was as low as 246 K, the observed ice nucleation mode for the processed CFA_Ja particles at 252 K can most likely be ascribed to the condensational growth of pre-existing ice, generated in the pores of the particles during the TCF cycle."*

*Based on our re-phrasing of the pre-activation experiments with CFA_Ja particles as discussed above, we suggest shortening the succeeding discussion of the CFA_Wh particles because their pre-activation behaviour was rather similar: We do not see any significant effect of the TCF cycle at temperatures where the particles' inherent heterogeneous ice nucleation ability is already very high. Only when further warming the pre-activated particle ensemble, a smaller ice nucleation mode, probably due to the condensational ice growth mode, becomes visible. Re-phrasing addresses the following three comments; the revised text reads as follows:*

*"Similar to the CFA_Ja particles, also the CFA_Wh particles did not significantly change their ice nucleation ability after the TCF cycle when probing them at starting temperatures of 248 K - 249 K (Fig. 7), i.e., in a temperature range where the particles' inherent heterogeneous ice nucleation ability is already very high. The smaller nucleation mode with $f_{ice}$ ~ 2 % that was observed after further warming the processed CFA_Wh particles to 256 K (Fig. S4, panel C), however, is likely again due to the condensational ice growth mode. The CFA_Mi particles showed the smallest variation with respect to their ice nucleation ability after the TCF cycle. In addition to the comparable ice nucleation behaviour before and after temperature cycling at a starting temperature around 250 K (Fig. 7; Fig. S2 panels A & B), the processed CFA_Mi particles also revealed only a tiny condensational ice growth mode at a higher starting temperature of 255 K with $f_{ice,max}$ ~ 0.1 % (Fig. S2, panel C)."*

Page 9, line 26: $T_{start}$ = 256 K is not shown in Fig. 7.

Page 9, line 28 – 29: "occurred in a shorter temperature step": please formulate better.

Page 9, line 30: "pre-activation by PCF may not be very important compared to other particles that are less ice-active". Improve formulation.

Page 9, line 33: "and the relative humidity which summary is given in Figs 6 & 7." Improve formulation.

*This part of the sentence now reads: "the relative humidity as summarized in Figs. 6 & 7."*

Page 10, lines 29 – 30: An additional factor that should be discussed here is the competition between pre-activation and immersion freezing. The ice nucleation mode is given in Table 2 but this table is neither connected with the text nor with the figures. Figure 7 needs to be improved to clearly state when water saturation is reached.

*This is true, in the revised paragraphs describing the pre-activation experiments with the CFA_Ja and CFA_Wh particles (see above), we have added some discussion on the competition between pre-activation and immersion freezing.*

*Additionally, we have now better linked Table 2 to the appropriate text and figures.*

*For all the data shown in Fig. 7, ice particles were detected only after water saturation was exceeded. Detailed AIDA data of these expansion runs were added to the Supplementary Information document (Figs. S2, S3, and S4). We have also symbolized the temperatures where water saturation was reached during the expansion experiments by vertical, dotted lines in Fig. 7.*

Page 10, lines 35 – 36: "Depending on the transport of CFA particles in the atmosphere, they can pass through different altitudes and temperature regimes which can naturally provide a temperature-cycling and freezing process for these particles to be pre-activated." Improve formulation.

*This statement was changed to: "In the atmosphere, CFA particles can be transported through different relative humidity and temperature regimes. This can provide a natural temperature-cycling and freezing process and lead to the pre-activation of these particles."*

Page 11, line 8: "There is a need by the modelling community to study the impact that…". Improve formulation.

*This sentence part now reads: "We suggest that future modeling work should focus on the impact that …"*

Page 11, line 26: This is the first and only time that the chemical compositions of CFA particles is mentioned in this manuscript. Indeed, the chemical composition might be relevant to explain the differences in immersion freezing of the different CFA samples. If chemical composition is mentioned in the conclusions, it should also be discussed in the section "Results and Discussion".

*Absolutely, we did not show any results on the chemical compositions of the CFA samples that were explored in this study. Although Garimella (2016) gave information on the different classes of the USA CFA samples which were based on chemical composition but the details of such compositions were not reported. Referring to P11L26, we have removed chemical compositions from that line.*

*We have now added the statements below to Section 3.1.*

*"The observed differences in their inherent ice-nucleating abilities may also be due to variabilities in their chemical and mineralogical compositions. Garimella (2016) reported that the four CFA samples from the USA belonged to different classes of fly ash and these groupings are based on the chemical compositions (Garimella, 2016)."*

Page 18, caption of Table 1: How was the median diameter determined?

*We added: "The median diameter was determined from the combined data of the APS and the SMPS instruments."*

Page 18, Table 2: Consider to add $f_{ice}$ and the freezing onset temperature to Table 2.

*$f_{\text{ice,max}}$, $RH_{ice}@ f_{\text{ice,max}}$, and $T_{\text{ice onset}}$ have been added in Table 2.*

Figure 3: the tags on the y-axes should be increased for better visibility. The measurements shown in this figure should be related to the experiments listed in Table 2.

*We have increased the labels on the y-axes and have related this figure to the experiments listed in Table 2. This correction has been applied to all similar figures.*

Figure 6, figure caption: It should be made clear whether the start temperatures of the experiments are shown in this figure.

*We have added the following statement to the figure caption: "The temperatures referenced on the x-axis are the temperature at which the maximum ice-activated fraction was reached during each experiment."*

Figure 7: It would be helpful to indicate the temperature where water saturation is reached for all experiments. Consider to add experiments with CFA_UK for better comparison.

*We have now indicated the temperature where water saturation occurred in the experiments.*

**Technical comment:**

Page 4, line 10: "range" might be more adequate than "limit" since a range is given in brackets.

*Changed to "range".*

Page 4, line 28: "microscope" instead of "microscopy".

*Changed to "microscope".*

Page 5, line 31: "β" does not appear correctly in the pdf.

*Sorted.*

Page 5, line 33: "γ" does not appear correctly in the pdf.

*Sorted.*

Page 8, line38: "their" instead of "its".

*This line has been removed in the revised version.*

Page 9, line 29: remove "an".

*Done.*

Page 10, line 14: "than those of other CFA particles" instead of "than the other CFA particles".

*Corrected.*

Page 12, line 3: "need" instead of "needs".

*Corrected.*

Page 14, line 26: John is the first name of J. G. Morris. Please revise reference.

*Done.*

**Review of "Enhanced ice nucleation activity of coal fly ash aerosol particles initiated by ice-filled pores" by N. S. Umo et al. for Atmospheric Chemistry and Physics**

General comments:

In the present study, N. S. Umo and co-workers show and discuss the results of temperature cycling experiments with coal fly ash (CFA) particles at the AIDA cloud chamber. The aim of these experiments was to clarify whether the ice nucleation activity of CFA is increased by the pore condensation and freezing (PCF) mechanism under certain conditions. The authors achieve to convincingly demonstrate that this is the case for some of the used samples. The question of why some samples are more prone to PCF than others is not convincingly answered, but this cannot be expected in the case of CFA, which is a very complex and heterogeneous substance. From my point of view, the study is an interesting addition to recent findings concerning the ice nucleation behavior of CFA particles (Umo et al., 2015; Grawe et al., 2016; Garimella, 2016; Grawe et al., 2018; Losey et al., 2018). However, there are some content-related issues which need more discussion or clarification from my point of view. These are listed below (specific comments). Parts of the manuscript could benefit from editing with respect to wording and presentation of the results but this can be easily resolved (see technical corrections). The figures are mostly clear, but I was wondering why a significant part of the results are not shown anywhere. The authors should consider including an Appendix for presenting the missing figures. Generally, I feel that the topic fits the scope of ACP and that the study is worth publishing. To improve the overall significance and readability of the manuscript, I suggest the following minor points.

*Thank you for your kind interest in reviewing our work. We appreciate the painstaking efforts in giving this manuscript a thorough review, and providing us with very useful comments, suggestions, and corrections, which will greatly improve our work. We totally agree with your suggestion to include an Appendix for showing missing plots. We have now included a Supplementary Information document to the revised manuscript. We have worked through your comments and responded to each point accordingly.*

Specific comments:

1) Effect of size-dependent specific surface area and pore properties:

The investigation of PCF on CFA particles is a daunting task, since CFA is such a heterogeneous substance. I appreciate the authors' attempt at finding possible reasons for differences in the behavior of the different samples, but I also think that their approach lacks the discussion of an important point, i.e., the size-dependence of the particles' specific surface area, pore volume, and pore size. This type of information is probably very hard to come by and I do not expect the authors to perform further analyses. But it should at least be mentioned that the specific surface area and pore volume are very likely dependent of the particle size and that some particles might not even feature pores (Seames, 2003). Hence, the properties of the sieved bulk sample (0-20 µm) might not be representative for the properties of the particles that actually enter the AIDA chamber (< 2.5 µm). This should be made clear in the discussion (P9L37-P10L30).

*We completely agree that making specific surface area (SSA), pore volume and pore size measurements of size-selected CFA particles would be a great step forward for this type of investigation. But as mentioned, CFA particles are very complex and heterogeneous in*

*nature. We thought about making such measurements but it is highly difficult to collect a sufficient amount of aerosolized CFA particles in the required size range (i.e. < 3 µm) for BET measurements; and we do not have access to an appropriate size-selection instrument. There is no doubt that in some cases one can see a dependence of SSA, pore size, and pore volume on size e.g. (Schure et al., 1985). Also, we have mentioned the occurrence of cenospheres and plerosphere in CFA particles which makes it even more difficult to attribute certain properties to the bulk material. Most probably, there are cenospheres and plerospheres in our samples but we cannot estimate the percentage of particles in our sample which show these phenomena. It should also be noted that this can also change depending on the handling of such samples.*

*As suggested, we have now included a statement that reads: "In previous studies, it has been shown that the specific surface area and pore volume of fly ash particles generated from pulverized coal combustion are very likely dependent on the particle size (Schure et al., 1985; Seames, 2003)."*

2) Comparison to Wagner et al. (2016)

When viewing the results, I was wondering how CFA particles compare to other the substances which have already been investigated with the same instrumental setup and measurement routine by Wagner et al. (2016), i.e., zeolite, diatomaceous earth, mineral dust, volcanic ash, and soot. Concerning CFA_UK, it is mentioned very briefly (P6L19-20) that "a similar increase in the heterogeneous ice nucleation ability has been previously observed for zeolite and illite" but nothing is said about the other CFA samples and other substances tested by Wagner et al. (2016). This comparison could be expanded to put the CFA results into perspective.

*Thank you for pointing this out. We have now included a new Section (Section 3.3) to compare the ice nucleation enhancement by CFA particles reported in this study with other particles previously studied with a similar measurement routine at the AIDA chamber. The text of the new Section is presented below.*

*3.3 Ice nucleation enhancement by CFA particles versus other particle types*

*In a previous study, Wagner et al. (2016) investigated the pre-activation behaviour of INPs by the PCF mechanism in the AIDA cloud chamber with a similar measurement routine as described in Section 2.6. In this study, a wide range of INPs was tested including illite NX, diatomaceous earth, zeolites, dust samples from Canary Island, Sahara and Israel, Graphite Spark Generator soot (GSG soot), and volcanic ash (Wagner et al., 2016). It was reported that illite NX, diatomaceous earth, and mesoporous zeolite CBV 400 showed a significant ice nucleation enhancement in the depositional ice growth mode, with ice-active fractions of 5.9 %, 3.8 %, and 3.7 % at a starting temperature of ~ 250 K (Fig. 8). At higher starting temperatures, the ice-activated fractions in the condensational ice growth mode were typically around 1 %. Another group of INPs such as CBV 100 (untreated microporous zeolites), Canary Island dust, and GSG soot showed much smaller depositional ice growth modes with ice-activated fractions below 1 %. Finally, volcanic ash, water-processed GSG soot, as well as Saharan and Israeli dust particles did not show any enhancement after the pre-activation process, neither in the depositional nor the condensational ice growth mode.*

*In this context, the ice nucleation enhancement observed for the CFA_UK particles at a starting temperature of 250 K in the depositional growth mode with $f_{ice,max}$ ~ 11 % (Fig. 4A) is by far the highest value for any particle type investigated so far (Fig. 8). In contrast, the pre-*

*activation efficiency of the CFA particles from the US power plants is comparable in magnitude to the above-mentioned group of CBV100, Canary Island dust, and GSG soot particles with much lower ice-activated fractions. The mean diameters of the particles investigated by Wagner et al. (2016) ranged from 0.21 μm to 0.43 μm, and were thus smaller than the mean diameters of our CFA particles except for CFA_Mi (0.42 μm). Different pore sizes, morphology, and chemical composition of these INPs may control their susceptibility to the PCF pre-activation mechanism. More studies are required to investigate the role that each of these parameters play.*

3) Atmospheric implications

Although the authors discuss the atmospheric implications of their findings, they are missing one major point. How large is the probability that CFA particles reach such high altitudes where they experience 228 K? I understand that CFA particles can influence atmospheric ice nucleation close to the point of emission, but the number concentration of these particles at cirrus level is probably close to zero. Indeed, it is difficult to identify CFA particles in the atmosphere due to their similarities to mineral dust which is why there is a lack of information concerning atmospheric number concentrations of CFA particles. But despite this lack of information, the authors should not leave this issue completely unattended. A remark concerning this should be included in Sec. 3.3.

*There are a couple of studies that have shown fly ash as a composition of ice residues in cirrus clouds and mixed-phase clouds e.g.* (DeMott et al., 2003; Liu et al., 2018). *With these pieces of evidence, we can indeed assume that CFA particles can reach higher altitudes with temperatures down to 228 K. We agree with you that there is still a lack of information on the actual number concentration of CFA particles in the atmosphere. As already suggested in Umo et al. (2015), given that the compositions of CFA particles are similar to typical mineral dust particles, one could speculate that some ice residue measurements are wrongly attributed to mineral dust instead of CFA. At the moment, this is still uncertain and there is an urgent need for in-depth source apportionment studies. We assume that in some areas, where these particles are emitted and transported, their concentration may be higher than other INPs, but at the moment we cannot quantify it.*

*We have now included the following statement in the text (Section 3.4): "Despite the dearth of information on the number concentration of CFA particles in the atmosphere at higher altitudes, there are pieces of evidence that CFA particles are found in ice residues of cirrus and mixed-phase clouds (DeMott et al., 2003; Liu et al., 2018)."*

4) Explanation of the PCF mechanism

- Even though PCF has become an accepted concept in recent years, the process itself should be explained in more detail. The negative Kelvin effect, which is the reason why there is capillary condensation of water vapor at a relative humidity (RH) below water saturation, is not even mentioned. I suggest to include an explanation of the mechanism (capillary condensation of water vapor at RH below water saturation → formation of ice in pores at very low temperatures → pore ice persists as site for ice nucleation at warmer temperatures and ice-supersaturated conditions) where it is first mentioned (P2L14).

*We have added a brief explanation of the PCF mechanism to the introduction Section of the manuscript as suggested. The explanation reads: "PCF involves a two-step process – first, capillary condensation of liquid water in the particle pores, and second, freezing of the condensed water. The first step occurs when particles with pores are exposed to a certain*

*relative humidity (RH$_w$) below water saturation (RH$_w$ < 100 %). The RH$_w$ for pore filling to occur is well-described by the 'negative' Kelvin effect* (Fisher et al., 1981). *The negative exponential term of the Kelvin equation accounts for the concave meniscus of the condensed water in a pore* (Sjogren et al., 2007). *When pores with condensed water (step 1) are exposed to sufficiently low temperatures, ice can form in such pores. Ice-filled particle pores can then act as active sites for ice nucleation and growth in an ice-supersaturated environment. In a situation where ice-filled pores (step 2) are preserved even when the system is warmed, they can trigger ice nucleation at warmer temperatures. This process is relevant for understanding ice nucleation by porous particles or particles with surface defects."*

- At this point, it should also be mentioned, why PCF is restricted to certain pore sizes (P2L29). Furthermore, I expected a remark concerning the effect of the pore geometry (cylindrical pore, ink-bottle-shaped pore) on PCF. Which types of pores might be present in CFA? All of this needs to be explained in the introduction.

*We have now added the information below to the introduction.*

*"The PCF mechanism is restricted to a certain pore size range due to limitations related to the negative Kelvin effect for water condensation in the pores and the size of the critical ice embryo for ice nucleation and melting. According to classical nucleation theory, a certain critical ice embryo size is required to overcome the energy barrier defined by the Gibbs free energy* (Pruppacher and Klett, 2010). *Therefore, the pore size should be large enough to accommodate such a critical ice embryo and small enough to enable the capillary condensation of water in the first place. Calculations and previous reports have shown that pore sizes with 3 – 8 nm diameter are suitable for the PCF mechanism* (Wagner et al, 2016, Marcolli, 2017). *Also, pore geometry (e.g., cylindrical or ink-bottle-shaped pores) has been shown to be an important parameter for the initial step of the PCF mechanism* (Marcolli, 2014, 2017). *Moreover, the contact angle between the pore wall and the water curvature affects the onset of the capillary condensation of water according to the Kelvin equation."*

*Unfortunately, we do not have any information on the pore geometries of our CFA particles. Considering the uniqueness, heterogeneity, and complexity of the CFA particles, it is even more challenging using some models to predict it.*

- Instead of Fig. 2, the authors could describe the temperature cycling process in a similar manner as Marcolli (2017; see Fig. 1-4), i.e., showing RH with respect to temperature. The time scales of the different steps could be mentioned in the caption.

*We have looked at the illustrations presented in Marcolli (2017) which show the RH$_{ice}$ trajectories as a function of temperature for different assumed pore types. These graphs are indeed a very good representation of typical atmospheric trajectories for well-known pore geometries. But note that the trajectories of our temperature-cycling experiments, however, would just be a horizontal line in such graphs at RH$_{ice}$ close to 100 %, given that the ice-coating on the inner chamber walls controlled RH$_{ice}$ to an almost constant value close to saturation throughout the entire TCF cycle. The illustration that we presented in Figure 2 was meant to show the procedure of our experiments and not to serve as a model. We specifically put out Figure 2 to show the experimental routine that we performed in AIDA as a simple guide for other researchers who may be interested in repeating the type of experiments that we have reported here. The plot is not meant to present an idealized conceptualization of the PCF mechanism with respect to CFA particles. Based on these reasons, we would prefer to leave it that way.*

5) Methodology

- The AIDA chamber is a well-established instrument but the authors should consider describing the measurement and data evaluation techniques in more detail. For example, there is no mention of the uncertainty of the $f_{\text{ice}}$ determination. How does the large error of ±20 % of the ice particle number concentration affect the $f_{\text{ice}}$ error?

*We added this information is Section 3 in the manuscript but based on the suggestion below, it has now been moved to Section 2.6 in the revised manuscript.*

- The explanation of how $f_{\text{ice}}$ is calculated should be included in Sec. 2.6, not in Sec. 3.

*We have moved this part to Section 2.6 and also added that the uncertainty estimation is ~ ± 20 %* (Möhler et al., 2006).

- I do not understand how droplets can form in an environment which is slightly subsaturated with respect to liquid water (see Fig. 3, panels A and B). It looks like the black line (RH with respect to water) is below 100 % throughout the duration of the experiments. Is this due to the measurement uncertainty? Please clarify.

*Yes, this is due to measurement uncertainty which we report as ~ ± 5 %. We mentioned this in Section 2.2.*

- P5L17-18: How can the air in the AIDA chamber be subsaturated with respect to ice when there is an ice layer on the inner chamber walls? Shouldn't it be saturated? Please explain.

*The air was indeed a few percent below ice-saturation. The reason is that we usually observe that the gas temperature in the AIDA chamber is a few tenths of a Kelvin warmer than the wall temperature. This is probably due to some internal heat sources in the interior of the chamber like e.g. heated sampling lines. We have now added the statement below to Section 2.6. "The slight sub-saturation of the chamber air with respect to ice may be attributed to some internal heat sources which increased the gas temperature by a few tenths of a Kelvin compared to the wall temperature (Wagner et al., 2016)."*

6) Presentation of the results

- I was wondering why ice nucleation surface site densities are not included in the discussion of the results. The authors state that "Further analyses on the distribution of the ice nucleation active sites densities of these CFA particles is outside the scope of the current report and will be presented in a separate communication." (P6L32-34). An inclusion of these data in the current paper would make more sense to me, especially for a comparison of the intrinsic ice nucleation behavior of the CFA samples to the results other studies (P6L35-P7L21). Besides, which new information would this other report contain?

*The ultimate goal of the here presented work is to report the ice nucleation enhancement by CFA particles after a pre-activation by PCF. We understand the need to discuss the INAS concept but that is not the focus of this work. But as suggested also by Referee #1, we have added the following paragraph to compare the inherent IN behaviour of the CFA samples.*

*"In order to compare the inherent ice nucleation behaviour of the five CFA samples investigated, we have tabulated the maximum ice-activated fraction (%) for experiments with a similar starting temperature of about 250 K (Table 2, experiment numbers 3 & 5 - 8). The results reveal a significant spread in the ice-activated fractions, with CFA_Wh (~ 26 %) >*

*CFA_Ja (~ 17 %) >> CFA_Cy (~ 1.5 %) = CFA_Mi (~1.5 %) > CFA_UK (~ 0.17 %). This huge variation in the particles' inherent ice nucleation activity is probably related to differences in morphology, elemental composition, and/or surface functionalization."*

*Regarding the new information in the other report, we have suggested a combination of the drop freezing assay technique with AIDA expansion cooling runs to explore the ice nucleation behaviour of the CFA particles over an extended temperature range and have added the statement:*

*"A combination of both techniques in future studies could ultimately yield a parameterization of the heterogeneous ice nucleation activity of the CFA particles over the entire range of temperatures in the mixed-phase cloud regime."*

- The mentioning of $T_{start}$ instead of the actual temperature at which $f_{ice,max}$ values were derived is not intuitive (see P6L27-28, Sec. 3.2, …). Please include both $T_{start}$ and the actual $T$ in your discussion.

*The actual temperatures at the $f_{ice,max}$ are now included in the discussion and also used in the summary plots.*

- As stated above, parts of Sec. 3.2 are hard to follow. In the cases of CFA_Mi, CFA_Ja, and CFA_Wh, the authors describe specific features observed in the experiments, but they do not show the corresponding 3-panel plots. Some results are not even included in Fig. 7. I suggest to include the described examples in an Appendix so that the reader can follow the discussion.

*To make this Section easier to follow and also based on the concerns of Reviewer #1, we have completely revised this Section to provide a more succinct summary. We have also provided a Supplementary Information document containing the missing results.*

- I find the amount of arrows, overlapping shapes, and different colors in Fig. 8 very confusing. What is the difference between the blue, red, light gray and dark gray arrows? What is the difference between the black, light gray, and dark gray particles? A legend, or at least an explanation in the caption, would be helpful. Also, it looks like the sublimation of ice particles directly leads to cloud formation. The authors should revise and thin out this figure to make it more intuitively understandable.

*Figure 8 (now Fig. 9) has now been revised and a legend has been included with more explanation in the caption.*

- Why is the CFA_UK sample discussed separately from the U.S. American samples? Fig. 6 and 7 should be combined. I also suggest to change from a bar graph to a scatter plot for more clarity. A change to a logarithmic y-axis should be considered for both Fig. 6 and 7.

*The pre-activation efficiency of the CFA_UK was really "outstanding" and we started with its description. Whereas the pre-activation behaviour of the USA samples was much different from the CFA_UK sample but rather similar among the various USA samples. Notwithstanding, we have provided a ranking to compare all the CFA samples in Sections 3.1 and 3.2. Consequently, we prefer to keep Figures 6 (now Fig. 5) and 7 separately but we have changed Figure 7 to a scatter plot as suggested. The ordinates of both plots are now on a logarithmic scale. The reason is to clearly illustrate the changes in the CFA_UK after the pre-activation process.*

Technical corrections:

I generally feel that the authors need to be more precise with their formulations. Sentences and paragraphs are sometimes lengthy due to unnecessary fillers and repetitions. Transition words are partly misleading. There are grammatical errors. Below, I list the issues that caught my eye but I advise the authors to recheck their manuscript carefully to improve readability and understandability.

*Thank you for your comments. In response, we have carefully rechecked the manuscript and improved our sentence formulations. In addition, Section 3.2 was completely revised and shortened.*

P1L26          At which RH was this strong increase observed?

*This strong increase was observed at $RH_{ice} = 101 – 105$ %. This information has been added to the sentence and now reads "…we observed a strong increase (at a threshold relative humidity with respect to ice of 101 – 105 %) in the ice-active fraction for experiments performed at temperatures just below the homogeneous freezing of pure water."*

P1L27          Change to either "undergoing PCF" or "undergoing **the** PCF mechanism".

*Changed.*

P1L31-35     This sentence would benefit from being split into two.

*Done.*

P1L36-39     It is a bit unfortunate that you refer to PCF in general in the first sentence and specifically to PCF on CFA particles in the second. At least this is how I understood it. Please reword.

*We have switched the order of importance communicated in this sentence. It now reads: "On the one hand, the PCF mechanism can play a significant role in mixed-phase cloud formation in a case where the CFA particles are injected from higher altitudes and then transported to lower altitudes after being exposed to lower temperatures. On the other hand, the PCF mechanism could be the prevalent nucleation mode for ice formation at cirrus temperatures rather than the previously acclaimed deposition mode."*

P1L36          I suggest to introduce "intrinsic" as relating to unprocessed or not pre-activated particles.

*We have removed this word.*

P1L37          Change to "on the other **hand**".

*Changed.*

P1L41          Change to "highly relevant to **our understanding/knowledge/ comprehension… of** cloud formation".

*Changed to "highly relevant to our knowledge of cloud formation…".*

P2L3           Omit "primary". Heterogeneous ice nucleation is always a primary process. Also, define the terms "homogeneous" and "heterogeneous" before using them.

*We have omitted the word "primary". The terms "homogeneous" and "heterogeneous" are now defined in the text. We have also added Vali et al. (2015) here as a reference.*

P2L8          Change to "with **the surface of** a …".

*Changed.*

P2L11        Omit "however" at the start of the new paragraph.

*Omitted.*

P2L12-13     Change to either "such **a** particle" or "such particles".

*Changed to "such particles".*

P2L19        Change to "before ice nucleation **takes place**".

*Changed.*

P2L20        Change to "Here, we define…".

*Changed.*

P2L21        There are references to Wagner et al. (2016a), Wagner et al. (2016b), and Wagner et al. (2016). Yet, there's only one Wagner et al. paper from 2016 in the reference list. Please correct.

*Corrected.*

P2L27        "zeolite" and "illite" should not be capitalized.

*The two words are now in lowercase letters.*

P2L34-36     This sentence does not say anything else than the one on P2L16-18. I suggest to remove it (also the following sentence).

*We have removed the sentence and added the references to the statement on P2L16-18.*

P2L39        Change to "global **cloud** ice budget".

*Changed.*

P2L41        A "significant amount" is not very precise. Are there really no estimates of the emitted CFA mass?

*We would have liked to be more precise about the amount of CFA emitted directly into the atmosphere but unfortunately, we do not have that information.*

P3L4         There is no question that immersion freezing was investigated by Grawe et al. (2016, 2018) and Umo et al. (2015). Hence, differences in the freezing behavior of the investigated samples are not due to differences in the freezing mechanism. Please remove this part of the sentence. Furthermore, Grawe et al. (2018) showed that the immersion freezing behavior of CFA can be strongly dependent on the amount of time that the particles spend immersed in the

droplet prior to the initiation of freezing. This issue is worth mentioning here because it can also affect the immersion mode AIDA measurements.

*We have removed the part of the sentence that reads: "...; as well as variabilities in the actual freezing mechanisms, which could be influenced by surface defects or porosity of the particles." The statement now reads: "However, there are variabilities in the ice nucleation activities of the different CFA samples reported, which could be due to the difference in mineralogical compositions, and the extent to which these particles are processed in the atmosphere (Grawe et al., 2018; Losey et al., 2018)."*

P3L6        "various atmospheric conditions". Be more precise.

*We have rephrased this as: "...various temperature and relative humidity conditions."*

P3L7        Change to "different CFA **samples**". Also on P3L14.

*Changed.*

P3L15-17        "The results from these new laboratory measurements are presented in this report." This sentence is unnecessary and should be deleted.

*Deleted.*

P3L23-25        How representative is material from the EPs in comparison to material that is emitted into the atmosphere. Please include a statement concerning this matter.

*We have now included this statement in the text: "The CFA particles collected from EPs are the same particles that could have been directly released into the atmosphere in situations where EPs malfunction or are inefficient. Also, the CFA particles which are emitted indirectly into the atmosphere by road transportation, application in agricultural fields, industrial sites, road construction, and other sources are the same CFA particles as collected from the EPs (Buhre et al., 2005)."*

P3L25        Omit "However".

*Omitted.*

P3L25-29        I am aware that the authors do not focus on the effect of chemical composition of the samples on their ice nucleation behavior. However, it would be interesting to include some more information, e.g., are the samples of class C (high Ca) or class F (low Ca), since it has been shown that the composition affects the ice nucleation measurements. This information is easily obtainable from Garimella (2016) and should be mentioned here.

*We have included a few sentences to make reference to the CFA classification as follows: "Garimella (2016) grouped CFA_Ja and CFA_Wh fly ash samples as class C type, while CFA_Cy and CFA_Mi are class F which is broadly based on the calcium oxide (CaO) composition (Ahmaruzzaman, 2010). A new CFA standard classification system suggests that CFA samples can be sialic (S), calsialic (CS), ferrisialic (FS), and ferricalsialic (FCS) (Vassilev and Vassileva, 2007). However, no further information on chemical composition was provided by Garimella (2016) for a more quantitative classification of the USA CFA samples."*

P3L29         Please note that Losey et al. (2018) investigated the same sample set. This publication should be referenced here as well.

*Thank you for pointing us to the very interesting work by Losey et al. Losey et al. (2018) stated that the CFA samples which were used in their study are from the same four power plants in the USA that we also studied. They also obtained the CFA samples from Fly Ash Direct®. However, we would be careful to state that Losey and co-workers investigated the same sample set as we did in this work. This is because we do not have any evidence that they used the same batch of CFA samples that we also obtained. CFA samples from a power plant can differ due to many factors including the batch of the coal fuel burned, operating conditions, etc. We have therefore cited the Losey et al. work in the introduction but not at the particular place on P3L29.*

P3L30         Change "name" to "operators/owners". The name itself cannot prefer anonymity.

*Changed to "Operator".*

P3L31-33       Were the other samples not sieved? Why not?

*Yes, all raw CFA samples were sieved to the same diameter size fraction (0 – 20 μm). The sentence now reads: "First, all raw CFA samples were sieved with a Fritsch Sieve set-up (Analysette 3, 03.7020/06209, Germany) to obtain 0 – 20 μm diameter size fractions, which were later used for the experiments."*

P4L3         Please combine the instrument abbreviation and the manufacturer in one set of parentheses to avoid "(…) (…)". Check throughout the manuscript.

*Done.*

P4L20        Which types of cyclones were used? What is their cut-off diameter?

*We used custom-made cyclones that work in a similar way as cyclones 2 and 3 of a five-stage series cyclone (EPA, USA). Cyclone 2 ($D_{50}$ cut-off = 3.7 μm) and cyclone 3 ($D_{50}$ cut-off = 2.3 μm). We have added this sentence to Section 2.3: "Cyclone 2 ($D_{50}$ cut-off = 3.7 μm) was placed before cyclone 3 ($D_{50}$ cut-off = 2.3 μm) in the set-up.*

P4L26        Change "Min" to min".

*Changed.*

P4L27        Change "mins" to "min".

*Changed.*

P4L27        How does this coating affect the morphology of the particles? Could pores potentially be covered? Please include a short statement.

*The coating thickness was 1 nm and thus below the SEM resolution. Coating is a well-known and standard procedure used in scanning electron microscopy. To the best of our knowledge, we do not know any report of its potential influence on the morphology of scanned particles. Specifically, in our experiment we did not see any unusual occurrence on our samples; hence, we do not think that pores were masked by the coating process. We have added this sentence: "Coating of the filters did not affect the morphology of our samples because the coating thickness was 1 nm and thus below the SEM resolution."*

P4L36        "argon" and "nitrogen" should not be capitalized in running text.

*Corrected.*

P5L21        "inherent" and "intrinsic" seem to be used synonymously throughout the manuscript. I suggest to avoid the use of both terms and stick to one.

*We are now using "inherent".*

P5L31,33     Make sure that the empty squares are replaced by the Greek letters in the new version of the manuscript.

*Done.*

P5L35        Since you list all experiments in Table 2, you could also include the used size thresholds there.

*The average median diameters of the CFA particles for each of the samples investigated are already given in Table 1.*

P6L11        Change "$t =\sim 300$ s" to "$t \sim 300$ s". Also in all other occurring instances.

*Changed.*

P6L25        Insert "⁻" behind "CFA_Wh".

*We do not think a dash is required there. Instead, we have inserted a hyphen after CFA_Wh.*

P6L26        Insert "mode" behind "immersion freezing".

*Inserted.*

P6L35-

P7L21        I agree that a comparison to previous ice nucleation studies with CFA is interesting. However, by only reporting onset ice nucleation temperatures, the reader does not get an idea how the here investigated samples compare to those from previous studies. This could be resolved by including a figure showing $n_s(T)$.

*We have addressed this issue in our answer to your specific comment #6 "Presentation of the results" and outlined the changes made to the manuscript text.*

P7L3-4       "Both studies can access warmer freezing temperatures for INPs than the dry generation method that our system is designed for.". Please explain shortly why this is the case.

*We have modified this sentence to read "Both studies, however, were performed with drop freezing assay techniques and with much larger particles than reported here. Hence, the probability of observing freezing events at much warmer temperatures was higher than in the AIDA experiments where smaller particle sizes were explored. A combination of both techniques in future studies could ultimately yield a parameterization of the heterogeneous ice nucleation activity of the CFA particles over the entire range of temperatures in the mixed-phase cloud regime."*

P7L5         The Schnell et al. (1976) reference is not a good choice here. Actually, in this study "no detectable effects from a coal-fired powerplant plume" (see title) on atmospheric ice nucleation were found. Better cite Parungo et al. (1978), who

conducted a similar experiment and found an enhanced ice nucleation efficiency of the plume aerosol in comparison to the background aerosol.

*Thank you for pointing this out. We have corrected this reference to Parungo et al., 1978.*

P7L10        Which of the two studies are you referring to?

*We were referring to the Grawe et al (2018) study. The sentence now reads "Grawe et al. (2018) partly attributed the ice nucleation behaviour of…"*

P7L10        Actually, Grawe et al. (2018) state that the amount of hydratable components is important and that quartz only contributes in those samples which contain a small concentration of hydratable components. But this could also be included in the introduction (P3L4).

*We have rephrased this sentence and moved it to P3L4. The statement now reads "Grawe et al. (2018) partly attributed the ice nucleation behaviour of the CFA particles to the quartz composition of the CFA; however, the influence of quartz can be suppressed in a situation where hydratable components form a layer on the particle surface."*

P7L13        Please cite Garimella (2016) instead of Welti et al. (2009). Garimella (2016) investigated 300 and 700 nm CFA particles, not mineral dust, and found that the immersion freezing efficiency does not scale with the surface area as the smaller particles were relatively more efficient than the larger ones.

*Cited.*

P7L17        Please cite Grawe et al. (2018) instead of Hiranuma et al. (2018). Firstly, the manuscript by Hiranuma et al. (2018) is still under review. Secondly, the study by Grawe et al. (2018) is more relevant for the here presented work as they discuss the methodology-dependent freezing behavior of CFA particles, not cellulose.

*Cited.*

P8L11-13        Does this mean that 1.2 % of the particles contained pores suitable for PCF? If yes, then please say so.

*At this start temperature (264 K), the result indeed implies that at least 1.2 % of the particles are ice-active via the PCF mechanism. The fraction could even be higher because this was not the first expansion after the temperature-cycling process (TCF). We have added this statement to P8L13: "This implies that at least 1.3 % of the processed CFA_UK particles still contained ice-filled pores even after warming to 264 K."*

P8L14        Please check the Mahrt et al. (2018) reference. Mahrt et al. (2018) indeed describe PCF, but they did not conduct temperature cycling experiments. They saw a stepwise increase in the activated fraction of one type of soot particles due to condensation in pores and subsequent homogeneous freezing.

*We referenced Mahrt et al. (2018) because they described the condensational growth in soot particles, but they did not conduct temperature-cycling experiments. Due to the confusion that this reference might create here, we have removed it.*

P8L23        Figure 6 is discussed before Fig. 5. Please arrange the order of the attached figures accordingly.

*Figures 6 and 5 are now rearranged.*

P8L38-39     Please check this sentence for correctness. It does not relate to the previous statement and it seems that something is missing.

*During revisions, this sentence has been removed.*

P9L5-7     Change beginning of the sentence to "**This** suggests..".

*We have removed this sentence in the revised version of the manuscript.*

P9L7-8     "Confirm" is a very strong word here, given that the data of this example is not even shown and given that no error estimation for the $f_{ice}$ error is provided.

*We have removed this sentence in the revised version of the manuscript. All missing data are now available in the Supplementary Information document.*

P9L9     Change to "droplet activation". Also on P9L17.

*Changed.*

P9L16     Please omit "One thing is extremely clear that …".

*Omitted.*

P9L19-30     This paragraph would profit tremendously from the inclusion of a 3-panel-plot of the measurement that you are describing here.

*The 3-panel-plot is now available in the Supplementary Information document.*

P9L25     I do not understand the use of the word "although" here.

*We have removed the word "although".*

P9L28-29     "ice formation occurred in a shorter temperature step". I am not sure what you mean here. The temperature at which $f_{ice,max}$ was registered is of interest, not the temperature range. Please reword.

*We have reworded this part to read: "ice formation occurred at the instant of the expansion process for the processed CFA_Wh (Fig. 7 and Fig. S4C).*

P9L32 Change "comparison" to "ranking".

*Changed.*

P9L33     Change "$f_{ice}$" to "$f_{ice,max}$".

*Changed.*

P9L33     "which summary". Change to form a grammatically correct sentence.

*Changed. The sentence now reads: "The ranking is based on the start temperature, $f_{ice,max}$ and the relative humidity as summarized in Figs. 5 & 7.*

P10L21-23     Please reword this sentence.

*The sentence now reads "As another example, CFA_Cy (0.012 cm³ g⁻¹) has a PV similar to the CFA_Mi sample (0.013 cm³ g⁻¹), but only the processed CFA_Cy particles showed a clear pre-activation ability due to the PCF mechanism."*

P10L25      Is it realistic that cenospheres or plerospheres would be filled by capillary condensation under the conditions in your experiments? How large are the spheres and how large are the openings? According to Marcolli (2017), large pores need very high RH or very low temperatures to be filled.

*The behaviour strongly depends on the size of the spheres and openings. Unfortunately, we do not have the size information of the cenospheres or plerospheres for our samples and cannot further resolve this issue at the moment.*

P10L26      Fischer et al. (1976), who first discovered cenospheres and plerospheres, is the more appropriate reference here.

*We have added Fischer et al. (1976) to the existing references.*

P9L27-30      I suggest to change the formulation in such a way that it becomes clear that an estimation of the pore size and geometry is **not possible** for CFA. This is due to the heterogeneity of the particles. Other INP types might be better suited for such an estimation. Avoid using the word "pointless".

*We would like to avoid using the word "impossible" in our text. We have reformulated the sentence to read "Currently, it is highly difficult to estimate the pore sizes based on the PV of the CFA samples except in the case of a well-defined pore model and morphology."*

P10L37      "Rainout" is probably not the right term for cases where only the ice phase is involved.

*We have removed "rainout".*

P10L37-38      I cannot find this statement in the given reference. Please check and remove if necessary.

*Hong et al. (2004) is a reference supporting an atmospheric sedimentation process. We have moved it to the next sentence that lists the various atmospheric processes that could aid re-circulations of particles in the atmosphere.*

P11L3-4      This is shown in a very confusing way in Fig. 8. Please adjust.

*Figure 8 (now Fig. 9) has been adjusted.*

P11L12-13      "Currently, this is not well understood and requires further research." This sentence can be omitted. The following sentence is completely sufficient.

*Removed.*

P11L14      Change "clouds formation" to "cloud formation".

*Changed.*

P11L15      Change "studying the dominance of this occurrence" to simply "the occurrence".

*Changed.*

P11L16      Change "cloud system" to "cloud system**s**".

*Changed.*

P11L21      "However" does not seem like an appropriate transition word. The statement is not in contrast to the previous one. Please reword.

*We have changed "however" to "Also".*

P11L22-23      I am aware that there are lots of different pore types ("pores", "crevices", "cavities"), but it might be best to define one term in the beginning and stick to it throughout the manuscript.

*We have changed "cavities" to "pores". We have also included a definition of our pores in Section 1.*

P11L25      Change "ice formation" to "ice nucleation potential".

*Changed.*

P11L33      What is meant by "their"? The particles? The pores? Please clarify.

*We have clarified this by modifying part of the sentence to read "the particle's overall ice-nucleating efficiencies".*

P11L35-36      It should be mentioned here that CFA is not a suitable substance for the investigation of the effect of different pore geometries on PCF.

*We would prefer not to make this statement at this point because CFA particles could still be used for the investigation of the effect of pore geometries on the PCF mechanism in the future. Currently, we just do not know the prevalent geometry of the pores or the actual pore size of these particles. If future technology permits such properties to be well characterized, we think it could be a suitable material. Also, in the second part of the conclusion Section, we are suggesting future direction for PCF studies. This is not limited to CFA particles.*

P11L40      Omit "in this theme".

*Omitted.*

P12L3      Change to "On which time scale does a potential INP need to…"

*Changed.*

Fig. 1      Please change "stuff" in the figure caption to a more scientific word.

*We have changed "stuff" to "material".*

Fig. 3      Please include the actual Greek letters in the caption, not the written-out names.

*Done.*

Fig. 5      Please explain the meaning of the blue dashed line in the caption.

*We have added an explanation to the figure caption. "Figure 6: Freezing experiment data for unprocessed and processed CFA_Cy particles at 251 K and 253 K start temperatures ($T_{start}$). These data correspond to experiments #5 and #13 in Table 2, respectively. The individual panels contain the same data types as in Figure 3. The short-dashed blue lines indicate the beginning of the cloud droplet formation."*

Fig. 6      Change to "dark cyan bar" in the caption. There is only one.

*We have made changes in the caption. Instead of bars, we used columns: "The grey/black columns …" and "…cyan/dark cyan columns…". There are two "dark cyan bars" in Fig. 6.*

References (which are not already included in the manuscript):

*Thank you for pointing these references to us, we have included them at the appropriate places in the revised manuscript.*

Fisher, G. L., D. P. Y. Chang, and M. Brummer (1976). "Fly ash collected from electrostatic precipitators: microcrystalline structures and the mystery of the spheres". Science 192.4239, pp. 553–555.

*Cited and referenced.*

Losey, D., S. K. Sihvonen, D. Veghte, E. Chong, and M. A. Freedman (2018). "Acidic Processing of Fly Ash: Chemical Characterization, Morphology, and Immersion Freezing". Environmental Science: Processes & Impacts 20, pp. 1581–1592.

*Cited and referenced.*

Parungo, F. P., E. Ackerman, H. Proulx, and R. F. Pueschel (1978). "Nucleation properties of fly ash in a coal-fired power-plant plume". Atmospheric Environment 12, pp. 929–935.

*Cited and referenced.*

Seames, W. S. (2003). "An initial study of the fine fragmentation fly ash particle mode generated during pulverized coal combustion". Fuel Processing Technology 81.2, pp. 109–125.

*Cited and referenced.*

***New References not included in the original submission but cited in this response***

Ahmaruzzaman, M.: A review on the utilization of fly ash, Prog. Energy Combust. Sci., 36(3), 327–363, doi:10.1016/j.pecs.2009.11.003, 2010.

Buhre, B. J. P., Hinkley, J. T., Gupta, R. P., Wall, T. F. and Nelson, P. F.: Submicron ash formation from coal combustion, in Fuel, vol. 84, pp. 1206–1214, Elsevier., 2005.

DeMott, P. J., Cziczo, D. J., Prenni, A. J., Murphy, D. M., Kreidenweis, S. M., Thomson, D. S., Borys, R. and Rogers, D. C.: Measurements of the concentration and composition of nuclei for cirrus formation, Proc. Natl. Acad. Sci., 100(25), 14655–14660, doi:10.1073/pnas.2532677100, 2003.

Fisher, L. R., Gamble, R. A. and Middlehurst, J.: The Kelvin equation and the capillary condensation of water, Nature, 290(5807), 575–576, doi:10.1038/290575a0, 1981.

Jewell, R. B. and Rathbone, R. F.: Optical Properties of Coal Combustion Byproducts for Particle-Size Analysis by Laser Diffraction, Coal Combust. Gasif. Prod., 1, 1–7, 2009.

Liu, L., Zhang, J., Xu, L., Yuan, Q., Huang, D., Chen, J., Shi, Z., Sun, Y., Fu, P., Wang, Z., Zhang, D. and Li, W.: Cloud scavenging of anthropogenic refractory particles at a mountain site in North China, Atmos. Chem. Phys., 18(19), 14681–14693, doi:10.5194/acp-18-14681-2018, 2018.

Losey, D. J., Sihvonen, S. K., Veghte, D. P., Chong, E. and Freedman, M. A.: Acidic

processing of fly ash: chemical characterization, morphology, and immersion freezing, Environ. Sci. Process. Impacts, 20(11), 1581–1592, doi:10.1039/C8EM00319J, 2018.

Möhler, O., Field, P. R., Connolly, P., Benz, S., Saathoff, H., Schnaiter, M., Wagner, R., Cotton, R., Krämer, M., Mangold, A. and Heymsfield, A. J.: Efficiency of the deposition mode ice nucleation on mineral dust particles, Atmos. Chem. Phys., 6(10), 3007–3021, doi:10.5194/acp-6-3007-2006, 2006.

Schure, M. R., Soltys, P. A., Natusch, D. F. S. and Mauney, T.: Surface Area and Porosity of Coal Fly Ash, Environ. Sci. Technol., 19(1), 82–86, doi:10.1021/es00131a009, 1985.

Seames, W. S.: An initial study of the fine fragmentation fly ash particle mode generated during pulverized coal combustion, Fuel Process. Technol., 81(2), 109–125, doi:10.1016/S0378-3820(03)00006-7, 2003.

Sjogren, S., Gysel, M., Weingartner, E., Baltensperger, U., Cubison, M. J., Coe, H., Zardini, A. A., Marcolli, C., Krieger, U. K. and Peter, T.: Hygroscopic growth and water uptake kinetics of two-phase aerosol particles consisting of ammonium sulfate, adipic and humic acid mixtures, J. Aerosol Sci., 38(2), 157–171, doi:10.1016/j.jaerosci.2006.11.005, 2007.

Vassilev, S. V and Vassileva, C. G.: A new approach for the classification of coal fly ashes based on their origin, composition, properties, and behaviour, Fuel, 86(10–11), 1490–1512, doi:10.1016/j.fuel.2006.11.020, 2007.

---

## Referee Report (RR1)

**Review of revised manuscript by Umo et al., 2019:**

**General comment:**

The manuscript is strongly improved and the readability increased. I recommend publication after the following minor revisions.

**Specific comments:**

Page 2, line 24: It is questionable whether ice-filled pores should be termed active sites, since active sites are considered to consist of a material different from ice. Also, growth of ice on ice should not be considered a nucleation process. Please revise the formulation.

Page 3, lines 30 – 32: This sentence needs to be formulated better. It should become clear that the first statement refers to immersion freezing experiments and it should be explained what is meant by "hydratable components" (their origin and composition).

Page 4, lines 19 – 20: Can you specify the CaO content of classes C and classes F CFAs?

Page 6, lines 26 – 27: is this a correct definition of ice-activated fraction? Shouldn't the division be through the sum of all particles/droplets/crystals present in the chamber?

Page 6, lines 33 – 37: there are 3 panels in each row. It should therefore read: top panels represent … middle panels show … bottom panels show…

Page 6, lines 35 – 36: do you really mean that the freezing experiment was stopped and not just the expansion?

Page 7, line 16: "we discuss" instead of "we discussed".

Page 7, lines 21 – 22: this sentence reads as if the 0.19 % of the particles all nucleated at 244 K, Do you really mean this or not rather that about 0.19 % of the particles had nucleated ice at 244 K?

Page 7, line 23: the same issue again: do you mean "had increased…"?

Page 7, lines 27 – 28: can you give the value of the homogeneous freezing temperature that you measured in your experiments?

Page 7, line 30: why a difference of 8 K? 245 K – 228 K = 17 K, or to which numbers do you refer here?

Page 8, line 3: Either give the abbreviation of ice-activated fraction or write it out but you do not need to do both every time you mention it.

Page 8, line 20: is it only the size of the particles or also their number that is different? In drop freezing assays there are usually very many particles per drop.

Page 8, line 34: First was the size discussion, the aerosol composition is rather "second", and the measurement techniques "third". Consider to revise.

Page 8, line 36: Isono and Ikebe (1965) and Mason and Maybank (1958) are rather old references for this statement. Consider to add more recent ones.

Page 9, line 5: Consider to replace "behavior" by "activity".

Page 9, lines 21 – 31: The sequence of experiments is confusing. Table 2 suggests that the experiment with Tstart = 254 K is a new experiment. What is the history of this experiment? 250 K

→ 254 K → 264 K? If you did two such experiments, you might show the results of both in Table 2.

Page 9, line 28: Here you write that the same processed CFA_UK sample was warmed to 264 K, however, the previous section refers to a different experiment. Please make the history of the samples more transparent.

Page 10, line 10: could you mark the point where 244 K is reached in the figure e.g. with an arrow? This would increase the readability of the manuscript.

Page 10, line 27: again, could you mark the point where 246 K is reached in the figure?

Page 11, line 38: "be plerospheres" instead of "have plerospheres"

Page 12, lines 29 – 31: this sentence could be formulated better. Do you mean "processing" instead of "process"?

Page 13, line 14: "with which" instead of "that"

Page 13, line 12: "inherently expected" sounds strange. Try to improve formulation.

Page 13, lines 13 and 14: "than" instead of "that"

Page 13, line 24 – 25: "CFA will only show considerable or no ice nucleation potential." I do not understand the logics of this sentence. Please improve.

Page 13, lines 35 – 37: Relying just on pore volume and specific surface area is dangerous because the diameter is very relevant for pore filling and ice melting.

Figure caption to Fig. 1, line 5: consider to replace "irrespective" by "despite".

Figure caption to Fig. 2, lines  2 – 5: improve formulation.

---

## Author Response (AR3)

We are sincerely grateful to the two Reviewers for their interest in our work. The comments, suggestions, and corrections they have provided are very useful feedback to us, and they have gone a long way to improve the revised version of our work. Our responses to the Reviewers' comments are written in italicized **blue** text beneath each comment. The page and line numbers referred to in this response are those of the ACPD version. All changes - corrections, omissions, and additions - are updated in the revised manuscript which is submitted alongside this response. A Supplementary Information document, added to the revised manuscript version, is also submitted.

**Reviewer 1**

**Review Umo et al., 2019:**

**General comment:**

Umo et al. present ice nucleation experiments performed within the large cloud simulation chamber AIDA with different coal fly ash (CFA) samples collected from five different power plants, one situated in the UK and four in the USA. Samples were sieved to isolate the size fraction up to 20 μm diameter and characterized by environmental scanning electron microscopy. In addition, their specific surface areas and pore volume were determined by argon adsorption measurements. There were quite significant differences between the ice nucleation activities of the different CFA samples.
The UK sample, which is the best investigated one, showed a strong increase in the ice-active fraction for experiments performed just below the homogeneous freezing temperature of pure water. The authors concluded that this could be related to a pore condensation and freezing process (PCF). To further substantiate the role of pores for the ice nucleation ability of CFA, temperature cycling experiments were performed within the AIDA chamber by precooling the injected particles to 228 K at RH slightly below ice saturation before performing an expansion at warmer temperature. A strong pre-activation was found for the particles with the highest specific surface area and porosity. The authors conclude that the PCF mechanism could be prevalent for the ice nucleation at cirrus temperature and also significant for mixed-phase clouds when CFA particles are injected from higher altitudes.

This study presents innovative experiments aiming to elucidate the relevant ice nucleation mechanisms under cirrus and mixed-phase cloud conditions. Experiments were performed with CFA particles, which are a relevant class of ice nucleating particles from anthropogenic sources. The manuscript is well suited for Atmos. Chem. Phys. and can be recommended for publication after the following points have been addressed satisfactorily:

For all five CFA samples, pre-cooling experiments were described and discussed in the manuscript but only for the CFA UK sample, expansions that reached homogeneous freezing temperatures were mentioned. Have such measurements been carried out also for the CFA samples from the US? If yes, they should be described and discussed in the manuscript.

*No, expansions just below the homogeneous freezing temperatures of pure water were not carried out for the CFA samples from the USA.*

Experiments of processed and unprocessed samples with different starting temperatures are often compared without discussing the effect of the starting temperature. It would have been more meaningful if processed and unprocessed samples were compared in experiments with the same starting temperature. When such data is not available, the discussion needs to be improved to take the influence of the starting temperature better into account.
*In the manuscript, we compared results of experiments from processed and unprocessed CFA particles at similar start temperatures. In situations where this was not the case, we selected the start temperature of the processed samples which was closest to the start temperature of the unprocessed*

*samples. Moreover, we have improved our discussions taking into account the influence of the starting temperature.*

The individual experiments need to be characterized better. Table 2 gives an overview over all experiments relevant for this study; however, since the text and figures do not refer to this table, it does not help to obtain an overview over the experiments. Moreover, the table just gives the starting condition and lists no results except the observed freezing mode. The experiments are characterized by their starting temperature throughout the manuscript. Unfortunately, this information is not very useful because the RH and temperature at the freezing onset can only be guessed based on the starting temperature of the expansion. It would be helpful if the RH and the temperature of freezing onset together with $f_{ice}$ were added to Table 2. Moreover, the experiment name should be mentioned in the figure captions, so that the exact conditions can be looked up in Table 2.

*We have now referred to Table 2 in the text and figure captions of the revised manuscript where appropriate. Additionally, we have included more information in Table 2, namely (1) the maximum ice-active fraction reached during the experiment ($f_{ice,max}$) and (2) the corresponding temperatures ($T@ f_{ice,max}$) and relative humidities with respect to ice ($RH_{ice}@ f_{ice,max}$).*

Throughout the manuscript, the consistency of the use of present and past tense needs to be checked. Some sentences are hard to understand and should be clarified. Some examples are given in the special comments but the whole manuscript should be worked over.

*Thank you for your comments. We have worked through the entire manuscript and made necessary corrections, clarified, and simplified some sentences to improve their readability and understanding.*

**Specific comments:**

Page 1, line 25: what is meant here by "partly"? Homogeneous freezing temperatures were only reached with CFA UK. Is this statement based on experiments that are not shown?

*First, we have removed "partly" from this statement. You are correct, we performed experiments below homogeneous freezing temperatures only with CFA_UK. We have adjusted the statement to be more specific and it now reads: "In our current ice nucleation experiments with a particular CFA sample (CFA_UK), which we conducted in the Aerosol Interaction and Dynamics in the Atmosphere (AIDA) aerosol and cloud simulation chamber at the Karlsruhe Institute of Technology, Germany, we observed a strong increase in the ice-active fraction for experiments performed at temperatures just below the homogeneous freezing of pure water."*

Page 4, Sect. 2.5: the adsorption and desorption isotherms should be given as supplementary information.

*We have now provided the adsorption and desorption isotherms for the five CFA samples in the Supplementary Information document.*

Page 5, line 26: is the end of the freezing experiment the end of the expansion?

*For our experiments described in this manuscript - yes.*

Page 5, line 29: according to Sect. 2.3 the particles should be < 2.5 µm and not just < ~10 µm? Can you comment on this?

*The size classification of the scattering signals from the OPCs relies on certain assumptions for particle shape/orientation and refractive index. The size scale shown in this article relies on Mie calculations for a particle refractive index of 1.33. It is therefore directly applicable only to e.g. spherical water droplets that form during the expansion cooling runs. For spherical particles with a different refractive index and/or aspherical particles, the derived "optical" diameters do not agree with the "true" particle sizes measured with the SMPS and APS instruments. Concerning the CFA particles, their slightly aspherical particle habits and much larger refractive index of about 1.6*

*(Jewell and Rathbone, 2009) lead to a significant overestimation of their "true" diameters on a size scale calibrated for spherical particles with a refractive index of 1.33. Therefore, a small fraction of the CFA particles is classified at apparent diameters above the minimum cut-off size of our cyclones ($D_{50} = 2.3$ μm).*

*We have added the following clarification to P5L29: "Note that the size scale of the OPCs was calibrated for spherical particles with a refractive index of 1.33. The slightly aspherical shape and much larger refractive index of the CFA particles (Jewell and Rathbone, 2009) lead to a significant overestimation of their true diameters on this size scale. Therefore, some CFA particles are detected at apparent diameters above the minimum cut-off size of our cyclones ($D_{50} = 2.3$ μm)."*

Page 7, line 2: do you mean "at" instead of "from"?

*We have corrected this to "at".*

Page 7, lines 2 – 5: It is insinuated here that the generation method (dry vs. wet) might influence the ice nucleation activity of CFA particles. However, for a consistent comparison, the available INP area also needs to be taken into account. In Umo et al. (2015), the ice nucleation active site density does not rise above $10^5$ cm$^2$. The surface area of a spherical 1 μm radius particle is only $10^{-7}$ cm$^2$. Therefore, if the CFA particles had the same ice nucleation ability as reported in Umo et al. (2015), only a minor fraction of the particles should be active, which is indeed in accordance with the AIDA experiments.

*Yes, we agree. Also, the droplet freezing assay technique employed in Umo et al. (2015) is sensitive to INP at warmer temperatures compared to AIDA experiments, given that there is a higher chance to observe rare freezing events at warm temperatures (with small values for the ice nucleation active site density) due to the larger size of the droplets. We have extended our discussion on this issue in the revised text on P7L3-4:*

*"Both studies, however, were performed with drop freezing assay techniques and with much larger particles than reported here. Hence, the probability of observing freezing events at much warmer temperatures was higher than in the AIDA experiments where smaller particle sizes were explored. A combination of both techniques in future studies could ultimately yield a parameterization of the heterogeneous ice nucleation activity of the CFA particles over the entire range of temperatures in the mixed-phase cloud regime."*

Page 7, line 5: do you mean "at" instead of "from"?

*Yes, changed to "at".*

Page 7, lines 12 – 13: "In this work, the average median particle diameter was 0.58 μm for our CFA samples theirs was size-selected to 0.3 μm." Improve formulation.

*This sentence now reads: "The average median particle diameter of our CFA samples is 0.58 μm, whereas Grawe et al. (2016) reported an average diameter of 0.3 μm."*

Page 7, line 20: While the ice nucleation activity of the CFA particles investigated here is compared to the ones of other studies and other aerosol types, a comparison of the CFA particles investigated in this work among each other is lacking. Figure 7 shows that the ice nucleation activity of CFA_JA and CFA_Wh is one order of magnitude larger than the one of CFA_Cy and CFA_Mi. Are there differences in morphology, elemental composition, or surface functionalization that might explain the differences?

*Yes, it is true that there is a huge variation in the inherent heterogeneous ice nucleation activity amongst the various CFA particle types. To the best of our knowledge, it is not yet clear and has to be clarified in future investigations which one of the governing factors you mentioned (morphology,*

*elemental composition, surface functionalization) is the key parameter in influencing the particles' ice nucleation abilities.*

*We have included the following short paragraph to summarize and facilitate the comparison of the ice nucleation activity of the CFA particles investigated in our study: "In order to compare the inherent ice nucleation behaviour of the five CFA samples investigated, we have tabulated the maximum ice-activated fraction (%) for experiments with a similar starting temperature of about 250 K (Table 2, experiment numbers 3 & 5 - 8). The results reveal a significant spread in the ice-activated fractions, with CFA_Wh (~ 26 %) > CFA_Ja (~ 17 %) >> CFA_Cy (~ 1.5 %) = CFA_Mi (~1.5 %) > CFA_UK (~ 0.17 %). This huge variation in the particles' inherent ice nucleation activity is probably related to differences in morphology, elemental composition, and/or surface functionalization."*

Page 7, lines 31 – 33: "This occurred at a lower $RH_{ice}$ = ~105 % than the experiment with unprocessed CFA_UK particles which $RH_{ice}$ = ~130 % (corresponding to water saturation)." Improve formulation.

*This sentence now reads: "The processed CFA_UK particles nucleated ice at water-subsaturated conditions with a nucleation threshold in terms of $RH_{ice}$ of only about 101 %. In contrast, the unprocessed CFA_UK particles nucleated ice in the immersion freezing mode after exceeding water saturation during the expansion run (corresponding to $RH_{ice}$ ~ 130 %)."*

Page 8, lines 7 – 8: Was this the third expansion of the same sample or an expansion with a new sample? Please clarify.

*The expansion was performed with the same sample. The sentence now reads: "Afterwards, the same processed CFA_UK aerosol particles were warmed to $T_{start}$ = 264 K for another expansion cooling run (Fig. 4C)."*

Page 8: lines 33 – 35: "At $T_{start}$ = 253 K, the $f_{ice}$ for CFA_Cy particles after the pre-activation process was ~0.86 % slightly lower than what was observed for the unprocessed CFA_Cy particles." Improve formulation.

*The statement has been removed in the revised version.*

Page 9, lines 2 and 3: "$T_{start}$ = 255 K (Fig. 7)": Where is this starting temperature shown in Fig. 7?

*This start temperature is not indicated in Fig. 7. It is now shown in Fig. S2 in the Supplementary Information document. The statement has been removed in the revised version.*

Page 9, line 10: "256 K start temperature (Fig. 7)": Where is this starting temperature shown in Fig. 7?

*This start temperature is not indicated in Fig. 7. It is now shown in Fig. S3 in the Supplementary Information document. We have completely revised the discussions in this paragraph (please see the response to Page 9, lines 16 – 17 comments.*

Page 9, lines 10 – 12: "Again, for the processed CFA_Ja particles, no appreciable enhancement of its ice formation abilities was observed as the $f_{ice}$ at $T_{start}$ = 249 K was 2 % at $RH_{ice}$ = ~125 %." Do you mean "enhancement compared to the unprocessed CFA_Ja?" Yet, Fig. 7 shows a decrease in $f_{ice}$ rather than "no appreciable enhancement". Please clarify.

*Yes, we were referring to the enhancement compared to the unprocessed CFA_Ja particles. Now, we have completely revised the discussions in this paragraph (please see the response to Page 9, lines 16 – 17 comments).*

Page 9, lines 15 – 16: "We cannot completely rule out that the actual formation mechanism in both scenarios after the temperature cycling is not via a condensational freezing pathway." Formulate clearer, avoid double negative.

*Yes, we have completely re-phrased the paragraph describing the experiments with the CFA_Ja particles. This is described in our answer to the following comment).*

Page 9, lines 16 – 17: "This was not seen for the unprocessed CFA_Ja particles; after reaching water saturation, there was a time lag before ice particles were detected." Is this a valid comparison? According to Fig. 7, the unprocessed CFA_Ja sample had a higher starting temperature. The onset of $f_{ice}$ was therefore still at a warmer temperature for processed compared with unprocessed particles. This difference should be taken into account when discussing the effect of processing.

*This is correct – we will clarify our discussion in the revised manuscript text. At starting temperatures around 250 K, there was indeed not much change in the ice nucleation ability of the CFA_Ja particles before and after the TCF cycle (when taking into account the slightly different starting temperatures for the processed and unprocessed particle ensembles). Obviously, pre-activation cannot compete with the already very high inherent heterogeneous ice nucleation ability of the CFA_Ja particles at this temperature, meaning that there is no further detectable increase in the ice-activated fraction. The pre-activation phenomenon is then only visible when further warming the pre-activated CFA_Ja particles to a higher starting temperature (256 K, now included in the Supplementary Information document, Fig. S3, panel C). Here, the processed CFA_Ja particles showed a small nucleation mode with $f_{ice}$ ~ 1 % at 252 K just when exceeding water saturation during the expansion run. Given that the threshold temperature for exceeding an ice-activated fraction of 1 % for the unprocessed CFA_Ja particles is as low as 246 K, the observed ice nucleation mode for the processed CFA_Ja particles at 252 K can most likely be ascribed to the condensational growth of pre-existing ice, generated in the pores of the particles during the TCF cycle.*

*The revised paragraph will read as follows:*

*"Pre-activated CFA_Ja particles did not show any significant improvement of their ice nucleation ability after the temperature-cycling experiment for expansion cooling experiments started at around 250 K (Fig. 7). Obviously, pre-activation cannot compete with the already very high inherent heterogeneous ice nucleation ability of the CFA_Ja particles at this temperature, meaning that there is no further detectable increase in the ice-activated fraction after the TCF cycle. However, the pre-activation phenomenon becomes visible when further warming the pre-activated CFA_Ja particles to a higher starting temperature (256 K, Fig. S3, panel C). Here, the processed CFA_Ja particles showed a small nucleation mode with $f_{ice}$ ~ 1 % at 252 K just when exceeding water saturation during the expansion run. Given that the threshold temperature for exceeding an ice-activated fraction of 1 % for the unprocessed CFA_Ja particles was as low as 246 K, the observed ice nucleation mode for the processed CFA_Ja particles at 252 K can most likely be ascribed to the condensational growth of pre-existing ice, generated in the pores of the particles during the TCF cycle."*

*Based on our re-phrasing of the pre-activation experiments with CFA_Ja particles as discussed above, we suggest shortening the succeeding discussion of the CFA_Wh particles because their pre-activation behaviour was rather similar: We do not see any significant effect of the TCF cycle at temperatures where the particles' inherent heterogeneous ice nucleation ability is already very high. Only when further warming the pre-activated particle ensemble, a smaller ice nucleation mode, probably due to the condensational ice growth mode, becomes visible. Re-phrasing addresses the following three comments; the revised text reads as follows:*

*"Similar to the CFA_Ja particles, also the CFA_Wh particles did not significantly change their ice nucleation ability after the TCF cycle when probing them at starting temperatures of 248 K - 249 K*

*(Fig. 7), i.e., in a temperature range where the particles' inherent heterogeneous ice nucleation ability is already very high. The smaller nucleation mode with $f_{ice}$ ~ 2 % that was observed after further warming the processed CFA_Wh particles to 256 K (Fig. S4, panel C), however, is likely again due to the condensational ice growth mode. The CFA_Mi particles showed the smallest variation with respect to their ice nucleation ability after the TCF cycle. In addition to the comparable ice nucleation behaviour before and after temperature cycling at a starting temperature around 250 K (Fig. 7; Fig. S2 panels A & B), the processed CFA_Mi particles also revealed only a tiny condensational ice growth mode at a higher starting temperature of 255 K with $f_{ice,max}$ ~ 0.1 % (Fig. S2, panel C)."*

Page 9, line 26: $T_{start}$ = 256 K is not shown in Fig. 7.

Page 9, line 28 – 29: "occurred in a shorter temperature step": please formulate better.

Page 9, line 30: "pre-activation by PCF may not be very important compared to other particles that are less ice-active". Improve formulation.

Page 9, line 33: "and the relative humidity which summary is given in Figs 6 & 7." Improve formulation.

*This part of the sentence now reads: "the relative humidity as summarized in Figs. 6 & 7."*

Page 10, lines 29 – 30: An additional factor that should be discussed here is the competition between pre-activation and immersion freezing. The ice nucleation mode is given in Table 2 but this table is neither connected with the text nor with the figures. Figure 7 needs to be improved to clearly state when water saturation is reached.

*This is true, in the revised paragraphs describing the pre-activation experiments with the CFA_Ja and CFA_Wh particles (see above), we have added some discussion on the competition between pre-activation and immersion freezing.*

*Additionally, we have now better linked Table 2 to the appropriate text and figures.*

*For all the data shown in Fig. 7, ice particles were detected only after water saturation was exceeded. Detailed AIDA data of these expansion runs were added to the Supplementary Information document (Figs. S2, S3, and S4). We have also symbolized the temperatures where water saturation was reached during the expansion experiments by vertical, dotted lines in Fig. 7.*

Page 10, lines 35 – 36: "Depending on the transport of CFA particles in the atmosphere, they can pass through different altitudes and temperature regimes which can naturally provide a temperature-cycling and freezing process for these particles to be pre-activated." Improve formulation.

*This statement was changed to: "In the atmosphere, CFA particles can be transported through different relative humidity and temperature regimes. This can provide a natural temperature-cycling and freezing process and lead to the pre-activation of these particles."*

Page 11, line 8: "There is a need by the modelling community to study the impact that…". Improve formulation.

*This sentence part now reads: "We suggest that future modeling work should focus on the impact that ..."*

Page 11, line 26: This is the first and only time that the chemical compositions of CFA particles is mentioned in this manuscript. Indeed, the chemical composition might be relevant to explain the

differences in immersion freezing of the different CFA samples. If chemical composition is mentioned in the conclusions, it should also be discussed in the section "Results and Discussion".

*Absolutely, we did not show any results on the chemical compositions of the CFA samples that were explored in this study. Although Garimella (2016) gave information on the different classes of the USA CFA samples which were based on chemical composition but the details of such compositions were not reported. Referring to P11L26, we have removed chemical compositions from that line.*

*We have now added the statements below to Section 3.1.*

*"The observed differences in their inherent ice-nucleating abilities may also be due to variabilities in their chemical and mineralogical compositions. Garimella (2016) reported that the four CFA samples from the USA belonged to different classes of fly ash and these groupings are based on the chemical compositions (Garimella, 2016)."*

Page 18, caption of Table 1: How was the median diameter determined?

*We added: "The median diameter was determined from the combined data of the APS and the SMPS instruments."*

Page 18, Table 2: Consider to add $f_{ice}$ and the freezing onset temperature to Table 2.

*$f_{ice,max}$, $RH_{ice}$@ $f_{ice,max}$, and $T_{ice\ onset}$ have been added in Table 2.*

Figure 3: the tags on the y-axes should be increased for better visibility. The measurements shown in this figure should be related to the experiments listed in Table 2.

*We have increased the labels on the y-axes and have related this figure to the experiments listed in Table 2. This correction has been applied to all similar figures.*

Figure 6, figure caption: It should be made clear whether the start temperatures of the experiments are shown in this figure.

*We have added the following statement to the figure caption: "The temperatures referenced on the x-axis are the temperature at which the maximum ice-activated fraction was reached during each experiment."*

Figure 7: It would be helpful to indicate the temperature where water saturation is reached for all experiments. Consider to add experiments with CFA_UK for better comparison.

*We have now indicated the temperature where water saturation occurred in the experiments.*

**Technical comment:**

Page 4, line 10: "range" might be more adequate than "limit" since a range is given in brackets.
*Changed to "range".*

Page 4, line 28: "microscope" instead of "microscopy".
*Changed to "microscope".*

Page 5, line 31: "β" does not appear correctly in the pdf.
*Sorted.*

Page 5, line 33: "γ" does not appear correctly in the pdf.

*Sorted.*

Page 8, line38: "their" instead of "its".

*This line has been removed in the revised version.*

Page 9, line 29: remove "an".

*Done.*

Page 10, line 14: "than those of other CFA particles" instead of "than the other CFA particles".

*Corrected.*

Page 12, line 3: "need" instead of "needs".

*Corrected.*

Page 14, line 26: John is the first name of J. G. Morris. Please revise reference.

*Done.*

**Reviewer 2**

**Review of "Enhanced ice nucleation activity of coal fly ash aerosol particles initiated by ice-filled pores" by N. S. Umo et al. for Atmospheric Chemistry and Physics**

General comments:

In the present study, N. S. Umo and co-workers show and discuss the results of temperature cycling experiments with coal fly ash (CFA) particles at the AIDA cloud chamber. The aim of these experiments was to clarify whether the ice nucleation activity of CFA is increased by the pore condensation and freezing (PCF) mechanism under certain conditions. The authors achieve to convincingly demonstrate that this is the case for some of the used samples. The question of why some samples are more prone to PCF than others is not convincingly answered, but this cannot be expected in the case of CFA, which is a very complex and heterogeneous substance. From my point of view, the study is an interesting addition to recent findings concerning the ice nucleation behavior of CFA particles (Umo et al., 2015; Grawe et al., 2016; Garimella, 2016; Grawe et al., 2018; Losey et al., 2018). However, there are some content-related issues which need more discussion or clarification from my point of view. These are listed below (specific comments). Parts of the manuscript could benefit from editing with respect to wording and presentation of the results but this can be easily resolved (see technical corrections). The figures are mostly clear, but I was wondering why a significant part of the results are not shown anywhere. The authors should consider including an Appendix for presenting the missing figures. Generally, I feel that the topic fits the scope of ACP and that the study is worth publishing. To improve the overall significance and readability of the manuscript, I suggest the following minor points.

*Thank you for your kind interest in reviewing our work. We appreciate the painstaking efforts in giving this manuscript a thorough review, and providing us with very useful comments, suggestions, and corrections, which will greatly improve our work. We totally agree with your suggestion to include an Appendix for showing missing plots. We have now included a Supplementary Information document to the revised manuscript. We have worked through your comments and responded to each point accordingly.*

Specific comments:

1)      Effect of size-dependent specific surface area and pore properties:

The investigation of PCF on CFA particles is a daunting task, since CFA is such a heterogeneous substance. I appreciate the authors' attempt at finding possible reasons for differences in the behavior of the different samples, but I also think that their approach lacks the discussion of an important point, i.e., the size-dependence of the particles' specific surface area, pore volume, and pore size. This type of information is probably very hard to come by and I do not expect the authors to perform further analyses. But it should at least be mentioned that the specific surface area and pore volume are very likely dependent of the particle size and that some particles might not even feature pores (Seames, 2003). Hence, the properties of the sieved bulk sample (0-20 µm) might not be representative for the properties of the particles that actually enter the AIDA chamber (< 2.5 µm). This should be made clear in the discussion (P9L37-P10L30).

*We completely agree that making specific surface area (SSA), pore volume and pore size measurements of size-selected CFA particles would be a great step forward for this type of investigation. But as mentioned, CFA particles are very complex and heterogeneous in nature. We thought about making such measurements but it is highly difficult to collect a sufficient amount of aerosolized CFA particles in the required size range (i.e. < 3 µm) for BET measurements; and we do not have access to an appropriate size-selection instrument. There is no doubt that in some cases one*

*can see a dependence of SSA, pore size, and pore volume on size e.g. (Schure et al., 1985). Also, we have mentioned the occurrence of cenospheres and plerosphere in CFA particles which makes it even more difficult to attribute certain properties to the bulk material. Most probably, there are cenospheres and plerospheres in our samples but we cannot estimate the percentage of particles in our sample which show these phenomena. It should also be noted that this can also change depending on the handling of such samples.*

*As suggested, we have now included a statement that reads: "In previous studies, it has been shown that the specific surface area and pore volume of fly ash particles generated from pulverized coal combustion are very likely dependent on the particle size (Schure et al., 1985; Seames, 2003)."*

2)      Comparison to Wagner et al. (2016)

When viewing the results, I was wondering how CFA particles compare to other the substances which have already been investigated with the same instrumental setup and measurement routine by Wagner et al. (2016), i.e., zeolite, diatomaceous earth, mineral dust, volcanic ash, and soot. Concerning CFA_UK, it is mentioned very briefly (P6L19-20) that "a similar increase in the heterogeneous ice nucleation ability has been previously observed for zeolite and illite" but nothing is said about the other CFA samples and other substances tested by Wagner et al. (2016). This comparison could be expanded to put the CFA results into perspective.

*Thank you for pointing this out. We have now included a new Section (Section 3.3) to compare the ice nucleation enhancement by CFA particles reported in this study with other particles previously studied with a similar measurement routine at the AIDA chamber. The text of the new Section is presented below.*

*3.3      Ice nucleation enhancement by CFA particles versus other particle types*

*In a previous study, Wagner et al. (2016) investigated the pre-activation behaviour of INPs by the PCF mechanism in the AIDA cloud chamber with a similar measurement routine as described in Section 2.6. In this study, a wide range of INPs was tested including illite NX, diatomaceous earth, zeolites, dust samples from Canary Island, Sahara and Israel, Graphite Spark Generator soot (GSG soot), and volcanic ash (Wagner et al., 2016). It was reported that illite NX, diatomaceous earth, and mesoporous zeolite CBV 400 showed a significant ice nucleation enhancement in the depositional ice growth mode, with ice-active fractions of 5.9 %, 3.8 %, and 3.7 % at a starting temperature of ~ 250 K (Fig. 8). At higher starting temperatures, the ice-activated fractions in the condensational ice growth mode were typically around 1 %. Another group of INPs such as CBV 100 (untreated microporous zeolites), Canary Island dust, and GSG soot showed much smaller depositional ice growth modes with ice-activated fractions below 1 %. Finally, volcanic ash, water-processed GSG soot, as well as Saharan and Israeli dust particles did not show any enhancement after the pre-activation process, neither in the depositional nor the condensational ice growth mode.*

*In this context, the ice nucleation enhancement observed for the CFA_UK particles at a starting temperature of 250 K in the depositional growth mode with $f_{ice,max}$ ~ 11 % (Fig. 4A) is by far the highest value for any particle type investigated so far (Fig. 8). In contrast, the pre-activation efficiency of the CFA particles from the US power plants is comparable in magnitude to the above-mentioned group of CBV100, Canary Island dust, and GSG soot particles with much lower ice-activated fractions. The mean diameters of the particles investigated by Wagner et al. (2016) ranged from 0.21 μm to 0.43 μm, and were thus smaller than the mean diameters of our CFA particles except for CFA_Mi (0.42 μm). Different pore sizes, morphology, and chemical composition of these INPs may control their susceptibility to the PCF pre-activation mechanism. More studies are required to investigate the role that each of these parameters play.*

3)      Atmospheric implications

Although the authors discuss the atmospheric implications of their findings, they are missing one major point. How large is the probability that CFA particles reach such high altitudes where they experience 228 K? I understand that CFA particles can influence atmospheric ice nucleation close to the point of emission, but the number concentration of these particles at cirrus level is probably close to zero. Indeed, it is difficult to identify CFA particles in the atmosphere due to their similarities to mineral dust which is why there is a lack of information concerning atmospheric number concentrations of CFA particles. But despite this lack of information, the authors should not leave this issue completely unattended. A remark concerning this should be included in Sec. 3.3.

*There are a couple of studies that have shown fly ash as a composition of ice residues in cirrus clouds and mixed-phase clouds e.g.* (DeMott et al., 2003; Liu et al., 2018). *With these pieces of evidence, we can indeed assume that CFA particles can reach higher altitudes with temperatures down to 228 K. We agree with you that there is still a lack of information on the actual number concentration of CFA particles in the atmosphere. As already suggested in Umo et al. (2015), given that the compositions of CFA particles are similar to typical mineral dust particles, one could speculate that some ice residue measurements are wrongly attributed to mineral dust instead of CFA. At the moment, this is still uncertain and there is an urgent need for in-depth source apportionment studies. We assume that in some areas, where these particles are emitted and transported, their concentration may be higher than other INPs, but at the moment we cannot quantify it.*

*We have now included the following statement in the text (Section 3.4): "Despite the dearth of information on the number concentration of CFA particles in the atmosphere at higher altitudes, there are pieces of evidence that CFA particles are found in ice residues of cirrus and mixed-phase clouds (DeMott et al., 2003; Liu et al., 2018)."*

4) Explanation of the PCF mechanism

- Even though PCF has become an accepted concept in recent years, the process itself should be explained in more detail. The negative Kelvin effect, which is the reason why there is capillary condensation of water vapor at a relative humidity (RH) below water saturation, is not even mentioned. I suggest to include an explanation of the mechanism (capillary condensation of water vapor at RH below water saturation → formation of ice in pores at very low temperatures → pore ice persists as site for ice nucleation at warmer temperatures and ice-supersaturated conditions) where it is first mentioned (P2L14).

*We have added a brief explanation of the PCF mechanism to the introduction Section of the manuscript as suggested. The explanation reads: "PCF involves a two-step process – first, capillary condensation of liquid water in the particle pores, and second, freezing of the condensed water. The first step occurs when particles with pores are exposed to a certain relative humidity ($RH_w$) below water saturation ($RH_w < 100$ %). The $RH_w$ for pore filling to occur is well-described by the 'negative' Kelvin effect* (Fisher et al., 1981). *The negative exponential term of the Kelvin equation accounts for the concave meniscus of the condensed water in a pore* (Sjogren et al., 2007). *When pores with condensed water (step 1) are exposed to sufficiently low temperatures, ice can form in such pores. Ice-filled particle pores can then act as active sites for ice nucleation and growth in an ice-supersaturated environment. In a situation where ice-filled pores (step 2) are preserved even when the system is warmed, they can trigger ice nucleation at warmer temperatures. This process is relevant for understanding ice nucleation by porous particles or particles with surface defects."*

- At this point, it should also be mentioned, why PCF is restricted to certain pore sizes (P2L29). Furthermore, I expected a remark concerning the effect of the pore geometry (cylindrical pore, inkbottle-shaped pore) on PCF. Which types of pores might be present in CFA? All of this needs to be explained in the introduction.

*We have now added the information below to the introduction.*

*"The PCF mechanism is restricted to a certain pore size range due to limitations related to the negative Kelvin effect for water condensation in the pores and the size of the critical ice embryo for ice nucleation and melting. According to classical nucleation theory, a certain critical ice embryo size is required to overcome the energy barrier defined by the Gibbs free energy* (Pruppacher and Klett, 2010). *Therefore, the pore size should be large enough to accommodate such a critical ice embryo and small enough to enable the capillary condensation of water in the first place. Calculations and previous reports have shown that pore sizes with 3 – 8 nm diameter are suitable for the PCF mechanism (Wagner et al, 2016, Marcolli, 2017). Also, pore geometry (e.g., cylindrical or ink-bottle-shaped pores) has been shown to be an important parameter for the initial step of the PCF mechanism* (Marcolli, 2014, 2017). *Moreover, the contact angle between the pore wall and the water curvature affects the onset of the capillary condensation of water according to the Kelvin equation."*

*Unfortunately, we do not have any information on the pore geometries of our CFA particles. Considering the uniqueness, heterogeneity, and complexity of the CFA particles, it is even more challenging using some models to predict it.*

- Instead of Fig. 2, the authors could describe the temperature cycling process in a similar manner as Marcolli (2017; see Fig. 1-4), i.e., showing RH with respect to temperature. The time scales of the different steps could be mentioned in the caption.

*We have looked at the illustrations presented in Marcolli (2017) which show the $RH_{ice}$ trajectories as a function of temperature for different assumed pore types. These graphs are indeed a very good representation of typical atmospheric trajectories for well-known pore geometries. But note that the trajectories of our temperature-cycling experiments, however, would just be a horizontal line in such graphs at $RH_{ice}$ close to 100 %, given that the ice-coating on the inner chamber walls controlled $RH_{ice}$ to an almost constant value close to saturation throughout the entire TCF cycle. The illustration that we presented in Figure 2 was meant to show the procedure of our experiments and not to serve as a model. We specifically put out Figure 2 to show the experimental routine that we performed in AIDA as a simple guide for other researchers who may be interested in repeating the type of experiments that we have reported here. The plot is not meant to present an idealized conceptualization of the PCF mechanism with respect to CFA particles. Based on these reasons, we would prefer to leave it that way.*

5) Methodology

- The AIDA chamber is a well-established instrument but the authors should consider describing the measurement and data evaluation techniques in more detail. For example, there is no mention of the uncertainty of the $f_{ice}$ determination. How does the large error of ±20 % of the ice particle number concentration affect the $f_{ice}$ error?

*We added this information is Section 3 in the manuscript but based on the suggestion below, it has now been moved to Section 2.6 in the revised manuscript.*

- The explanation of how $f_{ice}$ is calculated should be included in Sec. 2.6, not in Sec. 3.

*We have moved this part to Section 2.6 and also added that the uncertainty estimation is ~ ± 20 %* (Möhler et al., 2006).

- I do not understand how droplets can form in an environment which is slightly subsaturated with respect to liquid water (see Fig. 3, panels A and B). It looks like the black line (RH with respect to water) is below 100 % throughout the duration of the experiments. Is this due to the measurement uncertainty? Please clarify.

*Yes, this is due to measurement uncertainty which we report as ~ ± 5 %. We mentioned this in Section 2.2.*

- P5L17-18: How can the air in the AIDA chamber be subsaturated with respect to ice when there is an ice layer on the inner chamber walls? Shouldn't it be saturated? Please explain.

*The air was indeed a few percent below ice-saturation. The reason is that we usually observe that the gas temperature in the AIDA chamber is a few tenths of a Kelvin warmer than the wall temperature. This is probably due to some internal heat sources in the interior of the chamber like e.g. heated sampling lines. We have now added the statement below to Section 2.6. "The slight sub-saturation of the chamber air with respect to ice may be attributed to some internal heat sources which increased the gas temperature by a few tenths of a Kelvin compared to the wall temperature (Wagner et al., 2016)."*

6) Presentation of the results

- I was wondering why ice nucleation surface site densities are not included in the discussion of the results. The authors state that "Further analyses on the distribution of the ice nucleation active sites densities of these CFA particles is outside the scope of the current report and will be presented in a separate communication." (P6L32-34). An inclusion of these data in the current paper would make more sense to me, especially for a comparison of the intrinsic ice nucleation behavior of the CFA samples to the results other studies (P6L35-P7L21). Besides, which new information would this other report contain?

*The ultimate goal of the here presented work is to report the ice nucleation enhancement by CFA particles after a pre-activation by PCF. We understand the need to discuss the INAS concept but that is not the focus of this work. But as suggested also by Referee #1, we have added the following paragraph to compare the inherent IN behaviour of the CFA samples.*

*"In order to compare the inherent ice nucleation behaviour of the five CFA samples investigated, we have tabulated the maximum ice-activated fraction (%) for experiments with a similar starting temperature of about 250 K (Table 2, experiment numbers 3 & 5 - 8). The results reveal a significant spread in the ice-activated fractions, with CFA_Wh (~ 26 %) > CFA_Ja (~ 17 %) >> CFA_Cy (~ 1.5 %) = CFA_Mi (~1.5 %) > CFA_UK (~ 0.17 %). This huge variation in the particles' inherent ice nucleation activity is probably related to differences in morphology, elemental composition, and/or surface functionalization."*

*Regarding the new information in the other report, we have suggested a combination of the drop freezing assay technique with AIDA expansion cooling runs to explore the ice nucleation behaviour of the CFA particles over an extended temperature range and have added the statement:*

*"A combination of both techniques in future studies could ultimately yield a parameterization of the heterogeneous ice nucleation activity of the CFA particles over the entire range of temperatures in the mixed-phase cloud regime."*

- The mentioning of $T_{start}$ instead of the actual temperature at which $f_{ice,max}$ values were derived is not intuitive (see P6L27-28, Sec. 3.2, …). Please include both $T_{start}$ and the actual $T$ in your discussion.

*The actual temperatures at the $f_{ice,max}$ are now included in the discussion and also used in the summary plots.*

- As stated above, parts of Sec. 3.2 are hard to follow. In the cases of CFA_Mi, CFA_Ja, and CFA_Wh, the authors describe specific features observed in the experiments, but they do not show the corresponding 3-panel plots. Some results are not even included in Fig. 7. I suggest to include the described examples in an Appendix so that the reader can follow the discussion.

*To make this Section easier to follow and also based on the concerns of Reviewer #1, we have completely revised this Section to provide a more succinct summary. We have also provided a Supplementary Information document containing the missing results.*

- I find the amount of arrows, overlapping shapes, and different colors in Fig. 8 very confusing. What is the difference between the blue, red, light gray and dark gray arrows? What is the difference between the black, light gray, and dark gray particles? A legend, or at least an explanation in the caption, would be helpful. Also, it looks like the sublimation of ice particles directly leads to cloud formation. The authors should revise and thin out this figure to make it more intuitively understandable.

*Figure 8 (now Fig. 9) has now been revised and a legend has been included with more explanation in the caption.*

- Why is the CFA_UK sample discussed separately from the U.S. American samples? Fig. 6 and 7 should be combined. I also suggest to change from a bar graph to a scatter plot for more clarity. A change to a logarithmic y-axis should be considered for both Fig. 6 and 7.

*The pre-activation efficiency of the CFA_UK was really "outstanding" and we started with its description. Whereas the pre-activation behaviour of the USA samples was much different from the CFA_UK sample but rather similar among the various USA samples. Notwithstanding, we have provided a ranking to compare all the CFA samples in Sections 3.1 and 3.2. Consequently, we prefer to keep Figures 6 (now Fig. 5) and 7 separately but we have changed Figure 7 to a scatter plot as suggested. The ordinates of both plots are now on a logarithmic scale. The reason is to clearly illustrate the changes in the CFA_UK after the pre-activation process.*

Technical corrections:

I generally feel that the authors need to be more precise with their formulations. Sentences and paragraphs are sometimes lengthy due to unnecessary fillers and repetitions. Transition words are partly misleading. There are grammatical errors. Below, I list the issues that caught my eye but I advise the authors to recheck their manuscript carefully to improve readability and understandability.

*Thank you for your comments. In response, we have carefully rechecked the manuscript and improved our sentence formulations. In addition, Section 3.2 was completely revised and shortened.*

P1L26        At which RH was this strong increase observed?

*This strong increase was observed at $RH_{ice} = 101 - 105$ %. This information has been added to the sentence and now reads "…we observed a strong increase (at a threshold relative humidity with respect to ice of 101 – 105 %) in the ice-active fraction for experiments performed at temperatures just below the homogeneous freezing of pure water."*

P1L27        Change to either "undergoing PCF" or "undergoing **the** PCF mechanism".

*Changed.*

P1L31-35      This sentence would benefit from being split into two.

*Done.*

P1L36-39     It is a bit unfortunate that you refer to PCF in general in the first sentence and specifically to PCF on CFA particles in the second. At least this is how I understood it. Please reword.

*We have switched the order of importance communicated in this sentence. It now reads: "On the one hand, the PCF mechanism can play a significant role in mixed-phase cloud formation in a case where the CFA particles are injected from higher altitudes and then transported to lower altitudes after being exposed to lower temperatures. On the other hand, the PCF mechanism could be the prevalent nucleation mode for ice formation at cirrus temperatures rather than the previously acclaimed deposition mode."*

P1L36       I suggest to introduce "intrinsic" as relating to unprocessed or not pre-activated particles.

*We have removed this word.*

P1L37       Change to "on the other **hand**".

*Changed.*

P1L41 Change to "highly relevant to **our understanding/knowledge/ comprehension… of** cloud formation".

*Changed to "highly relevant to our knowledge of cloud formation…".*

P2L3   Omit "primary". Heterogeneous ice nucleation is always a primary process. Also, define the terms "homogeneous" and "heterogeneous" before using them.

*We have omitted the word "primary". The terms "homogeneous" and "heterogeneous" are now defined in the text. We have also added Vali et al. (2015) here as a reference.*

P2L8       Change to "with **the surface of** a …".

*Changed.*

P2L11      Omit "however" at the start of the new paragraph.

*Omitted.*

P2L12-13     Change to either "such **a** particle" or "such particles".

*Changed to "such particles".*

P2L19      Change to "before ice nucleation **takes place**".

*Changed.*

P2L20      Change to "Here, we define…".

*Changed.*

P2L21 There are references to Wagner et al. (2016a), Wagner et al. (2016b), and Wagner et al. (2016). Yet, there's only one Wagner et al. paper from 2016 in the reference list. Please correct.

*Corrected.*

P2L27          "zeolite" and "illite" should not be capitalized.

*The two words are now in lowercase letters.*

P2L34-36      This sentence does not say anything else than the one on P2L16-18. I suggest to remove it (also the following sentence).

*We have removed the sentence and added the references to the statement on P2L16-18.*

P2L39          Change to "global **cloud** ice budget".

*Changed.*

P2L41 A "significant amount" is not very precise. Are there really no estimates of the emitted CFA mass?

*We would have liked to be more precise about the amount of CFA emitted directly into the atmosphere but unfortunately, we do not have that information.*

P3L4   There is no question that immersion freezing was investigated by Grawe et al. (2016, 2018) and Umo et al. (2015). Hence, differences in the freezing behavior of the investigated samples are not due to differences in the freezing mechanism. Please remove this part of the sentence. Furthermore, Grawe et al. (2018) showed that the immersion freezing behavior of CFA can be strongly dependent on the amount of time that the particles spend immersed in the droplet prior to the initiation of freezing. This issue is worth mentioning here because it can also affect the immersion mode AIDA measurements.

*We have removed the part of the sentence that reads: "...; as well as variabilities in the actual freezing mechanisms, which could be influenced by surface defects or porosity of the particles." The statement now reads: "However, there are variabilities in the ice nucleation activities of the different CFA samples reported, which could be due to the difference in mineralogical compositions, and the extent to which these particles are processed in the atmosphere (Grawe et al., 2018; Losey et al., 2018)."*

P3L6          "various atmospheric conditions". Be more precise.

*We have rephrased this as: "...various temperature and relative humidity conditions."*

P3L7          Change to "different CFA **samples**". Also on P3L14.

*Changed.*

P3L15-17      "The results from these new laboratory measurements are presented in this report." This sentence is unnecessary and should be deleted.

*Deleted.*

P3L23-25      How representative is material from the EPs in comparison to material that is emitted into the atmosphere. Please include a statement concerning this matter.

*We have now included this statement in the text: "The CFA particles collected from EPs are the same particles that could have been directly released into the atmosphere in situations where EPs malfunction or are inefficient. Also, the CFA particles which are emitted indirectly into the atmosphere by road transportation, application in agricultural fields, industrial sites, road construction, and other sources are the same CFA particles as collected from the EPs (Buhre et al., 2005)."*

P3L25    Omit "However".

*Omitted.*

P3L25-29    I am aware that the authors do not focus on the effect of chemical composition of the samples on their ice nucleation behavior. However, it would be interesting to include some more information, e.g., are the samples of class C (high Ca) or class F (low Ca), since it has been shown that the composition affects the ice nucleation measurements. This information is easily obtainable from Garimella (2016) and should be mentioned here.

*We have included a few sentences to make reference to the CFA classification as follows: "Garimella (2016) grouped CFA_Ja and CFA_Wh fly ash samples as class C type, while CFA_Cy and CFA_Mi are class F which is broadly based on the calcium oxide (CaO) composition (Ahmaruzzaman, 2010). A new CFA standard classification system suggests that CFA samples can be sialic (S), calsialic (CS), ferrisialic (FS), and ferricalsialic (FCS)* (Vassilev and Vassileva, 2007). *However, no further information on chemical composition was provided by Garimella (2016) for a more quantitative classification of the USA CFA samples."*

P3L29 Please note that Losey et al. (2018) investigated the same sample set. This publication should be referenced here as well.

*Thank you for pointing us to the very interesting work by Losey et al. Losey et al. (2018) stated that the CFA samples which were used in their study are from the same four power plants in the USA that we also studied. They also obtained the CFA samples from Fly Ash Direct®. However, we would be careful to state that Losey and co-workers investigated the* *same sample set* *as we did in this work. This is because we do not have any evidence that they used the same batch of CFA samples that we also obtained. CFA samples from a power plant can differ due to many factors including the batch of the coal fuel burned, operating conditions, etc. We have therefore cited the Losey et al. work in the introduction but not at the particular place on P3L29.*

P3L30 Change "name" to "operators/owners". The name itself cannot prefer anonymity.

*Changed to "Operator".*

P3L31-33    Were the other samples not sieved? Why not?

*Yes, all raw CFA samples were sieved to the same diameter size fraction (0 – 20 μm). The sentence now reads: "First, all raw CFA samples were sieved with a Fritsch Sieve set-up (Analysette 3, 03.7020/06209, Germany) to obtain 0 – 20 μm diameter size fractions, which were later used for the experiments."*

P4L3    Please combine the instrument abbreviation and the manufacturer in one set of parentheses to avoid "(…) (…)". Check throughout the manuscript.

*Done.*

P4L20 Which types of cyclones were used? What is their cut-off diameter?

*We used custom-made cyclones that work in a similar way as cyclones 2 and 3 of a five-stage series cyclone (EPA, USA). Cyclone 2 ($D_{50}$ cut-off = 3.7 µm) and cyclone 3 ($D_{50}$ cut-off = 2.3 µm). We have added this sentence to Section 2.3: "Cyclone 2 ($D_{50}$ cut-off = 3.7 µm) was placed before cyclone 3 ($D_{50}$ cut-off = 2.3 µm) in the set-up.*

P4L26 Change "Min" to min".

*Changed.*

P4L27 Change "mins" to "min".

*Changed.*

P4L27 How does this coating affect the morphology of the particles? Could pores potentially be covered? Please include a short statement.

*The coating thickness was 1 nm and thus below the SEM resolution. Coating is a well-known and standard procedure used in scanning electron microscopy. To the best of our knowledge, we do not know any report of its potential influence on the morphology of scanned particles. Specifically, in our experiment we did not see any unusual occurrence on our samples; hence, we do not think that pores were masked by the coating process. We have added this sentence: "Coating of the filters did not affect the morphology of our samples because the coating thickness was 1 nm and thus below the SEM resolution."*

P4L36 "argon" and "nitrogen" should not be capitalized in running text.

*Corrected.*

P5L21 "inherent" and "intrinsic" seem to be used synonymously throughout the manuscript. I suggest to avoid the use of both terms and stick to one.

*We are now using "inherent".*

P5L31,33     Make sure that the empty squares are replaced by the Greek letters in the new version of the manuscript.

*Done.*

P5L35 Since you list all experiments in Table 2, you could also include the used size thresholds there.

*The average median diameters of the CFA particles for each of the samples investigated are already given in Table 1.*

P6L11 Change "$t =\sim 300$ s" to "$t \sim 300$ s". Also in all other occurring instances.

*Changed.*

P6L25 Insert "-" behind "CFA_Wh".

*We do not think a dash is required there. Instead, we have inserted a hyphen after CFA_Wh.*

P6L26 Insert "mode" behind "immersion freezing".

*Inserted.*

P6L35-

P7L21 I agree that a comparison to previous ice nucleation studies with CFA is interesting. However, by only reporting onset ice nucleation temperatures, the reader does not get an idea how the here

investigated samples compare to those from previous studies. This could be resolved by including a figure showing ns(T).

*We have addressed this issue in our answer to your specific comment #6 "Presentation of the results" and outlined the changes made to the manuscript text.*

P7L3-4 "Both studies can access warmer freezing temperatures for INPs than the dry generation method that our system is designed for.". Please explain shortly why this is the case.

*We have modified this sentence to read "Both studies, however, were performed with drop freezing assay techniques and with much larger particles than reported here. Hence, the probability of observing freezing events at much warmer temperatures was higher than in the AIDA experiments where smaller particle sizes were explored. A combination of both techniques in future studies could ultimately yield a parameterization of the heterogeneous ice nucleation activity of the CFA particles over the entire range of temperatures in the mixed-phase cloud regime."*

P7L5 The Schnell et al. (1976) reference is not a good choice here. Actually, in this study "no detectable effects from a coal-fired powerplant plume" (see title) on atmospheric ice nucleation were found. Better cite Parungo et al. (1978), who conducted a similar experiment and found an enhanced ice nucleation efficiency of the plume aerosol in comparison to the background aerosol.

*Thank you for pointing this out. We have corrected this reference to Parungo et al., 1978.*

P7L10 Which of the two studies are you referring to?

*We were referring to the Grawe et al (2018) study. The sentence now reads "Grawe et al. (2018) partly attributed the ice nucleation behaviour of..."*

P7L10 Actually, Grawe et al. (2018) state that the amount of hydratable components is important and that quartz only contributes in those samples which contain a small concentration of hydratable components. But this could also be included in the introduction (P3L4).

*We have rephrased this sentence and moved it to P3L4. The statement now reads "Grawe et al. (2018) partly attributed the ice nucleation behaviour of the CFA particles to the quartz composition of the CFA; however, the influence of quartz can be suppressed in a situation where hydratable components form a layer on the particle surface."*

P7L13 Please cite Garimella (2016) instead of Welti et al. (2009). Garimella (2016) investigated 300 and 700 nm CFA particles, not mineral dust, and found that the immersion freezing efficiency does not scale with the surface area as the smaller particles were relatively more efficient than the larger ones.

*Cited.*

P7L17 Please cite Grawe et al. (2018) instead of Hiranuma et al. (2018). Firstly, the manuscript by Hiranuma et al. (2018) is still under review. Secondly, the study by Grawe et al. (2018) is more relevant for the here presented work as they discuss the methodology-dependent freezing behavior of CFA particles, not cellulose.

*Cited.*

P8L11-13 Does this mean that 1.2 % of the particles contained pores suitable for PCF? If yes, then please say so.

*At this start temperature (264 K), the result indeed implies that at least 1.2 % of the particles are ice-active via the PCF mechanism. The fraction could even be higher because this was not the first expansion after the temperature-cycling process (TCF). We have added this statement to P8L13:*

*"This implies that at least 1.3 % of the processed CFA_UK particles still contained ice-filled pores even after warming to 264 K."*

P8L14 Please check the Mahrt et al. (2018) reference. Mahrt et al. (2018) indeed describe PCF, but they did not conduct temperature cycling experiments. They saw a stepwise increase in the activated fraction of one type of soot particles due to condensation in pores and subsequent homogeneous freezing.

*We referenced Mahrt et al. (2018) because they described the condensational growth in soot particles, but they did not conduct temperature-cycling experiments. Due to the confusion that this reference might create here, we have removed it.*

P8L23 Figure 6 is discussed before Fig. 5. Please arrange the order of the attached figures accordingly.

*Figures 6 and 5 are now rearranged.*

P8L38-39    Please check this sentence for correctness. It does not relate to the previous statement and it seems that something is missing.

*During revisions, this sentence has been removed.*

P9L5-7    Change beginning of the sentence to "**This** suggests..".

*We have removed this sentence in the revised version of the manuscript.*

P9L7-8    "Confirm" is a very strong word here, given that the data of this example is not even shown and given that no error estimation for the $f_{ice}$ error is provided.

*We have removed this sentence in the revised version of the manuscript. All missing data are now available in the Supplementary Information document.*

P9L9   Change to "droplet activation". Also on P9L17.

*Changed.*

P9L16 Please omit "One thing is extremely clear that …".

*Omitted.*

P9L19-30    This paragraph would profit tremendously from the inclusion of a 3-panel-plot of the measurement that you are describing here.

*The 3-panel-plot is now available in the Supplementary Information document.*

P9L25 I do not understand the use of the word "although" here.

*We have removed the word "although".*

P9L28-29    "ice formation occurred in a shorter temperature step". I am not sure what you mean here. The temperature at which $f_{ice,max}$ was registered is of interest, not the temperature range. Please reword.

*We have reworded this part to read: "ice formation occurred at the instant of the expansion process for the processed CFA_Wh (Fig. 7 and Fig. S4C).*

P9L32 Change "comparison" to "ranking".

*Changed.*

P9L33 Change "$f_{ice}$" to "$f_{ice,max}$".

*Changed.*

P9L33 "which summary". Change to form a grammatically correct sentence.

*Changed. The sentence now reads: "The ranking is based on the start temperature, $f_{ice,max}$ and the relative humidity as summarized in Figs. 5 & 7.*

P10L21-23    Please reword this sentence.

*The sentence now reads "As another example, CFA_Cy (0.012 cm$^3$ g$^{-1}$) has a PV similar to the CFA_Mi sample (0.013 cm$^3$ g$^{-1}$), but only the processed CFA_Cy particles showed a clear pre-activation ability due to the PCF mechanism."*

P10L25    Is it realistic that cenospheres or plerospheres would be filled by capillary condensation under the conditions in your experiments? How large are the spheres and how large are the openings? According to Marcolli (2017), large pores need very high RH or very low temperatures to be filled.

*The behaviour strongly depends on the size of the spheres and openings. Unfortunately, we do not have the size information of the cenospheres or plerospheres for our samples and cannot further resolve this issue at the moment.*

P10L26    Fischer et al. (1976), who first discovered cenospheres and plerospheres, is the more appropriate reference here.

*We have added Fischer et al. (1976) to the existing references.*

P9L27-30    I suggest to change the formulation in such a way that it becomes clear that an estimation of the pore size and geometry is **not possible** for CFA. This is due to the heterogeneity of the particles. Other INP types might be better suited for such an estimation. Avoid using the word "pointless".

*We would like to avoid using the word "impossible" in our text. We have reformulated the sentence to read "Currently, it is highly difficult to estimate the pore sizes based on the PV of the CFA samples except in the case of a well-defined pore model and morphology."*

P10L37    "Rainout" is probably not the right term for cases where only the ice phase is involved.

*We have removed "rainout".*

P10L37-38    I cannot find this statement in the given reference. Please check and remove if necessary.

*Hong et al. (2004) is a reference supporting an atmospheric sedimentation process. We have moved it to the next sentence that lists the various atmospheric processes that could aid re-circulations of particles in the atmosphere.*

P11L3-4    This is shown in a very confusing way in Fig. 8. Please adjust.

*Figure 8 (now Fig. 9) has been adjusted.*

P11L12-13    "Currently, this is not well understood and requires further research." This sentence can be omitted. The following sentence is completely sufficient.

*Removed.*

P11L14    Change "clouds formation" to "cloud formation".

*Changed.*

P11L15  Change "studying the dominance of this occurrence" to simply "the occurrence".

*Changed.*

P11L16  Change "cloud system" to "cloud system**s**".

*Changed.*

P11L21  "However" does not seem like an appropriate transition word. The statement is not in contrast to the previous one. Please reword.

*We have changed "however" to "Also".*

P11L22-23 I am aware that there are lots of different pore types ("pores", "crevices", "cavities"), but it might be best to define one term in the beginning and stick to it throughout the manuscript.

*We have changed "cavities" to "pores". We have also included a definition of our pores in Section 1.*

P11L25  Change "ice formation" to "ice nucleation potential".

*Changed.*

P11L33  What is meant by "their"? The particles? The pores? Please clarify.

*We have clarified this by modifying part of the sentence to read "the particle's overall ice-nucleating efficiencies".*

P11L35-36 It should be mentioned here that CFA is not a suitable substance for the investigation of the effect of different pore geometries on PCF.

*We would prefer not to make this statement at this point because CFA particles could still be used for the investigation of the effect of pore geometries on the PCF mechanism in the future. Currently, we just do not know the prevalent geometry of the pores or the actual pore size of these particles. If future technology permits such properties to be well characterized, we think it could be a suitable material. Also, in the second part of the conclusion Section, we are suggesting future direction for PCF studies. This is not limited to CFA particles.*

P11L40  Omit "in this theme".

*Omitted.*

P12L3 Change to "On which time scale does a potential INP need to…"

*Changed.*

Fig. 1 Please change "stuff" in the figure caption to a more scientific word.

*We have changed "stuff" to "material".*

Fig. 3 Please include the actual Greek letters in the caption, not the written-out names.

*Done.*

Fig. 5 Please explain the meaning of the blue dashed line in the caption.

*We have added an explanation to the figure caption. "Figure 6: Freezing experiment data for unprocessed and processed CFA_Cy particles at 251 K and 253 K start temperatures ($T_{start}$). These*

*data correspond to experiments #5 and #13 in Table 2, respectively. The individual panels contain the same data types as in Figure 3. The short-dashed blue lines indicate the beginning of the cloud droplet formation."*

Fig. 6  Change to "dark cyan bar" in the caption. There is only one.

*We have made changes in the caption. Instead of bars, we used columns: "The grey/black columns …" and "…cyan/dark cyan columns…". There are two "dark cyan bars" in Fig. 6.*

References (which are not already included in the manuscript):

*Thank you for pointing these references to us, we have included them at the appropriate places in the revised manuscript.*

Fisher, G. L., D. P. Y. Chang, and M. Brummer (1976). "Fly ash collected from electrostatic precipitators: microcrystalline structures and the mystery of the spheres". Science 192.4239, pp. 553–555.

*Cited and referenced.*

Losey, D., S. K. Sihvonen, D. Veghte, E. Chong, and M. A. Freedman (2018). "Acidic Processing of Fly Ash: Chemical Characterization, Morphology, and Immersion Freezing". Environmental Science: Processes & Impacts 20, pp. 1581–1592.

*Cited and referenced.*

Parungo, F. P., E. Ackerman, H. Proulx, and R. F. Pueschel (1978). "Nucleation properties of fly ash in a coal-fired power-plant plume". Atmospheric Environment 12, pp. 929–935.

*Cited and referenced.*

Seames, W. S. (2003). "An initial study of the fine fragmentation fly ash particle mode generated during pulverized coal combustion". Fuel Processing Technology 81.2, pp. 109– 125.

*Cited and referenced.*

***New References not included in the original submission but cited in this response***

Ahmaruzzaman, M.: A review on the utilization of fly ash, Prog. Energy Combust. Sci., 36(3), 327–363, doi:10.1016/j.pecs.2009.11.003, 2010.

Buhre, B. J. P., Hinkley, J. T., Gupta, R. P., Wall, T. F. and Nelson, P. F.: Submicron ash formation from coal combustion, in Fuel, vol. 84, pp. 1206–1214, Elsevier., 2005.

DeMott, P. J., Cziczo, D. J., Prenni, A. J., Murphy, D. M., Kreidenweis, S. M., Thomson, D. S., Borys, R. and Rogers, D. C.: Measurements of the concentration and composition of nuclei for cirrus formation, Proc. Natl. Acad. Sci., 100(25), 14655–14660, doi:10.1073/pnas.2532677100, 2003.

Fisher, L. R., Gamble, R. A. and Middlehurst, J.: The Kelvin equation and the capillary condensation of water, Nature, 290(5807), 575–576, doi:10.1038/290575a0, 1981.

Jewell, R. B. and Rathbone, R. F.: Optical Properties of Coal Combustion Byproducts for Particle-Size Analysis by Laser Diffraction, Coal Combust. Gasif. Prod., 1, 1–7, 2009.

Liu, L., Zhang, J., Xu, L., Yuan, Q., Huang, D., Chen, J., Shi, Z., Sun, Y., Fu, P., Wang, Z., Zhang, D. and Li, W.: Cloud scavenging of anthropogenic refractory particles at a mountain site in North China, Atmos. Chem. Phys., 18(19), 14681–14693, doi:10.5194/acp-18-14681-2018, 2018.

Losey, D. J., Sihvonen, S. K., Veghte, D. P., Chong, E. and Freedman, M. A.: Acidic processing of fly ash: chemical characterization, morphology, and immersion freezing, Environ. Sci. Process. Impacts, 20(11), 1581–1592, doi:10.1039/C8EM00319J, 2018.

Möhler, O., Field, P. R., Connolly, P., Benz, S., Saathoff, H., Schnaiter, M., Wagner, R., Cotton, R., Krämer, M., Mangold, A. and Heymsfield, A. J.: Efficiency of the deposition mode ice nucleation on mineral dust particles, Atmos. Chem. Phys., 6(10), 3007–3021, doi:10.5194/acp-6-3007-2006, 2006.

Schure, M. R., Soltys, P. A., Natusch, D. F. S. and Mauney, T.: Surface Area and Porosity of Coal Fly Ash, Environ. Sci. Technol., 19(1), 82–86, doi:10.1021/es00131a009, 1985.

Seames, W. S.: An initial study of the fine fragmentation fly ash particle mode generated during pulverized coal combustion, Fuel Process. Technol., 81(2), 109–125, doi:10.1016/S0378-3820(03)00006-7, 2003.

Sjogren, S., Gysel, M., Weingartner, E., Baltensperger, U., Cubison, M. J., Coe, H., Zardini, A. A., Marcolli, C., Krieger, U. K. and Peter, T.: Hygroscopic growth and water uptake kinetics of two-phase aerosol particles consisting of ammonium sulfate, adipic and humic acid mixtures, J. Aerosol Sci., 38(2), 157–171, doi:10.1016/j.jaerosci.2006.11.005, 2007.

Vassilev, S. V and Vassileva, C. G.: A new approach for the classification of coal fly ashes based on their origin, composition, properties, and behaviour, Fuel, 86(10–11), 1490–1512, doi:10.1016/j.fuel.2006.11.020, 2007.

*Minor Revisions*

Dear Reviewers,

Thank you very much for your careful reading and the detailed comments and recommendations on our revised manuscript: *"Enhanced ice nucleation activity of coal fly ash aerosol particles initiated by ice-filled pores"*. Please find our responses to the minor revisions in blue text beneath each comment. We have also marked the corrections with a blue text in our latest submission.

**Report 1**

**Review of revised manuscript by Umo et al., 2019:**
**General comment:**

The manuscript is strongly improved and the readability increased. I recommend publication after the following minor revisions.

Thank you.

**Specific comments:**

Page 2, line 24: It is questionable whether ice-filled pores should be termed active sites, since active sites are considered to consist of a material different from ice. Also, growth of ice on ice should not be considered a nucleation process. Please revise the formulation.

The sentence has been revised as follows: *"In an ice-supersaturated environment, these ice-filled pores can then initiate the growth of macroscopic ice crystals on the particles."*

Page 3, lines 30 – 32: This sentence needs to be formulated better. It should become clear that the first statement refers to immersion freezing experiments and it should be explained what is meant by "hydratable components" (their origin and composition).

We have reformulated this sentence as: *"Grawe et al. (2018) partly attributed the ice nucleation behaviour of the CFA particles in the immersion freezing mode to the quartz content of the CFA particles. The influence of this quartz content on the particles' immersion freezing ability can be suppressed in a situation where hydratable components form a layer on the particle surface (Grawe et al., 2018). These hydratable components are chemical compounds (e.g. $CaSO_4$) contained in CFA particles that are capable of taking up water at elevated ambient relative humidity. This can lead to the formation of new compounds such as calcite and gypsum."*

Page 4, lines 19 – 20: Can you specify the CaO content of classes C and classes F CFAs?

We have added a sentence to indicate the percentage of CaO contents in classes C and F of CFA particles. *"A typical mass fraction of CaO in Class F CFA particles is ~ 1 – 12 wt.%, whereas Class C has higher CaO contents, sometimes up to 40 wt.%."*

Page 6, lines 26 – 27: is this a correct definition of ice-activated fraction? Shouldn't the division be through the sum of all particles/droplets/crystals present in the chamber?

Yes, this is true. All ice-activated fractions presented in the manuscript were also computed according to the above definition. The erroneous statement in lines 26/27 has been modified to: *"The fraction of ice frozen (i.e., the ice-activated fraction, $f_{ice}$) was calculated as the number of ice particles detected divided by the total number of seed aerosol particles present in the chamber (Vali, 1971)."*

Page 6, lines 33 – 37: there are 3 panels in each row. It should therefore read: top panels represent … middle panels show … bottom panels show…

Changed.

Page 6, lines 35 – 36: do you really mean that the freezing experiment was stopped and not just the expansion?

It was just the expansion that was stopped at that point. We have corrected this sentence to: *"The point where the pressure starts rising indicates when the expansion was stopped."*

Page 7, line 16: "we discuss" instead of "we discussed".

Changed.

Page 7, lines 21 – 22: this sentence reads as if the 0.19 % of the particles all nucleated at 244 K, Do you really mean this or not rather that about 0.19 % of the particles had nucleated ice at 244 K?

We have revised this sentence as: *"However, at $T_{start}$ = 253 K, about 0.19 % of the particles had nucleated ice via the immersion freezing mode in the course of the expansion cooling run until the minimum temperature of 244 K was reached (Fig. 3B)."*

Page 7, line 23: the same issue again: do you mean "had increased…"?

Here, we wanted to compare the detected ice-active fraction to that observed in the preceding experiment started at 253 K. The sentence now reads: *"The ice-active fraction encountered during the expansion cooling run at $T_{start}$ = 245 K was by a factor of 10 higher compared to the run started at 253 K."*

Page 7, lines 27 – 28: can you give the value of the homogeneous freezing temperature that you measured in your experiments?

We have added the homogenous freezing temperature that we observed in our experiments (i.e. referring to Fig. 3C). The sentence now reads: *"The homogeneous freezing threshold temperature observed in our experiments (237.0 K) agreed with previous reports (Benz et al., 2005; Schmitt, 2014)."*

Page 7, line 30: why a difference of 8 K? 245 K – 228 K = 17 K, or to which numbers do you refer here?

Here, we wanted to refer to the homogeneous freezing temperature of 236 K (now specified as 237.0 K, see above). We have revised this sentence as follows: *"This means that within a change of only 9 K from the homogeneous freezing temperature of pure water (237 K) to the expansion run started at 228 K, the ice-active fraction of the CFA_UK particles increased by almost 2 orders of magnitude."*

Page 8, line 3: Either give the abbreviation of ice-activated fraction or write it out but you do not need to do both every time you mention it.

Corrected.

Page 8, line 20: is it only the size of the particles or also their number that is different? In drop freezing assays there are usually very many particles per drop.

Yes, we will add the following statement after line 21: *"Moreover, in a drop freezing assay method, a droplet can contain many particles, whereas each cloud droplet activated in the AIDA chamber only contains a single particle."*

Page 8, line 34: First was the size discussion, the aerosol composition is rather "second", and the measurement techniques "third". Consider to revise.

We have now put the ordinal numbers (first, second, and third) in the right order (P8L38 to P9L1-6).

Page 8, line 36: Isono and Ikebe (1965) and Mason and Maybank (1958) are rather old references for this statement. Consider to add more recent ones.

We have added the following recent studies to the list – *"Fitzner et al., 2015; Harrison et al., 2016; Lupi et al., 2014"*

Page 9, line 5: Consider to replace "behavior" by "activity".

Changed.

Page 9, lines 21 – 31: The sequence of experiments is confusing. Table 2 suggests that the experiment with Tstart = 254 K is a new experiment. What is the history of this experiment? 250 K → 254 K → 264 K? If you did two such experiments, you might show the results of both in Table 2.

We performed our first temperature-cycling and freezing experiments with CFA_UK following $T_{start}$ ~ 250 K → 254 K → 264 K. Following the surprising and exciting results, we decided to repeat this experiment and confirm our observations, but this time with $T_{start}$ ~ 251 K → 254 K → 263 K. We have now added the complete data from both experiments in Table 2. Since we obtained similar results from both experiments, we have consistently based our discussion on the data from the first set of our experiments (i.e. Experiment #9, #10 and #11). We have replaced the data in Fig. 4B and 5 with the corresponding data of CAINIC _19 ($T_{start}$~254 K). For the sake of completeness, we have added a figure (Figure S2) showing the data of the second temperature-cycling process (Experiment #12, #13 and #14) to the Supplementary Information document.

In the revised manuscript, we have deleted the confusing statement on line 22 ("We conducted two independent experiments …") and clarified the experimental procedure by adding the following statement to P9L10:

*"Specifically, we conducted two independent series of experiments, each with a fresh load of aerosol particles, following the sequences $T_{start}$ ~ 250 K → 254 K → 264 K (series I, experiment #9, #10 and #11, data shown in Fig. 4) and $T_{start}$ ~ 251 K → 254 K → 263 K (series II, experiment #12, #13, and #14, data shown in Fig. S2). As the results from both series are very similar, we focus our discussion on the experiments conducted during series I."*

Page 9, line 28: Here you write that the same processed CFA_UK sample was warmed to 264 K, however, the previous section refers to a different experiment. Please make the history of the samples more transparent.

We have addressed this issue in our response to your previous comment.

Page 10, line 10: could you mark the point where 244 K is reached in the figure e.g. with an arrow? This would increase the readability of the manuscript.

The point where the temperature reached 244 K has been marked.

Page 10, line 27: again, could you mark the point where 246 K is reached in the figure?

The point where the temperature reached 246 K has been marked.

Page 11, line 38: "be plerospheres" instead of "have plerospheres"

Changed.

Page 12, lines 29 – 31: this sentence could be formulated better. Do you mean "processing" instead of "process"?

We have reformulated this sentence to: *"During their residence time in the atmosphere, the CFA particles can be transported through different relative humidity and temperature regimes. If the particles were temporarily exposed to temperatures below 237 K at high ambient relative humidity, their ice nucleation ability might improve by the formation of ice-filled pores."*

Page 13, line 14: "with which" instead of "that"

Changed.

Page 13, line 12: "inherently expected" sounds strange. Try to improve formulation.

Here, we think you are referring to P13L1. We have modified this sentence to: *"By convective atmospheric dynamics, these pre-activated particles could then be released to lower altitudes and trigger ice formation at higher temperatures than expected from their inherent ice nucleation ability."*

Page 13, lines 13 and 14: "than" instead of "that"

Changed.

Page 13, line 24 – 25: "CFA will only show considerable or no ice nucleation potential." I do not understand the logics of this sentence. Please improve.

We have corrected part of this sentence to read: *"…CFA will show very poor or no ice nucleation potential at all."*

Page 13, lines 35 – 37: Relying just on pore volume and specific surface area is dangerous because the diameter is very relevant for pore filling and ice melting.

Definitely, the diameter of pores is a key parameter for pore-filling/ice melting, that is why we need to get the estimate right. Here, we suggest that in a situation where the diameter of pores cannot be directly and accurately measured, models that are used for such estimation should take into consideration both the pore volume and the specific surface area of such aerosol particles. We think that this type of model will be more useful and give a better estimate than current models that just rely on cylindrical- or spherical-shaped pore assumptions. We have modified this statement to read:

*"We also suggest that in order to overcome the bias associated with pore models in estimating pore sizes and diameters for natural aerosol particles, a parameter based on the pore volume, pore size/diameter, and specific surface area should be adopted."*

Figure caption to Fig. 1, line 5: consider to replace "irrespective" by "despite".

Replaced.

Figure caption to Fig. 2, lines 2 – 5: improve formulation.

Part of the Figure caption now reads: *"A schematic showing the temperature-cycling and freezing (TCF) process adopted in our experiments. The temperatures indicated by the grey circles represent*

*the start temperatures ($T_{start}$) for the ice nucleation experiments conducted after the warming of the AIDA chamber. For each CFA sample, only a subset of the indicated starting temperatures was chosen to conduct the expansion cooling runs (see Table 2). The start temperature of the successive experiment was individually selected based on the degree of activity observed in the previous freezing experiment. ..."*

**Report 2**

P2L28: Change "fissure" to "fissures".

Changed.

P4L18: Change to "This is the same set of samples…".

Changed.

E.g., P6L28-29, P7L11, SI P2L13, …: In some text passages, "&" is used instead of "and". I advise to avoid the use of the ampersand for consistency.

Changed.

P8L24-26: Please check Parungo et al. (1978). They found an enhanced ice nucleation potential of particles from a coal-fired power plant plume sampled on filters. This observation is reported for a temperature range between 263 and 253 K. Please revise the passage accordingly.

We have revised this passage accordingly: *"In another study, particles in a plume from a coal-fired power plant were not considered ice active at temperatures above 253 K (Schnell et al., 1976). However, when similar experiments were conducted at a higher supersaturation, the particles' ice nucleation ability increased, indicating that CFA particles could act as good INPs even at temperatures as high as 263 K (Parungo et al., 1978). However, in these experiments, not many details on the exact experimental conditions are available for a direct comparison with our experiments."*

P11L15: Change to "but that a network of … is necessary".

Changed.

P11L34: Change to "Another example is CFA Cy, which has…"

Changed.

**References**

*Fitzner, M., Sosso, G. C., Cox, S. J. and Michaelides, A.: The Many Faces of Heterogeneous Ice Nucleation: Interplay Between Surface Morphology and Hydrophobicity, , doi:10.1021/JACS.5B08748, 2015.*

*Harrison, A. D., Whale, T. F., Carpenter, M. A., Holden, M. A., Neve, L., O'Sullivan, D., Vergara Temprado, J. and Murray, B. J.: Not all feldspars are equal: a survey of ice nucleating properties across the feldspar group of minerals, Atmos. Chem. Phys., 16(17), 10927–10940, doi:10.5194/acp-16-10927-2016, 2016.*

*Lupi, L., Hudait, A. and Molinero, V.: Heterogeneous Nucleation of Ice on Carbon Surfaces, J. Am. Chem. Soc., 136(8), 3156–3164, doi:10.1021/ja411507a, 2014.*

*Schnell, R. C., Van Valin, C. C. and Pueschel, R. F.: Atmospheric ice nuclei: No detectable effects from a coal-fired powerplant plume, Geophys. Res. Lett., 3(11), 657–660, doi:10.1029/GL003i011p00657, 1976.*